# Determining the effects of paternal obesity on sperm chromatin at histone H3 lysine 4 tri-methylation in relation to the placental transcriptome and cellular composition

**Anne-Sophie Pepin[1], Patrycja A Jazwiec[2], Vanessa Dumeaux[3], Deborah M Sloboda[2,4,5]\*, Sarah Kimmins[1,6]\***

[1]Department of Pharmacology and Therapeutics, Faculty of Medicine, McGill University, Montreal, Canada; [2]Department of Biochemistry and Biomedical Sciences, McMaster University, Hamilton, Canada; [3]Departments of Anatomy & Cell Biology and Oncology, Western University, London, Canada; [4]Farncombe Family Digestive Health Research Institute, McMaster University Hamilton, Hamilton, Canada; [5]Departments of Obstetrics and Gynecology, and Pediatrics, McMaster University, Hamilton, Canada; [6]Department of Pathology and Molecular Biology, University of Montreal, University of Montreal Hospital Research Center, Montreal, Canada

**\*For correspondence:**
sloboda@mcmaster.ca (DMS);
sarah.kimmins@mcgill.ca (SK)

**Competing interest:** The authors declare that no competing interests exist.

**Abstract** Paternal obesity has been implicated in adult-onset metabolic disease in offspring. However, the molecular mechanisms driving these paternal effects and the developmental processes involved remain poorly understood. One underexplored possibility is the role of paternally induced effects on placenta development and function. To address this, we investigated paternal high-fat diet-induced obesity in relation to sperm histone H3 lysine 4 tri-methylation signatures, the placenta transcriptome, and cellular composition. C57BL6/J male mice were fed either a control or high-fat diet for 10 weeks beginning at 6 weeks of age. Males were timed-mated with control-fed C57BL6/J females to generate pregnancies, followed by collection of sperm, and placentas at embryonic day (E)14.5. Chromatin immunoprecipitation targeting histone H3 lysine 4 tri-methylation (H3K4me3) followed by sequencing (ChIP-seq) was performed on sperm to define obesity-associated changes in enrichment. Paternal obesity corresponded with altered sperm H3K4me3 at promoters of genes involved in metabolism and development. Notably, altered sperm H3K4me3 was also localized at placental enhancers. Bulk RNA-sequencing on placentas revealed paternal obesity-associated sex-specific changes in expression of genes involved in hypoxic processes such as angiogenesis, nutrient transport, and imprinted genes, with a subset of de-regulated genes showing changes in H3K4me3 in sperm at corresponding promoters. Paternal obesity was also linked to impaired placenta development; specifically, a deconvolution analysis revealed altered trophoblast cell lineage specification. These findings implicate paternal obesity effects on placenta development and function as one potential developmental route to offspring metabolic disease.

## Editor's evaluation

In this interesting report, the authors probe the correlative effect of paternal obesity on placental development. In the substantively revised manuscript, the authors provide solid epigenomic data arising from dietary differences in fat intake in the male, subsequent chromatin changes in sperm,

and link these outcomes to transcriptomic changes in offspring placenta. These useful insights will be of interest to those studying the epigenetic transmission of germline changes in the next generation, and those studying the relationship to metabolic exposure in the parental generation to offspring.

## Introduction

The placenta is an extraembryonic organ that regulates fetal growth and development, and contributes to long-term adult health (*Regnault et al., 2002*). This complex tissue arises from the differentiation of distinct cell subtypes important for its functions. In the mouse, the cells that give rise to the placenta, the trophectoderm (TE) cell lineage, first appear in the preimplantation blastocyst at embryonic day 3.5 (E3.5). Blastocyst implantation begins at E4.5, triggering a cascade of paracrine, endocrine, and immune-related events that participate in endometrial decidualization. Cells of the TE overlying the embryonic inner cell mass serve as a source of multipotent trophoblast stem cells that diversify as a result of spatially and epigenetically regulated transcriptional cascades, giving rise to specialized trophoblast subtypes. The first placental fate segregation is between the extraembryonic ectoderm and ectoplacental cone (EPC). Cells of the EPC in direct contact with the decidua give rise to the cells with invasive and endocrine capacity, including trophoblast giants (TGCs), glycocen trophoblast, and spongiotrophoblast (SPT). Cells of the chorion will produce two layers of fused, multinucleated syncytiotrophoblast (SynT-I and SynT-II) and sinusoidal TGCs. From E8.5, the embryonic allantois becomes fused with the chorion, permitting invagination of mesoderm-derived angiogenic progenitors that form the basis of the placental vascular bed (*Hemberger et al., 2020*). Together, these cells form a transportive interface, the placental labyrinth zone, which is functionally critical for sustaining fetal growth throughout gestation (*Rossant and Cross, 2001*). Interhemal transfer between maternal and fetal circulation begins at E10.5, and by E12.5, all terminally differentiated cell types of the mature placenta are present.

Genetic studies of placental development using mouse mutants have identified key genes for development, differentiation, maintenance, and function (*Perez-Garcia et al., 2018*; *Rossant and Cross, 2001*). For example, homeobox transcription factors are required for trophoblast lineage development (e.g. genes *Cdx2, Eomes*) (*Chawengsaksophak et al., 1997*; *Chen et al., 2013*; *Ciruna and Rossant, 1999*; *Kunath et al., 2004*; *Russ et al., 2000*), and maintenance of SPT requires the genes *Ascl2* and *Egfr* (*Guillemot et al., 1995*; *Guillemot et al., 1994*; *Sibilia and Wagner, 1995*; *Tanaka et al., 1997*; *Threadgill et al., 1995*). Another family of genes that have been shown to play critical roles in proper placenta development are imprinted genes. There exists 228 imprinted genes identified in humans and 260 in mice; many are strongly expressed in the placenta (*Coan et al., 2005*; *Morison et al., 2005*; *Tucci et al., 2019*; *Wang et al., 2013*). Genomic imprinting refers to monoallelic gene expression that is dependent on whether the gene was inherited maternally or paternally (*Ferguson-Smith et al., 1991*). The expression of imprinted genes is regulated by DNA methylation, acting in concert with chromatin modifications, such as histone H3 lysine 4 tri-methylation (H3K4me3) and histone H3 lysine 9 di-methylation (H3K9me2) (*Dindot et al., 2009*; *McEwen and Ferguson-Smith, 2010*; *Wen et al., 2008*). Importantly, disruption of placental imprinting is associated with placentation defects and aberrant fetal growth (*McMinn et al., 2006*; *Monk, 2015*; *Zadora et al., 2017*).

Placental defects can result in obstetrical complications such as pre-eclampsia, stillbirth, preterm birth, and fetal growth restriction (*Brosens et al., 2011*). Intrauterine growth restriction (IUGR) in turn is associated with a heightened risk for adult-onset cardiometabolic diseases, coronary heart disease, and stroke, supporting a placental role in long-term health of offspring (*Brodszki et al., 2005*; *Crispi et al., 2010*; *Cruz-Lemini et al., 2016*; *Eriksson et al., 2016*; *Menendez-Castro et al., 2018*; *Mierzynski et al., 2016*; *Morsing et al., 2014*; *Sarvari et al., 2017*). Despite the many adverse pregnancy outcomes involving placental defects, the molecular and cellular factors that impact placental development are poorly understood (*Naismith and Cox, 2021*; *Perez-Garcia et al., 2018*). Until recently, most studies on the origins of placental pathology have focused on maternal factors. For example, placental insufficiency occurs in 10–15% of pregnancies, and underlying causes include advanced maternal age (*Ales et al., 1990*; *Torous and Roberts, 2020*; *Wu et al., 2019*), hypertension (*Krielessi et al., 2012*), obesity (*Delhaes et al., 2018*; *Lutsiv et al., 2015*; *MacInnis et al., 2016*; *Mission*

*et al., 2015*; *Sohlberg et al., 2012*), cigarette smoking (*Pintican et al., 2019*), drug and alcohol use, and medications (*Sebastiani et al., 2018*). However, emerging studies indicate that the paternal pre-conception environment including diet and obesity also play a critical role in placental development and offspring health (*Binder et al., 2015a*; *Binder et al., 2012*; *Jazwiec et al., 2022*; *Lambrot et al., 2013*). Notably, genetic manipulation studies have determined that the paternal genome has greater potential for extraembryonic and trophoblast development compared to the maternal genome, and paternally expressed genes dominate placenta gene expression (*Barton et al., 1985*; *Barton et al., 1984*; *Mcgrath and Solter, 1986*; *McGrath and Solter, 1984*; *Surani et al., 1984*; *Wang et al., 2013*). Additionally, we recently showed that sperm chromatin profiles highly resemble that of TE and placenta tissue, and H3K4me3-enriched regions in sperm are expressed in these tissues, highlighting a molecular connection from the male germline to the developing placenta (*Pepin et al., 2022*).

The connection between paternal gene expression and placenta development has led to a growing interest in the role of paternal factors in placental development and function and offspring health (*Wang et al., 2013*). In mice, we demonstrated that paternal folate deficiency was associated with an altered sperm epigenome, differential gene expression in the placenta, and abnormal fetal development (*Lambrot et al., 2013*). In other mouse models, advanced paternal age and toxicant exposure have been linked to altered placental imprinting and reduced placental weight, and paternal obesity was linked to alterations in placental DNA methylation, aberrant allocation of cell lineage to TE (*Binder et al., 2012*; *Binder et al., 2015b*; *Denomme et al., 2020*; *Ding et al., 2018*). Male partner metabolic syndrome and being overweight have been associated with an increased risk for pre-eclampsia and negative pregnancy outcomes (*Lin et al., 2022*; *Murugappan et al., 2021*).

In two research settings using different obesity mouse models, we studied male mice exposed to either a preconception control (CON) or a high-fat diet (HFD), and the paternal effects on offspring. As we described in our paternal obesity model (*Pepin et al., 2022*), sperm from obese males had aberrant H3K4me3 enrichment at genes implicated in metabolism and placenta development, and sired offspring with metabolic disturbances (*Pepin et al., 2022*). Here, we used sperm and placenta from a similar paternal obesity model described in *Jazwiec et al., 2022*, where we demonstrated paternal offspring born to obese fathers have impaired whole body energetics and are glucose intolerant. In addition, we showed that pregnancies sired by obese males were associated with placenta showing characteristics of hypoxia, accompanied by histo-morphological changes in blood vessel integrity. Specifically, histological and immunohistology analyses of placentas collected at E14.5 and E18.5 in *Jazwiec et al., 2022*, revealed that paternal obesity resulted in significant changes in

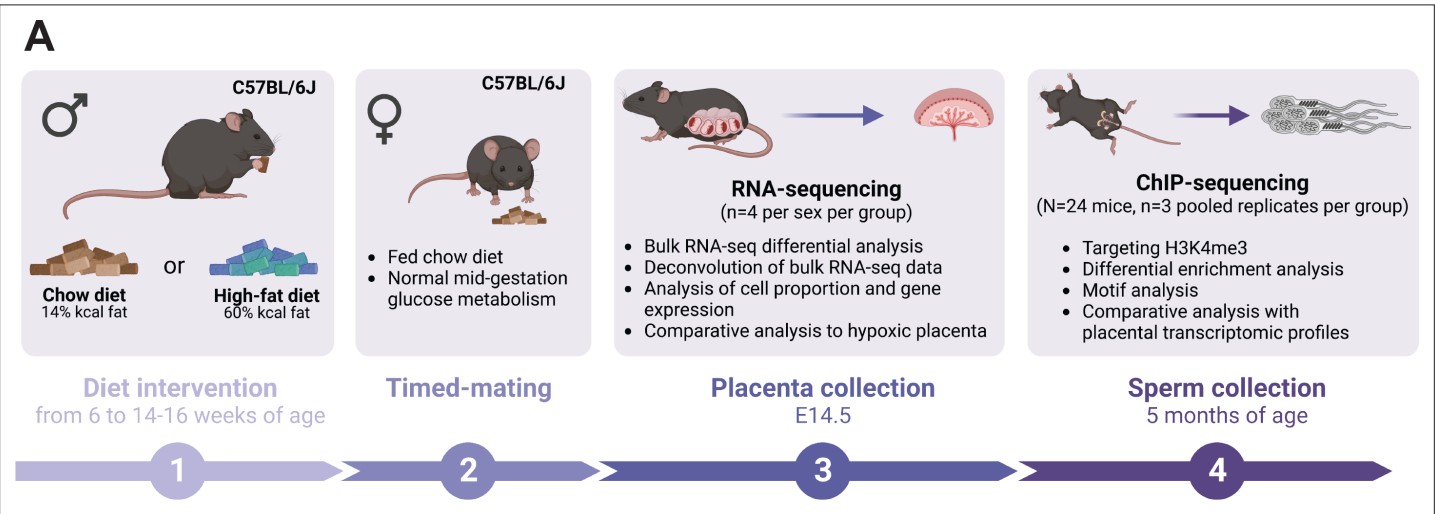

**Figure 1.** Experimental design showing the timeline and methods used to study the consequences of an obesity-induced altered sperm epigenome on the placenta. (**A**) Six-week-old C57BL/6J sires were fed either a control or high-fat diet (CON or HFD, respectively) for 8–10 weeks. Males were then time-mated with CON-fed C57BL/6J females to generate pregnancies. Pregnant females were sacrificed at embryonic day (E)14.5 and placentas were collected to perform RNA-sequencing (RNA-seq, n=4 per sex per dietary group). Sires were sacrificed at 5 months of age and sperm from cauda epididymides was collected for chromatin immunoprecipitation sequencing (ChIP-seq, n=3 replicates per dietary group) targeting histone H3 lysine 4 tri-methylation (H3K4me3). Figure created with BioRender.com.

transcription factors that regulate blood vessel development, blood vessel integrity, and signaling pathways governed by hypoxia (HIF1A, VEGF, and VEGFR-2).

In the current study, we aimed to explore the relationship between the sperm epigenome and placenta gene expression. Our key finding was identifying that paternal obesity was linked with transcriptomic defects in the placenta suggesting that the placenta is implicated in paternal transmission of metabolic disease.

## Results

### HFD-induced obesity alters the sperm epigenome at regions implicated in metabolism and cellular stress

Here, we used the same pre-conception paternal obesity model as described in *Jazwiec et al., 2022*, where offspring were previously phenotyped for metabolic function and placenta was histopathologically characterized (*Figure 1*). Of note, this model does not compare diets that are experimentally controlled for macro- or micronutrients, but instead is used to assess the effects of obesity on the sperm epigenome and its impacts on placental development. First, we determined whether sperm H3K4me3 was similarly affected in the *Jazwiec et al., 2022*, model in comparison to *Pepin et al., 2022*. Sperm from CON- (n=8) and HFD-fed (n=16) sires was profiled using chromatin immunoprecipitation followed by sequencing (ChIP-seq) targeting H3K4me3 (*Supplementary file 1a*). Analysis of sequencing data revealed 35,184 regions enriched for H3K4me3 (*Figure 2—figure supplement 1A*; Materials and methods) that were highly concordant across samples (*Figure 2—figure supplement 1B*). Principal component analysis (PCA) on counts at sperm H3K4me3-enriched regions revealed separation of samples along principal component 1 (PC1) according to dietary treatment (*Figure 2—figure supplement 1C*). For downstream analyses, we focused on the top 5% regions (n=1760; *Figure 2*) and the top 10% regions (n=3519) contributing to PC1, as we considered these to be most sensitive to obesity (*Figure 2—figure supplement 1C*). Of the top 5% and top 10% differentially enriched H3K4me3 (deH3K4me3) regions, some were common to those previously identified in *Pepin et al., 2022* (128 and 423 overlapping deH3K4me3 regions, Fisher's exact test p=2.2e-16 and p=2.2e-16, *Figure 2—figure supplement 1Di and ii*, respectively). This was despite substantial differences in animal models (timing of diet exposure [3 versus 6 weeks of age], control diet [chow versus low-fat diet], and mouse substrain [C57BL/6J versus C57BL/6NCrl]). Focusing on the top 5% regions most impacted by obesity, the majority of regions showed an increase in enrichment for H3K4me3, consistent with our previous study (71.4%, n=1257 versus n=503, *Figure 2A and B*). Regions losing H3K4me3 showed moderate H3K4me3 enrichment in CON sperm, with predominantly low CpG density, whereas regions gaining H3K4me3 showed low-to-moderate enrichment, with mainly high CpG density (*Figure 2C*). Regions not impacted by diet showed high H3K4me3 enrichment in CON sperm, with low and high CpG density (*Figure 2C*). Similarly to our previous findings, regions losing H3K4me3 were predominantly located >5 kilobase (kb) from the transcription start site (TSS), likely at transposable elements, putative enhancer regions, and intergenic regions (*Figure 2Di*; *Pepin et al., 2022*). Regions gaining H3K4me3 in HFD-sperm were located near the TSS (within 1 kb), likely at promoter regions (*Figure 2Dii*). Furthermore, obesity-associated deH3K4me3 at promoter regions detected in each study showed enrichment at similar gene ontology (GO) processes, such as metabolic processes, cellular stress responses, transcription, and development (*Figure 2—figure supplement 1E*, *Supplementary files 1b-d and 2*). Examples of genes showing deH3K4me3 in sperm include *Cbx7* (chromobox protein homolog 7, a component of the polycomb repressive complex 1, involved in transcriptional regulation of genes including the *Hox* gene family), *Prdx6* (peroxiredoxin 6, an antioxidant enzyme involved in cell redox regulation by reducing molecules such as hydrogen peroxide and fatty acid hyperoxides), and *Slc19a1* (solute carrier family 19 member 1 or folate transporter 1; a folate organic phosphate antiporter involved in the regulation of intracellular folate concentrations) (*Figure 2E*). We identified deH3K4me3 in HFD-sperm at *Igf2* (*Figure 2Eii* – insulin-like growth factor 2), a paternally expressed imprinted gene with an essential role in promoting cellular growth and proliferation in the placenta. Interestingly, *Igf2* was also previously shown to be differentially expressed in placentas derived from obese males (*Jazwiec et al., 2022*). Importantly, the function of the gene *Igf2* encoding protein IGF-2 has been related to metabolic disease and obesity (*Kadakia and Josefson, 2016*; *Livingstone and Borai, 2014*; reviewed in *St-Pierre et al., 2012*). Other imprinted genes with deH3K4me3 included

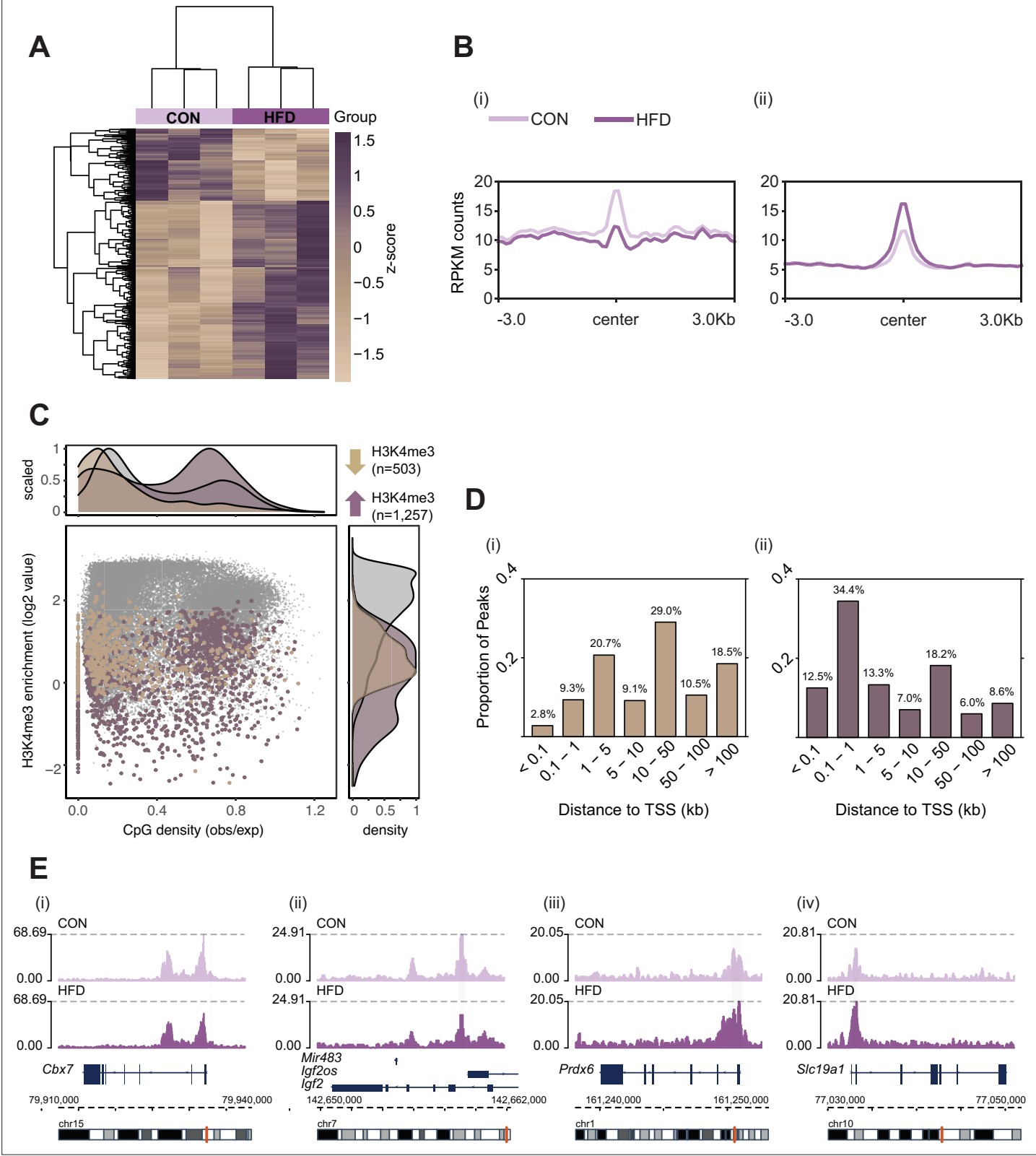

**Figure 2.** Histone H3 lysine 4 tri-methylation (H3K4me3) signal profile at obesity-sensitive regions in sperm. (**A**) Heatmap of log2 normalized counts for obesity-sensitive regions in sperm (n=1760). Columns (samples) and rows (genomic regions) are arranged by hierarchical clustering with complete-linkage clustering based on Euclidean distance. Samples are labeled by dietary group (light and dark purple). (**B**) Profile plots showing RPKM H3K4me3 counts ±3 kilobase around the center of genomic regions with decreased (**i**) and increased (**ii**) H3K4me3 enrichment in high-fat diet (HFD)-sperm

*Figure 2 continued on next page*

Figure 2 continued

compared to control (CON)-sperm. (**C**) Scatter plot showing H3K4me3 enrichment (log2 counts) versus CpG density (observed/expected) for all H3K4me3-enriched regions in sperm (n=35,184, in gray), regions with HFD-induced decreased H3K4me3 enrichment (n=503, in beige), and regions with increased H3K4me3 enrichment (n=1257, in purple). The upper and right panels represent the data points density for CpG density and H3K4me3 enrichment, respectively. (**D**) Bar plots showing the proportion of peaks for each category of distance from the transcription start site (TSS) of the nearest gene in kilobase (kb), for obesity-sensitive regions with decreased (**i**) and increased (**ii**) H3K4me3 enrichment in HFD-sperm. (**E**) Genome browser snapshots showing genes with altered sperm H3K4me3 at promoter regions (CON light purple, HFD dark purple).

The online version of this article includes the following figure supplement(s) for figure 2:

**Figure supplement 1.** Sperm histone H3 lysine 4 tri-methylation (H3K4me3) ChIP-sequencing data quality and normalization.

the homeodomain-containing TF *Otx2* (involved in brain and sense organs development), and the voltage-gated potassium channel *Kcnq1* gene (required for cardiac action potential).

## Differential H3K4me3 enrichment in HFD-sperm occurred at enhancers involved in placenta development, and at TF binding sites

To gain functional insight into how deH3K4me3 in sperm may impact embryonic gene expression, we assessed the association between deH3K4me3 with tissue-specific and embryonic enhancers. Notably, deH3K4me3 localized to putative enhancers including at those implicated in gene regulation of the testes, placenta, and embryonic stem cells (*Figure 3—figure supplement 1A and B*; *Shen et al., 2012*). Interestingly, when searching for closest genes potentially regulated by placenta-specific enhancers, three were paternally expressed imprinted genes (*Tucci et al., 2019*). These genes included those coding for the transmembrane protein TMEM174, the zinc finger protein PLAGL1 (a suppressor of cell growth), and the growth factor PDGFB (a member of the protein family of platelet-derived and vascular endothelial growth factors). This growth factor is essential for embryonic development, angiogenesis, and cell proliferation and migration.

Many TSS in sperm marked by H3K4me3 are enriched for transcription factor (TF) binding sites and bound by components of the transcriptional machinery complex (phosphorylated RNA pol II and Med12). In sperm, such TF interactions have been reported to occur at open chromatin regions and suggested to potentially confer gene expression in the embryo (*Jung et al., 2017*). To explore how altered H3K4me3 in sperm may lead to differential TF interactions, we searched for TF binding motifs enriched in deH3K4me3 regions (Materials and methods). The regions that gained H3K4me3 were significantly enriched for 202 TF binding motifs (p<0.05, binomial statistical test, q-value<0.05; *Figure 3* and *Supplementary file 3*; *Heinz et al., 2010*). In contrast, regions that had reduced H3K4me3 were not significantly enriched for TF binding motifs (q-value>0.05). Of the top 10 motifs enriched at regions with increased H3K4me3 signal in HFD-sperm, these genomic sequences were predicted to be bound by TFs belonging to the ETS, THAP, and ZF motif families (p<1e-10, q-value<0.0001; *Figure 3*). Of interest, SP1, a TF associated with recurrent miscarriage, was found to be among the top TF-associated motif hits (p = 1e-16, q-value<0.0001) in regions gaining H3K4me3 in sperm from obese sires (*Tang et al., 2021*). Activating transcription factor 7 (ATF7; p-value=1e-3, q-value=0.0044, *Supplementary file 3*) binding sites were also enriched and this TF has been associated with oxidative stress-induced epigenetic changes in male germ cells in a mouse model of low-protein diet (*Yoshida et al., 2020*). Overall, these findings suggest a potential route by which diet-induced deH3K4me3 may impact TF interactions.

## Placental gene expression is altered by paternal HFD-induced obesity with sex-specific de-regulation

Next, we aimed to assess whether paternal obesity was associated with changes in gene expression of the placenta. We isolated RNA from E14.5 placentas from CON- or HFD-fed sires and performed RNA-sequencing (RNA-seq), with a high level of correlation between samples (Spearman correlation coefficient>0.89; *Figure 4—figure supplement 1A–C*). There were differentially expressed genes (DEGs) in female (n=2035) and male (n=2365) placentas associated with paternal obesity (*Figure 4A and B*). The placenta is a rich source of hormone production, is highly vascularized, and secretes neurotransmitters (*Hemberger et al., 2020*; *Rosenfeld, 2021*). Disruption in these functions is suggested in the significantly enriched pathways that included genes involved in the transport of cholesterol, angiogenesis,

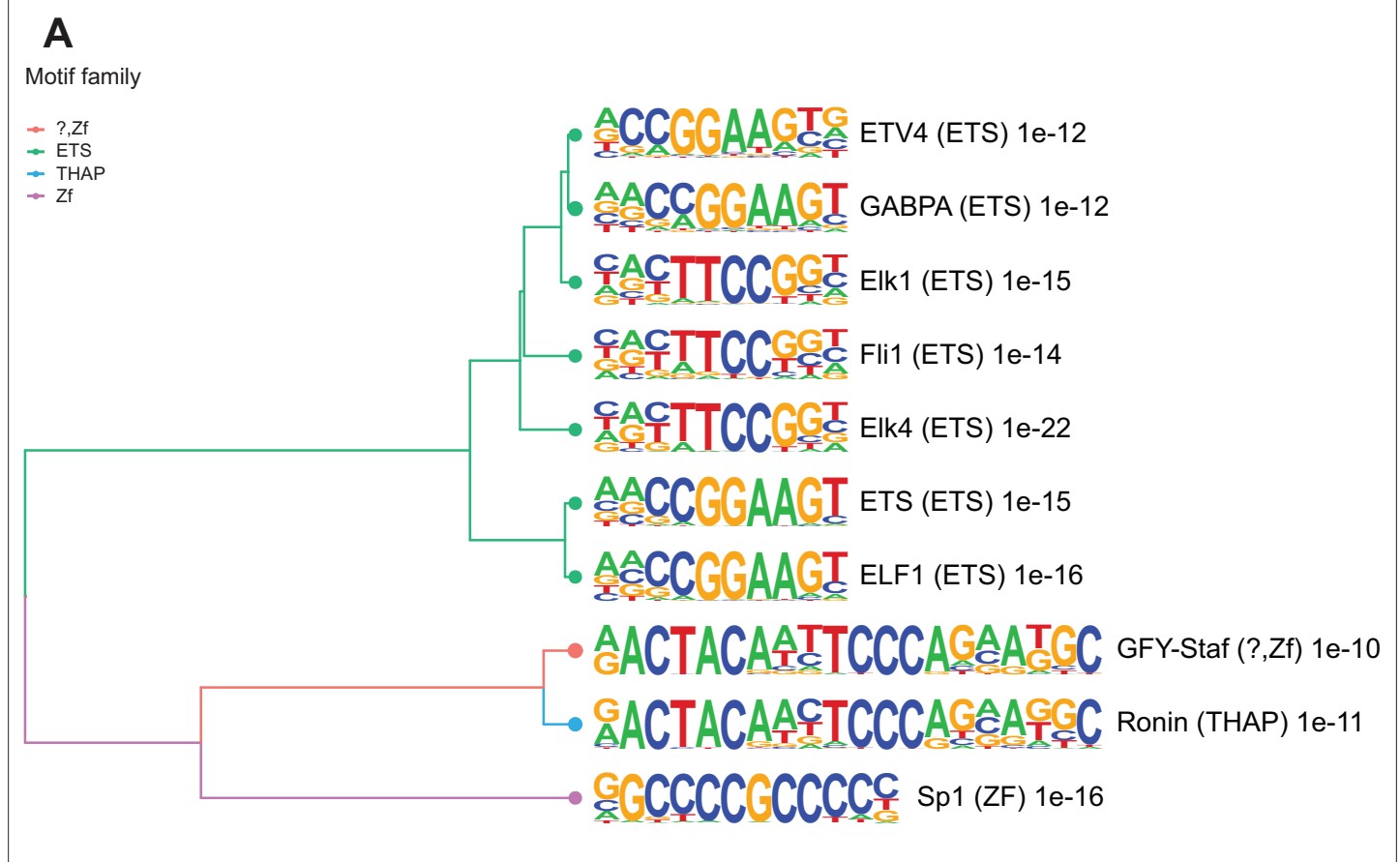

**Figure 3.** Enriched motifs at obesity-sensitive regions in sperm. (**A**) Top 10 significantly enriched known motifs at obesity-sensitive regions with increased histone H3 lysine 4 tri-methylation (H3K4me3) enrichment in high-fat diet (HFD)-sperm. Motifs are clustered based on sequence similarity with hierarchical clustering. Branches of the dendrogram tree are color-coded by motif family. The name of the motif is indicated on the right, with the motif family in parenthesis, and the associated p-value for enrichment significance (binomial statistical test). The full list of enriched motifs can be found in *Supplementary file 3*.

The online version of this article includes the following figure supplement(s) for figure 3:

**Figure supplement 1.** Obesity-sensitive regions in sperm are found at tissue-specific enhancers important for development.

and neurogenesis (*Figure 4C and D*, *Supplementary file 1e and f*). Other significantly enriched processes included genes implicated in nutrient and vitamin transport (*Figure 4C and D*). Of note, the gene *Irs1* was found to be differentially expressed in placentas in association with paternal obesity, validating our previous findings in the initial characterization of the model (*Jazwiec et al., 2022*).

Given that correct imprinted gene expression is critical for placental growth and development, and that aberrant expression of a single imprinted gene is sufficient to induce placental defects, it is worth noting that in HFD-sired female and male placentas, 23 and 28 imprinted genes were differentially expressed, respectively (*Figure 4E and F*; *Tucci et al., 2019*). Although a significant number of DEGs overlapped between female and male placentas (n=359, Fisher's exact test p=1.5e-19; *Figure 4—figure supplement 1Di*), 82% of female DEGs and 85% of male DEGs were uniquely de-regulated in response to paternal obesity. These findings may reflect the previously observed sex-specific effects of paternal factors on offspring metabolism (*Binder et al., 2015a*; *Claycombe-Larson et al., 2020*; *Glavas et al., 2021*; *Jazwiec et al., 2022*; *Pepin et al., 2022*). This suggests some of the sexually dimorphic responses may originate in utero due to differences in placental development and function.

To assess whether there was overlap between obese-sired sperm deH3K4me3 and the placental transcriptome, we compared deH3K4me3 at gene promoters (n=508 and top 10% n=1093 promoters) with DEGs in the placenta. This approach identified 45 and 48 DEGs in female and male placentas, respectively, that overlapped the top 5% deH3K4me3 regions at promoters but was not significant

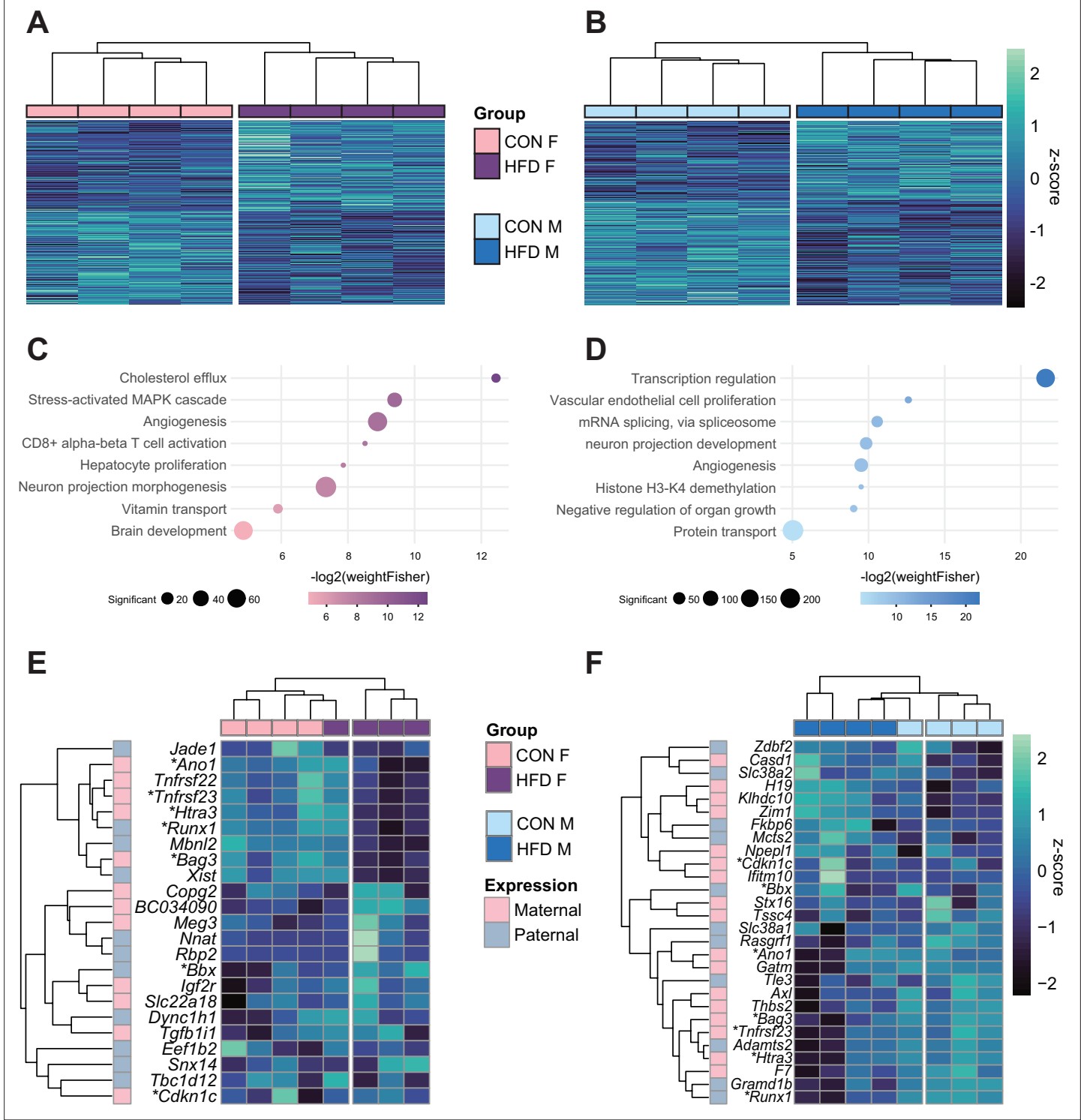

**Figure 4.** Paternal obesity alters the $F_1$ placental transcriptome in a sex-specific manner. (**A–B**) Heatmaps of normalized counts scaled by row (z-score) for transcripts that code for the detected differentially expressed genes (Lancaster p<0.05) in female (A, n=2035 genes) and male (B, n=2365 genes) placentas. Rows are orders by k-means clustering and columns are arranged by hierarchical clustering with complete-linkage based on Euclidean distances. (**C–D**) Gene ontology (GO) analysis for differentially expressed genes in female (**C**) and male (**D**) placentas. The bubble plot highlights eight significantly enriched GO terms, with their -log2(p-value) depicted on the y-axis and with the color gradient. The size of the bubbles represents the number of significant genes annotated to a specific enriched GO term. ***Supplementary file 1e and f*** includes the full lists of significant GO terms. (**E–F**) Heatmaps of normalized counts scaled by row (z-score) for detected differentially expressed imprinted genes (Lancaster p<0.05) in female (E, n=23

*Figure 4 continued on next page*

*Figure 4 continued*

genes) and male (F, n=28 genes) placentas. Genes are labeled based on their allelic expression (paternally expressed genes in pale gray, maternally expressed genes in pale pink). Imprinted genes differently expressed in *both* male and female placentas (n=7) are marked with an asterisk (*). Rows are orders by k-means clustering and columns are arranged by hierarchical clustering with complete-linkage based on Euclidean distances.

The online version of this article includes the following figure supplement(s) for figure 4:

**Figure supplement 1.** Placenta RNA-sequencing data quality assessment.

(*Figure 4—figure supplement 1Di*, Fisher's exact test p=0.62 and p=0.94, respectively). Similarly, repeating this analysis using the top 10% regions at gene promoters impacted by obesity in sperm (n=1093) identified 98 and 110 DEGs in female and male placentas, respectively (*Figure 4—figure supplement 1Dii*, Fisher's exact test p=0.59 and p=0.94, respectively). Next, as we had identified 139 putative enhancers with increased H3K4me3 and 46 with reduced H3K4me3 in HFD-sperm (*Figure 3—figure supplement 1A and B*), we assessed deH3K4me3 in sperm at putative placenta-specific enhancers in relation to placenta DEGs. We focused the analysis on the predicted genes (200 kb range) regulated by these putative enhancers (*Heintzman et al., 2007*; *Shen et al., 2012*), and defined 18 genes that were DEG in female and 19 in male placentas (*Figure 4—figure supplement 1Ei–ii*, Fisher's exact test p=0.32 and p=0.48, respectively). Expanding the analysis to the top 10% deH3K4me3 regions impacted by diet, of the 346 genes predicted to be regulated by placenta-specific enhancer regions showing deH3K4me3 in HFD-sperm, 35 were differentially expressed in female and 35 in male placentas (*Figure 4—figure supplement 1Eiii and iv*, Fisher's exact test p=0.23 and p=0.61, respectively). However, despite the overlap not being statistically significant, this does not rule out a significant biological impact which we elaborate on in the Discussion.

## Deconvolution analysis of bulk RNA-seq reveals paternal obesity alters placental cellular composition

To assess whether there were changes in placental cellular composition associated with paternal obesity, we performed a deconvolution analysis on our bulk RNA-seq data (*Figure 5—figure supplement 1*; *Aliee and Theis, 2021*) using a single-cell RNA-seq reference dataset (*Han et al., 2018*) that matched the developmental stage (E14.5) and mouse strain (C57BL/6J) used here. Of the 28 different cell types identified (*Han et al., 2018*; *Figure 5—figure supplement 1A*, *Supplementary file 1g*), we detected 15 cell types in our deconvolved placenta bulk RNA-seq data (*Figure 5A*, *Figure 5—figure supplement 2A*). The remaining undetected cell types include blood/inflammatory cells and rare or poorly characterized trophoblast subtypes that are found in relatively lower abundances in the reference dataset (*Supplementary file 1g*). The bulk placenta profiles were enriched for three trophoblast, one stroma and one endothelial cell subtypes (*Figure 5A*). Two of the three trophoblast cell types belonged to the SPT lineage including the invasive SPT cells and SPT cells molecularly defined by high expression of 11-β hydroxysteroid dehydrogenase type 2 (*Hsd11b2*). Paternal obesity was associated with changes in both SPT cell populations (*Figure 5A*); we detected a significant decrease in invasive SPT cell relative abundance in female placentas (p=0.02; *Figure 5A*) and increase in high-*Hsd11b2* SPT cells in both male and female placentas (p=0.01 and p=0.06, respectively; *Figure 5A*). These paternal HFD-induced changes in SPT cellular composition indicated by this analysis could reflect impairments in trophoblast specification and thereby contribute to adult-onset metabolic dysfunction in offspring as observed in previous studies (*Jazwiec et al., 2022*; *Pepin et al., 2022*).

To further identify gene expression changes associated with paternal obesity, we repeated the differential gene expression analysis for male and female placentas but adjusted for estimated cell-type proportions (*Figure 5—figure supplement 3A–F*) as recently described (*Campbell et al., 2023*). We first encoded cell-type composition using the top 4 and 3 principal components identified by PCA (*Figure 5—figure supplement 3A, B, D, and E*). As expected, cell types contributing the most to the sample variances for both male and female placentas included the most abundant cell types – namely invasive SPT and spiral artery TGCs, and decidual stromal cells, and endodermal cells (*Figure 5—figure supplement 3C and F*). After adjustment for placental cellular composition, we detected de-regulated genes in female (n=423 DEGs) and male placentas (n=1487 DEGs, *Figure 5B and C*, *Figure 5—figure supplement 3G and H*), respectively. Similarly to our previous characterization of the model, here we detected differential expression of *Igf2* in placentas sired by obese sires (*Jazwiec*

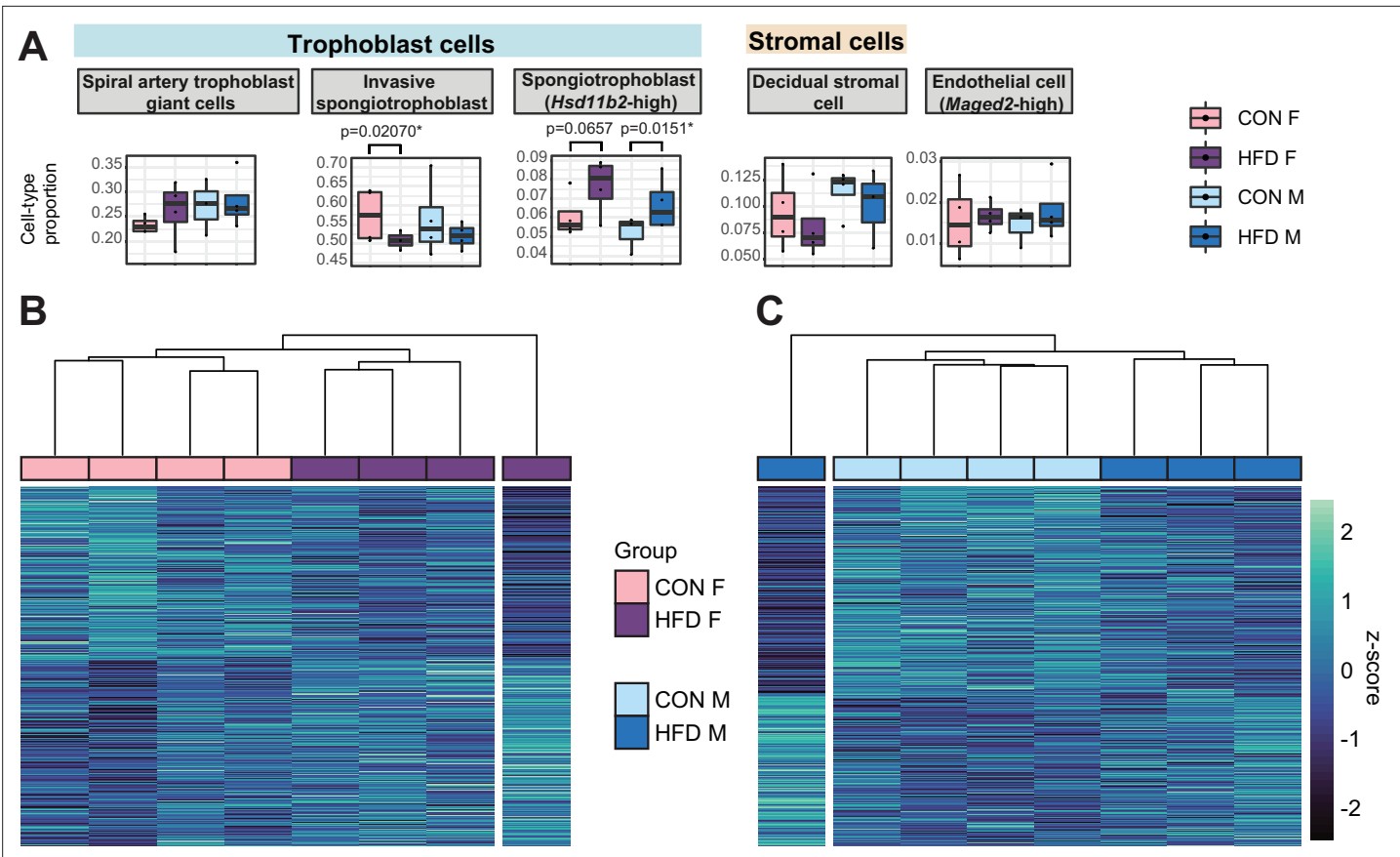

**Figure 5.** Paternal obesity-induced changes in placental cellular composition and differential expression. (**A**) Boxplots showing sample-specific proportions for the top 5 cell types with highest proportions detected in the bulk RNA-sequencing (RNA-seq) data deconvolution analysis across experimental groups (n=4 per sex per group). Beta regression was used to assess differences in cell-type proportions associated with paternal obesity for each placental sex. $p < 0.05$ was considered significant. (**B–C**) Heatmaps of normalized counts scaled by row (z-score) for transcripts that code for the detected differentially expressed genes (Lancaster $p < 0.05$) in female (B, n=423 genes) and male (C, n=1487 genes) placentas, after adjusting for cell-type proportions. Rows are orders by k-means clustering and columns are arranged by hierarchical clustering with complete-linkage based on Euclidean distances.

The online version of this article includes the following figure supplement(s) for figure 5:

**Figure supplement 1.** Cell-type-specific marker genes selection using reference mouse embryonic day (E)14.5 placenta single-cell RNA-sequencing dataset.

**Figure supplement 2.** Estimated cell-type proportions across experimental groups for male and female embryonic day (E)14.5 bulk placenta tissues derived from control (CON)- and high-fat diet (HFD)-fed sires.

**Figure supplement 3.** Principal component analysis (PCA) of estimated cell-type proportions.

*et al., 2022*). The reduction in the number of detected DEGs before versus after accounting for cellular composition suggests that changes in cell-type proportions at least partly drive tissue-level differential expression. This is consistent with the recent finding that pre-eclampsia-associated cellular heterogeneity in human placentas mediates previously detected bulk gene expression differences (*Campbell et al., 2023*). There were similarities between the bulk RNA-seq and deconvoluted analysis in that there was overlap of DEGs detected before and after adjusting for cell-type proportions (*Figure 5—figure supplement 3G and H*, Fisher's exact test p=1.8e-105 and p=0e+00, respectively). This differential gene expression analysis accounting for cellular composition provides insight into how paternal obesity may impact placental development and function and underscores the contribution of cellular heterogeneity in this process.

## Hypoxic and paternal obese-sired placentas show common transcriptomic de-regulation and cell-type composition changes

During placental development, hypoxia is a tightly regulated process that is essential for proper vascular formation supporting fetal growth. Hypoxia is also a hallmark of placental insufficiency, reflecting poor oxygen and nutrient supply to the fetus, resulting in fetal growth restriction, low birth weight, and consequently heightened risk for cardiometabolic disease. Placentas derived from obese sires, like hypoxic placentas, exhibit changes in gene expression and altered angiogenesis, vasculature, and development (*Binder et al., 2015a*; *Binder et al., 2012*; *Jazwiec et al., 2022*; *Lin et al., 2022*; *McPherson et al., 2015*; *Mitchell et al., 2017*, this study). To determine whether transcriptomic and pathological phenotypes in paternal obese-sired placentas relate to that of hypoxic placentas, we compared our HFD placenta RNA-seq data to a hypoxia-induced IUGR mouse model RNA-seq dataset (*Chu et al., 2019*). In this model, hypoxia was induced during late gestation, which resulted in pregnancies with aberrant placental transcriptome, IUGR, decreased birth weights, and offspring exhibiting adult-onset cardiometabolic disturbances (*Chu et al., 2019*). We conducted differential gene expression analysis of the RNA-seq data from the IUGR mouse model using the same approach as used for the obese-sired placenta analysis. Because this dataset did not include a sufficient number of female placenta samples, we focused the analysis on male samples only (n=5 control, n=5 hypoxic placentas). This differential analysis identified 1935 DEGs in hypoxic placentas (*Figure 6—figure supplement 1A–C*). Likewise, we applied our deconvolution analysis described above to this bulk RNA-seq data from hypoxic placentas and detected the same principal cell types as those detected in our samples; a total of 17 different cell types were detected (*Figure 6A*, *Figure 6—figure supplement 1D*). Remarkably, the proportion values for each individual cell types were highly comparable across placentas from the HFD-sire and the hypoxia mouse models (*Figure 6B*). Similarly to placentas derived from obese sires, hypoxic placentas showed a significant decrease in invasive SPT cell abundance (p=0.003, *Figure 6A*). Hypoxic placentas also showed a significant increase in progenitor trophoblast (*Gjb3*-high), primitive endoderm (PE) lineage (*Gkn2*-high), erythroblast (*Hbb-y*-high), and endodermal (*Afp*-high) cells, compared to control (p=0.000004, p=0.01, p=0.000003, p=0.005, respectively; *Figure 6A*). It is intriguing that despite the hypoxia model being a direct exposure to the developing placenta and the obesity model being a paternal pre-conception exposure, there were similar trends for directionality of changes in cell-type abundances (*Figure 6B*).

Next, we sought to investigate how much the observed changes in cellular composition within hypoxic placental tissues might contribute to the differential gene expression observed between conditions. PCA on placental cellular proportion values revealed a separation of samples between the control and hypoxic placentas (*Figure 6—figure supplement 1E*). Similarly to the analysis on HFD placentas, we used the top principal components (n=4 explaining 98.8% of the sample cell proportion variance) to adjust the differential expression analysis for cellular composition (*Figure 6—figure supplement 1F*). Consistent with our data on placentas derived from HFD-fed sires, the cell types contributing the most to sample variance included the invasive SPT cells, endodermal cells (*Afp*-high), and decidual stromal cells. (*Figure 6—figure supplement 1G*). Additionally, erythroblast cells (*Hbb-y*-high) and spiral artery TGCs also strongly contributed to sample variance (*Figure 6—figure supplement 1G*). Accounting for cell-type proportions allowed for the detection of 1477 DEGs associated with hypoxia and growth restriction (*Figure 6C*), of which 356 overlapped with those initially detected before cellular composition adjustment (24%, Fisher's exact test p=8.4e-15; *Figure 6—figure supplement 1H*). These data suggest that like paternal obesity-induced placental de-regulated genes, differential gene expression in hypoxic placentas is partly driven by changes in cellular composition.

Importantly, after adjusting for cell-type proportions, 207 of the paternal-obesity-induced dysregulated genes in male placentas were also found to be differentially expressed in hypoxic placentas (Fisher's exact test p=5.1e-16; *Figure 6D*). A key gene supporting this similarity in the molecular pathology of hypoxic placenta and obese sired placenta was the dysregulation of the imprinted gene *Igf2*. These data are concordant with our previous characterization of paternal obese-sired placentas, whereby placentas showed vascular integrity defects and elevated hypoxia markers that are suggestive of placental insufficiency. Furthermore, both models resulted in metabolic disturbances in offspring, supporting the role of the placenta at the origin of paternally induced metabolic maladaptation. Collectively, our comparative analyses of placental transcriptomic data from both models indicate that paternal obesity, like gestational hypoxia, induces pathological and molecular characteristics that are

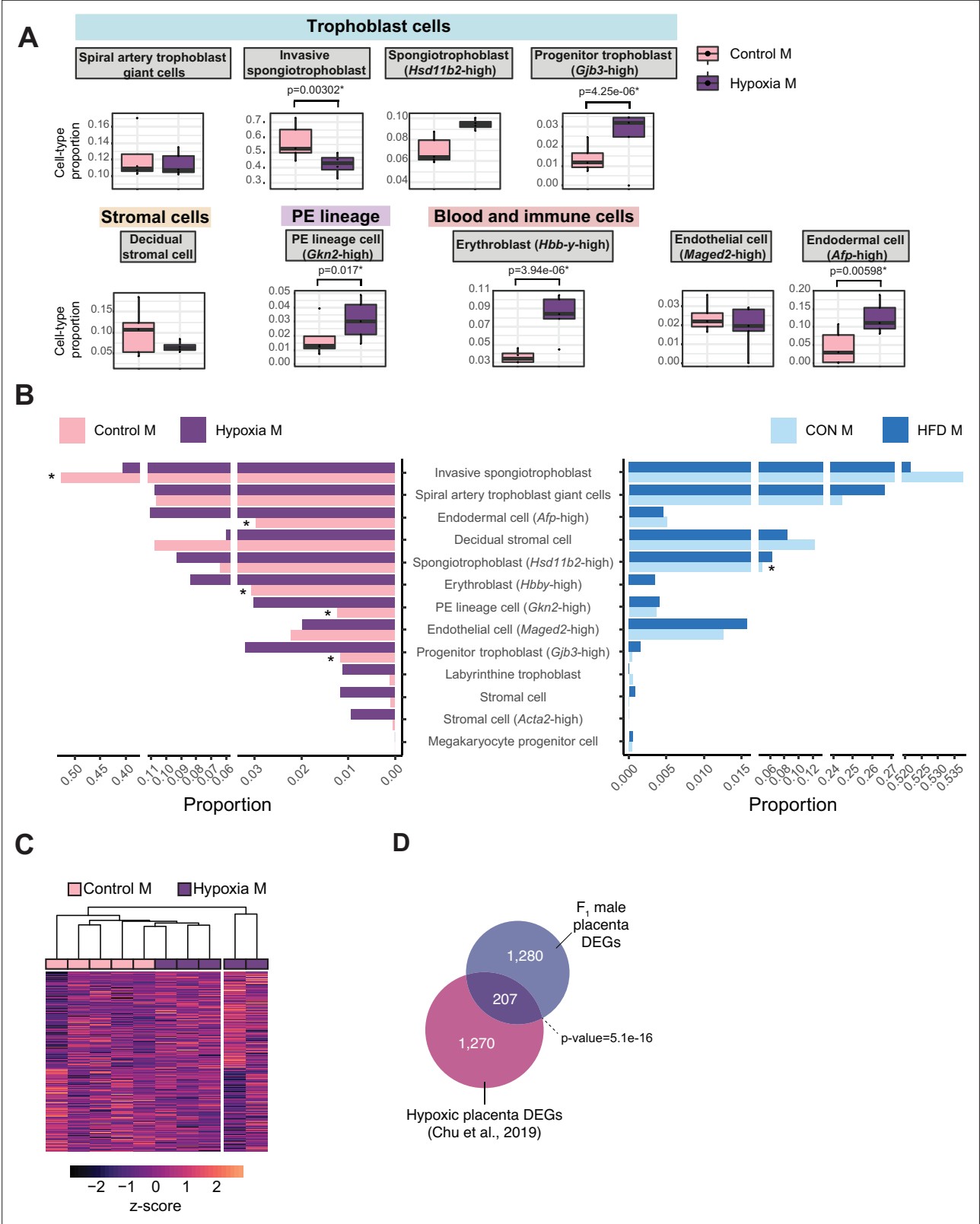

**Figure 6.** Hypoxia-induced growth restriction is associated with changes in placental cellular composition and differential expression. (**A**) Boxplots showing sample-specific proportions for the top 10 cell types with highest proportions detected in the bulk RNA-sequencing data deconvolution analysis across experimental groups (n=5 per group). Beta regression was used to assess differences in cell-type proportions associated with hypoxia-induced intrauterine growth restriction. p<0.05 was considered significant. (**B**) Pyramid plot showing the median values of cell-type proportions

*Figure 6 continued on next page*

*Figure 6 continued*

commonly detected in both datasets assessed. The asterisks (*) denote significance (p<0.05) between control versus hypoxia groups or control (CON) M versus high-fat diet (HFD) groups, as calculated by beta regression. (**C**) Heatmap of normalized counts scaled by row (z-score) for transcripts that code for the detected differentially expressed genes (Lancaster p<0.05, n=1477 genes) in hypoxic placentas, after adjusting for cell-type proportions. Rows are orders by k-means clustering and columns are arranged by hierarchical clustering with complete-linkage based on Euclidean distances. (**D**) Venn diagram showing overlap between hypoxia-induced de-regulated genes in an intrauterine growth restriction model (*Chu et al., 2019*), and paternal obesity-induced de-regulated genes (this study) in male placentas. Fisher's exact test was used to assess significance of overlap across gene sets, using the common universe (background) of the datasets being compared, and p<0.05 was considered significant.

The online version of this article includes the following figure supplement(s) for figure 6:

**Figure supplement 1.** Quality assessment, processing, differential analysis, and deconvolution of RNA-sequencing data from mouse placenta in a hypoxia-induced intrauterine growth restriction mouse model.

hallmarks of placental insufficiency. The placental defects may elicit serious pregnancy complications like pre-eclampsia, and contribute to maladaptive metabolic programming of the fetus, resulting in adult-onset heightened risk to cardiometabolic diseases. Overall, these findings have implications for both maternal and offspring health and underscore the contribution of the placenta in the paternal origins of health and disease.

## Discussion

Paternal health and environmental exposures impact the establishment of the sperm epigenome and are associated with altered development of the placenta, embryo, and offspring health. However, the molecular and cellular mechanisms underlying paternal obesity effects on offspring are still unclear. Our findings build on prior knowledge to show that paternal obesity is associated with altered sperm chromatin, specifically H3K4me3, and changes in the placental transcriptome. We show that paternal obesity influences the placenta transcriptome and this is one mechanism that could underlie paternally induced placental pathogenesis, growth impeded embryo development, and adult-onset metabolic phenotypes.

We further observed that paternal HFD-induced obesity alters the placental transcriptome in a sex-specific manner. There is strong evidence demonstrating sex disparity in metabolic phenotypes and cardiometabolic disease risks (reviewed in *Tramunt et al., 2020*). These sex-specific effects are thought to be driven by sex chromosomes, hormonal factors, the gut microbiome, as well as differential fetal programming across sex in response to pre-conception and in utero exposures (reviewed in *Sandovici et al., 2022*). Here, our findings suggest some of the postnatal metabolic disturbances observed in paternally induced offspring sexually dimorphic phenotypes are established in the placenta. Interestingly, some of the de-regulated genes included imprinted genes. These genes are epigenetically controlled and inherited in a parent-of-origin manner, and the placenta is a key organ for imprinted gene function (*Wood and Oakey, 2006*). According to the conflict hypothesis, maternally imprinted genes (paternally expressed) support fetal growth, whereas paternally imprinted genes (maternally expressed) restrict fetal growth (*Bressan et al., 2009*; *Haig and Graham, 1991*; *Moore and Haig, 1991*). Some of the dysregulated imprinted genes we identified have been implicated in placental defects and pregnancy complications. For example, deletion of the gene *Htra3* (identified here as a DEG in female placentas) in mice has been implicated in IUGR owing to the disorganization of placental labyrinthine capillaries and thereby affecting offspring growth trajectories postnatally (*Li et al., 2017*). The maternally expressed gene *Copg2* (identified here as a DEG in female placentas) has been associated with pregnancies with small for gestational age infants (*Kappil et al., 2015*). Loss of the paternally expressed gene *Snx14* in mice (identified here as a DEG in female placentas) causes severe placental pathology involving aberrant SynT differentiation, leading to mid-gestation embryonic lethality (*Bryant et al., 2020*). The paternally expressed gene *Zdbf2* (DEG in male placentas) has been implicated in reduced fetal growth in mice, associated with altered appetite signals in the hypothalamic circuit (*Glaser et al., 2022*). Placental deficiency of the paternally expressed gene *Slc38a2* (identified here as a DEG in male placentas) leads to fetal growth restriction in mice (*Vaughan et al., 2021*). Lastly, mice deficient for the paternally expressed transcriptional co-repressor *Tle3* (identified here as a DEG in male placentas) show abnormal placental development including TGC differentiation failure, resulting in fetal death (*Gasperowicz et al., 2013*). Importantly,

disrupting the expression of a single imprinted gene can result in placental defect and consequently compromise fetal health or survival. It is therefore likely that the differential expression of imprinted genes detected in female and male placentas as a result of paternal obesity could at least partly explain metabolic phenotypes observed in this mouse model (*Jazwiec et al., 2022*; *Pepin et al., 2022*).

We identified changes in sperm H3K4me3 associated with paternal obesity, some of which were enriched for transcription factor binding sites. Interestingly, changes in sperm DNA methylation upon HFD feeding has been previously reported, and ETS motifs (enriched in obesity-associated deH3K4me3 in sperm) have been found to be DNA methylation sensitive, including in spermatogonial stem cells (*Domcke et al., 2015*; *Dura et al., 2022*; *Lea et al., 2018*; *Yin et al., 2017*). Given the interplay between H3K4 and DNA methylation, it is conceivable that HFD-induced epimutations at either of these marks could influence one another (reviewed in *Janssen and Lorincz, 2022*), and these could in turn alter TF functions. This phenomenon has been described in a mouse model of paternal low-protein diet, where oxidative stress-induced phosphorylation of the ATF7 TF was suggested to impede its DNA-binding affinity in germ cells, leading to a decrease in H3K9me2 at target regions (*Yoshida et al., 2020*). As in the low-protein diet model, oxidative stress is a hallmark of obesity and increased levels of reactive oxygen species have been observed in testes of diet-induced obesity mouse models and linked to impaired embryonic development (*Fullston et al., 2012*; *Lane et al., 2014*; *Mitchell et al., 2011*). These findings provide avenues for further investigation such as whether epigenetic changes on paternal alleles may impact TF binding during early embryogenesis.

We previously reported in the same model from which our placenta samples are derived that paternal obesity leads to functional and histopathological abnormalities in placenta and dysfunctional offspring metabolism. The epigenomic link between sperm enrichment differences in H3K4me3 and DEGs in placenta are minor. While not statistically significant we would argue that this does not rule out the likely biological significance of deH3K4me3 in sperm. For example, in a recent epigenome editing model from *Takahashi et al., 2023*, using embryonic stem cells, CpG islands in promoters for either the *Ankrd26* or *Ldlr* genes were targeted to alter DNA methylation then injected into 8-cell embryos. These embryos gave rise to mice with metabolic phenotypes. This is a key example showing that a single epigenetic change at one gene can give rise to a phenotype (*McNamara et al., 2018*; *Nativio et al., 2011*; *Takahashi et al., 2023*). In addition, the limited overlap between sperm deH3K4me3 and placenta DEGs may also reflect the terminally differentiated state and heterogenous nature of the placenta at E14.5. Perhaps a greater correspondence between sperm deH3K4me3 may have been observed at earlier developmental time points such as at first lineage segregation corresponding to trophoblast formation. In future studies, it will be worthwhile to examine the trophoblast gene expression in comparison to deH3K4me3 in sperm. Indeed, we previously showed by in silico analysis that most regions bearing H3K4me3 in sperm are enriched for this mark in TE, correlate with TE H3K4me3 signal, and correspond to genes expressed in TE (*Pepin et al., 2022*). Another limitation of this study is that placenta profiles are from bulk RNA-seq and measures average gene expression across a heterogenous cell population and identification of DEGs can therefore be confounded by cell composition. To address this, we used a deconvolution approach in our analysis using a single-cell RNA-seq dataset.

The identification of alterations of cell-type proportion must be considered within the limitations of a deconvolution analysis. This analysis only provides estimates of cell type relative within a heterogeneous tissue. While this allowed us to adjust for the effect of differences in cell-type composition, the exact cell-type composition and their specific gene expression changes need to be validated by single-cell approaches such as single-cell RNA-seq or spatial transcriptomics. Furthermore, even though we used a reference dataset which included cells representative of placental tissues, the detection capacity of this approach is limited for low-abundant cell types, such as blood cells, immune cells, and inflammatory cells, which would be informative of placental pathological states. For example, aberrant abundance of decidual inflammatory cells, such as natural killer cells, has been linked to the pathogenesis of pre-eclampsia (*Aneman et al., 2020*; *Bachmayer et al., 2006*; *Du et al., 2022*; *Milosevic-Stevanovic et al., 2016*; *Williams et al., 2009*). Incidentally, it was previously shown that paternal diet-induced obesity is associated with placental inflammation (*Claycombe-Larson et al., 2020*; *Jazwiec et al., 2022*). Interestingly, many GO terms related to inflammatory processes were enriched in the obesity-induced deH3K4me3 in sperm (*Figure 2E*, *Supplementary file 1c*, and *Pepin*

*et al., 2022*), suggesting sperm deH3K4me3 might be partly influencing placental inflammation. However, due to the low representation of immune cells in the dataset this could not be assessed.

## Speculation and perspectives

Many of the DEGs in the paternal obese-sired placentas were involved in the regulation of the heart and brain. This is in line with paternal obesity associated with the developmental origins of neurological, cardiovascular, and metabolic disease in offspring (*Andescavage and Limperopoulos, 2021*; *Binder et al., 2015a*; *Binder et al., 2012*; *Chambers et al., 2016*; *Cropley et al., 2016*; *de Castro Barbosa et al., 2016*; *Fullston et al., 2012*; *Fullston et al., 2013*; *Grandjean et al., 2015*; *Huypens et al., 2016*; *Jazwiec et al., 2022*; *Mitchell et al., 2011*; *Ng et al., 2010*; *Pepin et al., 2022*; *Perez-Garcia et al., 2018*; *Terashima et al., 2015*; *Thornburg et al., 2016*; *Thornburg and Marshall, 2015*; *Ueda et al., 2022*; *Wei et al., 2014*). The brain-placenta and heart-placenta axes refer to their developmental linkage to the trophoblast which produces various hormones, neurotransmitters, and growth factors that are central to brain and heart development (*Parrettini et al., 2020*; *Rosenfeld, 2021*). This is further illustrated in studies where placental pathology is linked to cardiovascular and heart abnormalities (*Andescavage and Limperopoulos, 2021*; *Thornburg et al., 2016*; *Thornburg and Marshall, 2015*). For example, in a study of the relationship between placental pathology and neurodevelopment of infants, possible hypoxic conditions were a significant predictor of lower Mullen Scales of Early Learning (*Ueda et al., 2022*). A connecting factor between the neural and cardiovascular phenotypes is the neural crest cells which make a critical contribution to the developing heart and brain (*Hemberger et al., 2020*; *Perez-Garcia et al., 2018*). Notably, neural crest cells are of ectodermal origin which arises from the TE (*Prasad et al., 2019*), which is in turn governed by paternally driven gene expression. It is worth considering the routes by which TE dysfunction may be implicated in the paternal origins of metabolic and cardiovascular disease. First, altered placenta gene expression beginning in the TE could influence the specification of neural crest cells which are a developmental adjacent cell lineage in the early embryo. TE signaling to neural crest cells could alter their downstream function. Second, altered trophoblast endocrine function will influence cardiac and neurodevelopment (*Hemberger et al., 2020*).

In line with these possible routes to developmental origins of obesity and metabolic disease, paternal obesity was associated with altered trophoblast lineage specification. During placentation, invasive SPT have the ability to migrate and invade the maternal-fetal interface and replace maternal vascular endothelial cells, a critical step for maternal arterial remodeling to facilitate low resistance and high volume blood flow to the fetus (*Silva and Serakides, 2016*). Consequently, improper trophoblastic invasion has been linked to various obstetrical complications, including premature birth, fetal growth restriction, pre-eclampsia, and placenta creta (*Barrientos et al., 2017*; *Duzyj et al., 2018*; *O'Tierney-Ginn and Lash, 2014*). Paternal obesity also induced changes in trophoblasts expressing the glucocorticoid metabolizing enzyme HSD11B2. In the placenta, HSD11B2 is responsible for the conversion of cortisol into its inactive form, cortisone, which limits fetal exposure to maternal glucocorticoid levels. Interestingly, de-regulation of the gene *Hsd11b2* has been observed in rodent fetal growth restriction models (*Chu et al., 2019*; *Cuffe et al., 2014*). These aberrant cellular composition profiles suggest that paternal factors, such as diet, can induce functional changes in the placenta that mirror placental defects associated with adult-onset cardiometabolic phenotypes.

Next, it will be important to assess earlier developmental time points to determine when and how these paternally induced effects originate. Indeed, studies have shown that paternal obesity alters preimplantation development, such as cellular allocation to TE versus ICM lineages (*Binder et al., 2012*). Investigating multiple and earlier time points would help reveal the dynamic trajectory of paternally induced de-regulated transcriptomic and epigenetic signatures which might be at the origin of adult-onset disease. Translating these findings to humans will be beneficial to further understand and emphasize the paternal preconception contribution to pregnancy outcomes, placental integrity, and offspring health.

# Materials and methods

**Key resources table**

| Reagent type (species) or resource | Designation | Source or reference | Identifiers | Additional information |
|---|---|---|---|---|
| Biological sample (*Mus musculus*, male) | Spermatozoa | C57BL/6J, The Jackson Laboratory | | Isolated from *Mus musculus* (C57BL/6J) |
| Biological sample (*Mus musculus*) | Placenta | C57BL/6J, The Jackson Laboratory | | Isolated from *Mus musculus* (C57BL/6J) |
| Antibody | Tri-Methyl-Histone H3 (Lys4) (C42D8) (Rabbit monoclonal) | Cell Signaling Technology | Cat#:9751 | (5 µg) |
| Commercial assay or kit | ChIP DNA Clean and Concentrator | Zymo Research | Cat#:D5201 | |
| Commercial assay or kit | RNeasy Mini Kit | QIAGEN | Cat#:74104 | |
| Chemical compound, drug | Dithiothreitol | Bio Shop | Cat#:3483-12-3 | |
| Chemical compound, drug | Micrococcal nuclease (MNase) | Roche | Cat#:10107921001 | |
| Chemical compound, drug | Complete Tablets EASYpack | Roche | Cat#:04693116001 | |
| Chemical compound, drug | DynaBeads, Protein A | Thermo Fisher Scientific | Cat#:10002D | |
| Chemical compound, drug | Bovine Serum Albumin (BSA) | Sigma-Aldrich | Cat#:BP1600-100 | |
| Chemical compound, drug | RNase A | Sigma-Aldrich | Cat#:10109169001 | |
| Chemical compound, drug | Proteinase K | Sigma-Aldrich | Cat#:P2308 | |
| Software, algorithm | R (version 4.0.2) | *R Core Team, 2018* | | |
| Software, algorithm | Python (version 3.7.4) | *Van Rossum and Drake, 2009* | | |
| Software, algorithm | Trimmomatic (version 0.36) | *Bolger et al., 2014* | | |
| Software, algorithm | Bowtie2 (version 2.3.4) | *Langmead and Salzberg, 2012* | | |
| Software, algorithm | SAMtools (version 1.9) | *Li et al., 2009* | | |
| Software, algorithm | Deeptools (version 3.2.1) | *Ramírez et al., 2016* | | |
| Software, algorithm | Trim Galore (version 0.5.0) | *Krueger, 2015* | | |
| Software, algorithm | Hisat2 (version 2.1.0) | *Kim et al., 2015* | | |
| Software, algorithm | Stringtie (version 2.1.2) | *Pertea et al., 2015* | | |
| Software, algorithm | Seaborn (version 0.9.0) | *Waskom, 2021* | | |
| Software, algorithm | Betareg (version 3.1–4) | *Ferrari and Cribari-Neto, 2004* | | |
| Software, algorithm | Csaw (version 1.22.1) | *Lun and Smyth, 2016* | | |
| Software, algorithm | Sva (version 3.36.0) | *Leek et al., 2012; Zhang et al., 2020* | | |
| Software, algorithm | topGO (version 2.40.0) | *Alexa et al., 2006* | | |
| Software, algorithm | trackplot | *Bolger et al., 2014* | | |
| Software, algorithm | Rtracklayer (version 1.48.0) | *Lawrence et al., 2009* | | |
| Software, algorithm | HOMER (version 4.10.4) | *Heinz et al., 2010* | | |
| Software, algorithm | ViSEAGO (version 1.2.0) | *Brionne et al., 2019* | | |
| Software, algorithm | DESeq2 (version 1.28.1) | *Love et al., 2014* | | |
| Software, algorithm | Aggregation (version 1.0.1) | *Yi et al., 2018* | | |
| Software, algorithm | Corrplot (version 0.88) | *Taiyun and Simko, 2021* | | |
| Software, algorithm | Pheatmap (version 1.0.12) | *Kolde, 2019* | | |
| Software, algorithm | Numpy (version 1.17.2) | *Harris et al., 2020* | | |
| Software, algorithm | Pandas (version 0.25.2) | *McKinney, 2010* | | |

*Continued on next page*

Continued

| Reagent type (species) or resource | Designation | Source or reference | Identifiers | Additional information |
|---|---|---|---|---|
| Software, algorithm | Pickle (version 4.0) | *Van Rossum, 2020* | | |
| Software, algorithm | Scanpy (version 1.8.2) | *Wolf et al., 2018* | | |
| Software, algorithm | Scipy (version 1.7.3) | *Virtanen et al., 2020* | | |
| Software, algorithm | Autogenes (version 1.0.4) | *Aliee and Theis, 2021* | | |

## Resource availability

### Lead contact
Further information and requests for resources and reagents should be directed to and will be fulfilled by the Lead Contacts, D Sloboda (sloboda@mcmaster.ca) and S Kimmins (sarah.kimmins@mcgill.ca).

### Materials availability
This study did not generate new unique reagents.

## Experimental model and subject details

### Animals' husbandry and dietary treatment
Animal experiments were conducted at the McMaster University Central Animal Facility, approved by the Animal Research Ethics Board, and in accordance with the Canadian Council on Animal Care guidelines, under the Animal Utilization Protocol #16-09-35. Six-week-old C57BL/6J male mice were co-housed (2 per cage) and randomly allocated to either the control (n=8; CON; standard chow diet, Harlan 8640, Teklad 22/5 Rodent Diet; 17% kcal fat, 54% kcal carbohydrates, 29% kcal protein, 3 kcal/g) or HFD (n=16; HFD; Research Diets Inc, D12492; 20% kcal protein, 20% kcal carbohydrates, 60% kcal fat, 5.21 kcal/g) group, for 10–12 weeks. All animals had free access to water and food ad libitum, housed in the same room which was maintained at 25°C on a controlled 12 hr/12 hr light/dark cycle. Two weeks prior mating and throughout the mating period, male mice were housed individually to prevent fighting and aggression, and to facilitate timed mating. We have previously reported and characterized this model of HFD-induced obesity (*Jazwiec et al., 2022*). As reported in our previous work, male mice fed a HFD became significantly heavier weighing approximately 40 g (20 g in weight gain); had elevated body adiposity; elevated fasting blood glucose; were glucose intolerant; had significantly increased serum insulin concentrations, and were insulin resistant as assessed by HOMA-IR (*Jazwiec et al., 2022*). Food consumption data was not collected. High-fat male mice had impaired mating efficiency (*Jazwiec et al., 2022*), and therefore more males were needed in the high fat-fed group compared to controls to generate a sufficient number of pregnancies and placental tissue for RNA-seq experiments.

After the diet intervention, male mice were housed with one or two virgin C57BL/6J females overnight. To confirm mating, females were examined the following morning, and the presence of a copulatory plug was referred to as E0.5. Females confirmed as pregnant were individually housed throughout gestation and fed a standard chow diet (Harlan 8640, Teklad 22/5 Rodent Diet). Pregnant females (n=4 CON; n=5 HFD) were sacrificed at E14.5 by cervical dislocation to collect placenta samples for RNA-seq. One male and one female placenta samples per dam were collected. Placenta were cut in half, with one half snap-frozen in liquid nitrogen and kept at –80°C until RNA extraction. CON- and HFD-fed male mice were sacrificed at 4–5 months of age via cervical dislocation, and sperm was collected.

## Methods details

### Sperm isolation
Sperm was collected at necropsy from paired caudal epididymides as previously described (*Hisano et al., 2013*; *Lismer et al., 2021*; *Pepin et al., 2022*). Caudal epididymides were cut in 5 mL of Donners medium (25 mM NaHCO₃, 20 mg/mL Bovine Serum Albumin [BSA], 1 mM sodium pyruvate, 0.53% vol/vol sodium DL-lactate in Donners stock), and spermatozoa were allowed to swim out by agitating the solution for 1 hr at 37°C. Sperm cells were collected by passing the solution through a

40 µm strainer (Fisher Scientific, #22363547) followed by three washes with phosphate-buffered saline (PBS). The sperm pellet was cryopreserved at –80°C in freezing medium (Irvine Scientific, cat#:90128) until used for the chromatin immunoprecipitation.

## Chromatin immunoprecipitation, library preparation, and sequencing

Chromatin immunoprecipitation experiment was performed as previously described (*Hisano et al., 2013*; *Lismer et al., 2021*; *Pepin et al., 2022*). In brief, samples were thawed on ice and washed with PBS. Spermatozoa were counted under a microscope using a hemocytometer and 12 million cells were used per experiment. Pools of sperm from 2 to 7 male mice per sample were used for ChIP-seq (*Supplementary file 1a*). Each pool was equalized in cell number and comprised of sperm from n=8 control males, or n=16 HFD males. This pooling was required to create replicates with an equal amount of total sperm. We used 1 M dithiothreitol (DTT, Bio Shop, cat#:3483-12-3) to decondense the chromatin and *N*-ethylmaleimide was used to quench the reaction. Cell lysis was performed with a lysis buffer (0.3 M sucrose, 60 mM KCl, 15 mM Tris-HCl pH 7.5, 0.5 mM DTT, 5 mM MgCl$_2$, 0.1 mM EGTA, 1% deoxycholate, and 0.5% NP40). DNA digestion was performed in aliquots containing 2 million spermatozoa (6 aliquots per sample), with micrococcal nuclease (MNase, 15 units per tube; Roche, #10107921001) in an MNase buffer (0.3 M sucrose, 85 mM Tris-HCl pH 7.5, 3 mM MgCl$_2$, and 2 mM CaCl$_2$) for 5 min at 37°C. The reaction was stopped with 5 mM EDTA. Supernatants of the 6 aliquots were pooled back together for each sample after a 10 min centrifugation at maximum speed. A 1× solution of protease inhibitor (complete Tablets EASYpack, Roche, #04693116001) was added to each tube. Magnetic beads (DynaBeads, Protein A, Thermo Fisher Scientific, #10002D) used in subsequent steps were pre-blocked in 0.5% BSA (Sigma-Aldrich, #BP1600-100) solution for 4 hr at 4°C. Pre-clearing of the chromatin was done with the pre-blocked beads for 1 hr at 4°C. Magnetic beads were allowed to bind with 5 µg of antibody (Histone H3 Lysine 4 tri-methylation; H3K4me3; Cell Signaling Technology, cat#:9751) by incubating for 8 hr at 4°C. The pre-cleared chromatin was pulled down with the beads-antibody suspension overnight at 4°C. Beads-chromatin complexes were subjected to three rounds of washes; one wash with a low-salt buffer (50 mM Tris-HCl pH 7.5, 10 mM EDTA, 75 mM NaCl) and two washes with a high-salt buffer (50 mM Tris-HCl pH 7.5, 10 mM EDTA, 125 mM NaCl). Elution of the chromatin was done in two steps with 250 µL (2×125 µL) of elution buffer (0.1 M HaHCO$_3$, 0.2% SDS, 5 mM DTT) by shaking the solution at 400 rpm for 10 min at 65°C, vortexing vigorously, and transferring the eluate in a clean tube. The eluate was subjected to an RNase A (5 µL, Sigma-Aldrich, #10109169001) treatment shaking at 400 rpm for 1 hr at 37°C, followed by an overnight Proteinase K (5 µL, Sigma-Aldrich, #P2308) treatment at 55°C. The *ChIP DNA Clean and Concentrator* (Zymo Research, #D5201) kit was used following the manufacturer's protocol to purify the eluted DNA with 25 µL of the provided elution buffer. Libraries were prepared and sequenced at the McGill University and *Génome Québec* Innovative Centre, with single-end 100 base-pair reads on the Illumina HiSeq 2500 sequencing platform (n=3 pooled samples per diet group, *Supplementary file 1a*).

## RNA extraction, library preparation, and sequencing

Extraction of RNA from placentas was performed using the RNeasy Mini Kit (QIAGEN, cat#:74104) following the manufacturer's protocol. In brief, 10–20 mg of frozen placenta were cut on dry ice. Samples were lysed in a denaturing buffer and homogenized with homogenizer pestles. Lysates were centrifuged, supernatants transferred into a clean tube, and 70% ethanol was added to lysates. An additional DNase digestion step was performed to avoid DNA contamination. Spin columns were washed twice, and total RNA was eluted with 30 µL of RNase-free water. Libraries were prepared and sequenced at the McGill Genome Centre with paired-end 100 base-pair reads on the illumina NovaSeq 6000 sequencing platform (n=4 per sex per diet group).

## Pre-processing
### Sperm ChIP-seq data

Pre-processing of the data was performed as previously described (*Pepin et al., 2022*). Sequencing reads were trimmed using the *Trimmomatic* package (version 0.36) on single-end mode filtering out adapters and low-quality reads (parameters: ILLUMINACLIP:2:30:15 LEADING:30 TRAILING:30)

(*Bolger et al., 2014*). Reads were aligned to the mouse genome assembly (*Mus musculus*, mm10) with *Bowtie2* (version 2.3.4) (*Langmead and Salzberg, 2012*). *SAMtools* (version 1.9) was used to filter out unmapped reads and *Perlcode* to remove reads with more than three mismatches (*Li et al., 2009*). BAM coverage files (BigWig) files were created with *deeptools2* (version 3.2.1) (parameters: -of bigwig -bs 25 -p 20 `--normalizeUsing` RPKM -e 160 `--ignoreForNormalization` chrX) (*Ramírez et al., 2016*).

## Placenta RNA-seq data

Sequencing data was pre-processed as previously described (*Pepin et al., 2022*). Sequencing reads were trimmed with *Trim Galore* (version 0.5.0) in paired-end mode to remove adapters and low-quality reads (parameters: `--paired --retain_unpaired --phred33 --length` 70 -q 5 `--strin-gency` 1 -e 0.1) (*Krueger, 2015*). Reads were aligned to the mouse reference primary assembly (GRCm38) with *hisat2* (version 2.1.0, parameters -p 8 --dta) (*Kim et al., 2015*). The generated SAM files were converted into BAM format and sorted by genomic position with *SAMtools* (version 1.9) (*Li et al., 2009*). *Stringtie* (version 2.1.2) was used to build transcripts and calculate their abundances (parameters: -p 8 -e -B -A) (*Pertea et al., 2015*).

## Publicly available datasets

Raw files for bulk RNA-seq in control and hypoxic placentas (n=7 and 8, respectively) were downloaded from the National Centre for Biotechnology Information (NCBI) with the Sequencing Read Archive (SRA) Toolkit (NCBI SRA: SRP137723) (*Chu et al., 2019*). Files were pre-processed as described above for RNA-seq on single-end mode.

Processed files with raw counts for single-cell RNA-seq data from E14.5 mouse placenta were downloaded from NCBI (GEO: GSE108097) and metadata matrix and cluster annotations were downloaded from https://figshare.com/s/865e694ad06d5857db4b (*Han et al., 2018*).

# Quantification and statistical analysis

## Visualization, statistical, and bioinformatic analyses

Bioinformatic data analyses were conducted using R (version 4.0.2) (*R Core Team, 2018*) and Python (version 3.7.4) (*Van Rossum and Drake, 2009*). Figures were generated using the R package ggplot2 (version 3.3.3) (*Wickham, 2016*) and the Python package *seaborn* (version 0.9.0) (*Waskom, 2021*). Statistical analysis was conducted using R version 4.0.2 (*R Core Team, 2018*). For all statistical tests, a p-value less than 0.05 was considered significant. To assess significance of overlap between different sets of genes, a Fisher's exact test was performed using the *fisher.test* function from the *stats* package (version 4.0.2), and the numbers that were used to assess statistical significance were those found in the common universe (background) of both lists being compared. To assess differences in cell-type proportions across experimental groups, a beta regression was performed using *betareg* function from the *betareg* package (version 3.1–4) (*Ferrari and Cribari-Neto, 2004*).

## Sperm ChIP-seq data

ChIP-sequencing data was processed and analyzed as previously described (*Pepin et al., 2022*). Using *csaw* (version 1.22.1), sequencing reads were counted into 150 base-pair windows along the genome, and those with a fold-change enrichment of 4 over the number of reads in 2000 base-pair bins were considered as genomic regions enriched with H3K4me3 in sperm (*Lun and Smyth, 2016*). Enriched windows less than 100 base-pair apart were merged allowing a maximum width of 5000 base-pair (n=35,186 merged enriched regions in total). Reads were counted in those defined regions, and those with a mean count below 10 across samples were filtered out (conferring a total of n=35,184 regions). Read counts within enriched regions were normalized with TMM and corrected for batch effects arising from experimental day, using the *sva* package (version 3.36.0) (*Leek et al., 2012*; *Zhang et al., 2020*). Spearman correlation heatmaps were generated using *corrplot* (version 0.88) and mean average plots with *graphics* packages (*Taiyun and Simko, 2021*).

To detect the obesity-sensitive regions, PCA was performed. We selected the top 5% regions contributing the separation of samples according to diet group along PC1, conferring a total of 1760 regions associated with dietary treatment. Those regions were split according to directionality change based on positive and negative $\log_2$ fold-change values (increased versus decreased enrichment in

HFD group, respectively) from the median normalized counts of each group. The selected obesity-sensitive regions were visualized with *Pheatmap* (version 1.0.12) (*Kolde, 2019*). Profile plots were generated using *deeptools* (*Ramírez et al., 2016*). The distance from the nearest TSS from each selected region was calculated and visualized with *chipenrich* (version 2.12.0) (*Welch et al., 2014*). The genes for which their promoters overlapped the detected obesity-sensitive regions were used in the GO analysis using *topGO* (version 2.40.0) with Biological Process ontology category and Fisher's exact test (*weight01Fisher* algorithm) to test enrichment significance (*Alexa et al., 2006*). A *weight01Fisher* p-value below 0.05 was considered significant. Genome browser snapshots of examples of detected obesity-sensitive regions were generated using *trackplot* (*Pohl and Beato, 2014*). Annotations for tissue-specific enhancers were downloaded from ENCODE (*Shen et al., 2012*) (GEO: GSE29184) and genome coordinates were converted from the mm9 to the mm10 mouse assembly using the *liftOver* function from the *rtracklayer* package (version 1.48.0) (*Lawrence et al., 2009*). To determine the corresponding genes that could be regulated by tissue-specific enhancers, we scanned the landscape surrounding putative enhancer genomic coordinates, and selected the nearest gene located less than 200 kb away, given that enhancers interact with promoters located within the same domain (*Heintzman et al., 2007*; *Shen et al., 2012*). To retrieve the gene annotations, we used the function *annotateTranscripts* with the annotation database *TxDb.Mmusculus.UCSC.mm10.knownGene* (version 3.10.0) and the annotation package *org.Mm.eg.db* (version 3.11.4) from the *bumphunter* package (version 1.30.0) (*Aryee et al., 2014*; *Jaffe et al., 2012*). From the same package, the function *matchGenes* was used to annotate the putative tissue-specific enhancer genomic coordinates with the closest genes. Annotations for transposable elements and repeats were obtained from *annotatr* (version 1.14.0) (*Cavalcante and Sartor, 2017*) and RepeatMasker (https://www.repeatmasker.org/). Upset plots were generated using the UpSetR package (version 1.4.0, *Conway et al., 2017*). The motif analysis was performed using HOMER (version 4.10.4, *Heinz et al., 2010*), with the binomial statistical test and standard parameters. *ViSEAGO* (version 1.2.0, *Brionne et al., 2019*) was used for visualization, semantic similarity, and enrichment analysis of GO (*Figure 2—figure supplement 1E*). Gene symbols and annotations were obtained from the *org.Mm.eg.db* database for the *Mus musculus* species. The Biological Process ontology category was used, and statistical significance was assessed with a Fisher's exact test with the classic algorithm. A p-value less than 0.01 was considered significant. Enriched terms are clustered by hierarchical clustering based on Wang's semantic similarity distance and the *ward.D2* aggregation criterion.

## Placenta RNA-seq data

Placenta bulk RNA-seq data from this study and from *Chu et al., 2019*, was processed and analyzed using the same approach, as previously described (*Pepin et al., 2022*). In brief, transcripts with low read counts were filtered out (mean count<10), for a total of 47,268 and 49,999 transcripts detected in male and female placentas, respectively, and 32,392 transcripts in placentas from *Chu et al., 2019*. Differential analysis was conducted with *DESeq2* (version 1.28.1) (*Love et al., 2014*). For the data generated in this study, we included the batch information (RNA extraction day) and dietary group in the design formula and performed a stratified analysis by running male and female samples separately (*Figure 4—figure supplement 1B and C*). For the data generated in *Chu et al., 2019*, only male samples were analyzed given there was not a sufficient number of female samples, and we included the experimental group in the formula. Independent hypothesis weighting (IHW, version 1.16) (*Ignatiadis et al., 2016*) was used for multiple testing correction and prioritization of hypothesis testing. We performed a gene-level analysis at single-transcript resolution using the Lancaster method (*aggregation* package, version 1.0.1, *Yi et al., 2018*). This method aggregates p-values from individual transcript to detect DEGs based on changes at the transcript level. A p-value less than 0.05 was considered significant.

For visualization, variance stabilized transcript counts were used without blind dispersion estimation (*Love et al., 2014*). Spearman correlation heatmaps were plotted with *corrplot* (version 0.88) (*Taiyun and Simko, 2021*) with samples clustered by hierarchical clustering. Transcripts coding for detected DEGs were visualized with *Pheatmap* (version 1.0.12) (*Kolde, 2019*), with samples clustered with hierarchical clustering and transcripts by k-means clustering (n kmeans = 2). GO analysis was performed as described above for the sperm ChIP-seq data. For the genomic imprinting analysis, the list of known mouse imprinted genes was retrieved from *Tucci et al., 2019*.

## Deconvolution analysis

We used single-cell RNA-seq datasets from mouse E14.5 placenta to deconvolute our bulk RNA-seq data (*Han et al., 2018*). The following Python packages were used: *seaborn* (version 0.9.0) (*Waskom, 2021*), *numpy* (version 1.17.2) (*Harris et al., 2020*), *pandas* (version 0.25.2) (*McKinney, 2010*), *pickle* (version 4.0) (*Van Rossum, 2020*), *scanpy* (version 1.8.2) (*Wolf et al., 2018*), *scipy* (version 1.7.3) (*Virtanen et al., 2020*), and *autogenes* (version 1.0.4) (*Aliee and Theis, 2021*). The *pyplot* module was loaded from the *matplotlib* library (version 3.4.2) (*Hunter, 2007*). The deconvolution analysis was performed following the AutoGeneS package's available code (version 1.0.4) (*Aliee and Theis, 2021*). In brief, single-cell counts were log normalized and the 4000 most highly variable genes were selected. A principal component analysis was performed (*Figure 5—figure supplement 1A*) and the cell types previously annotated in *Han et al., 2018*, were visualized (*Figure 5—figure supplement 1B*). The means of each centroids for each cell type cluster was used for optimization and feature selection. AutoGeneS uses a multi-objective optimization approach to select marker genes. In this process, a search algorithm explores a set of optimal solutions (commonly called Pareto-optimal solutions) and evaluates the objective functions (in this case, correlation and distance between the cell-type-specific clusters; *Figure 5—figure supplement 1C and D*). This optimization technique allows to select the 400 marker genes (*Figure 5—figure supplement 1E*). Lastly, the Nu-support vector machine regression model (*Pedregosa et al., 2011*) was used to estimate the cell-type proportions for the bulk RNA-seq data from this study and from *Chu et al., 2019*. The estimated cell-type proportions were visualized as boxplots for each cell type. The cell types with percent abundance values of zero across all samples were excluded. Statistical significance across experimental groups was assessed with beta regression on the cell types that had a median relative abundance of at least 1.5%.

## Placenta RNA-seq differential analysis with cell-type proportion adjustment

To adjust for cell-type proportions in the differential analysis, while reducing the number of covariates in the model, and to account for dependence between the cell-type proportions, a PCA was performed with the deconvoluted cell-type proportions using the *prcomp* function from R's base statistics. The top 3 or 4 principal components were selected to capture most of the sample variance (*Figure 5—figure supplement 3A and D*, *Figure 6—figure supplement 1E*). The differential analysis described above was repeated, with the selected principal components added as covariates in the design formula to form the cell-type adjusted model.

## Acknowledgements

We thank the team from Genome Quebec for the sequencing of the ChIP-seq experiment, and the team from the Applied Genomics Innovation Core of the McGill Genome Centre for the sequencing of the RNA-seq experiment.

## Additional information

### Funding

| Funder | Grant reference number | Author |
| --- | --- | --- |
| Canadian Institutes of Health Research | 358654 | Sarah Kimmins |
| Canadian Institutes of Health Research | 350129 | Sarah Kimmins |
| Canadian Institutes of Health Research | 146333 | Deborah M Sloboda |
| Canadian Institutes of Health Research | 175293 | Deborah M Sloboda |

The funders had no role in study design, data collection and interpretation, or the decision to submit the work for publication.

## Author contributions
Anne-Sophie Pepin, Resources, Data curation, Software, Formal analysis, Investigation, Visualization, Methodology, Writing – original draft, Writing – review and editing; Patrycja A Jazwiec, Resources, Writing – review and editing; Vanessa Dumeaux, Conceptualization, Resources, Software, Supervision, Funding acquisition, Writing – review and editing, Methodology; Deborah M Sloboda, Sarah Kimmins, Conceptualization, Resources, Supervision, Funding acquisition, Writing – original draft, Writing – review and editing

## Author ORCIDs
Anne-Sophie Pepin ![ORCID] https://orcid.org/0000-0002-1894-9391
Vanessa Dumeaux ![ORCID] https://orcid.org/0000-0002-1280-6541
Sarah Kimmins ![ORCID] https://orcid.org/0000-0002-0168-7233

## Ethics
Animal experiments were conducted at the McMaster University Central Animal Facility, approved by the Animal Research Ethics Board, and in accordance with the Canadian Council on Animal Care guidelines, under the Animal Utilization Protocol #16-09-35.

## Decision letter and Author response
Decision letter https://doi.org/10.7554/eLife.83288.sa1
Author response https://doi.org/10.7554/eLife.83288.sa2

# Additional files

## Supplementary files
Supplementary file 1. Data quality statistics and gene ontology analysis. (a) ChIP-sequencing sample information and read statistics. (b) Significant gene ontology terms enriched in high-fat diet (HFD)-sperm differentially expressed histone H3 lysine 4 tri-methylation (deH3K4me3) regions at promoters detected in our previous study and this study, related to *Figure 2—figure supplement 1E*. (c) Significant gene ontology terms enriched in HFD-sperm at regions showing a decrease in H3K4me3 at promoters, related to *Figure 2—figure supplement 1E*. (d) Significant gene ontology terms enriched in HFD-sperm at regions showing an increase in H3K4me3 at promoters, related to *Figure 2—figure supplement 1E*. (e) Significant gene ontology terms enriched in differentially expressed genes in female placentas derived from HFD-sires, related to *Figure 4C*. (f) Significant gene ontology terms enriched in differentially expressed genes in male placentas derived from HFD-sires, related to *Figure 4D*. (g) Reference single-cell RNA-sequencing data information (from *Han et al., 2018*) – number of cells per cell type, related to *Figure 5—figure supplement 1*.

Supplementary file 2. Interactive heatmap for significant gene ontology terms enriched in high-fat diet (HFD)-sperm deH3K4me3 regions at promoters detected in our previous study (*Pepin et al., 2022*) and this study, related to *Figure 2—figure supplement 1E*.

Supplementary file 3. Motif analysis, showing significantly enriched known motifs in regions gaining H3K4me3 in high-fat diet (HFD)-sperm, related to *Figure 3*.

MDAR checklist

## Data availability
Sequencing data have been deposited in GEO under the SuperSeries accession code GSE207326.

The following dataset was generated:

| Author(s) | Year | Dataset title | Dataset URL | Database and Identifier |
|---|---|---|---|---|
| Pepin AS, Kimmins S | 2022 | Paternal obesity alters the sperm epigenome and is associated with changes in the placental transcriptome and its cellular composition | https://www.ncbi.nlm.nih.gov/geo/query/acc.cgi?acc=GSE207326 | NCBI Gene Expression Omnibus, GSE207326 |

The following previously published datasets were used:

| Author(s) | Year | Dataset title | Dataset URL | Database and Identifier |
|---|---|---|---|---|
| Chu A, Casero D, Thamotharan S, Cosi A, Wadehra M, Devaskar SU | 2019 | The placental transcriptome in late gestational hypoxia resulting in murine intrauterine growth restriction predicts adult cardiometabolic disease | https://www.ncbi.nlm.nih.gov/geo/query/acc.cgi?acc=GSE112755 | NCBI Gene Expression Omnibus, GSE112755 |
| Han X, Wang R | 2018 | Mapping Mouse Cell Atlas by Microwell-seq | https://www.ncbi.nlm.nih.gov/geo/query/acc.cgi?acc=GSE108097 | NCBI Gene Expression Omnibus, GSE108097 |

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
