## [Editor Report]

In this interesting report, the authors probe the correlative effect of paternal obesity on placental development. In the substantively revised manuscript, the authors provide solid epigenomic data arising from dietary differences in fat intake in the male, subsequent chromatin changes in sperm, and link these outcomes to transcriptomic changes in offspring placenta. These useful insights will be of interest to those studying the epigenetic transmission of germline changes in the next generation, and those studying the relationship to metabolic exposure in the parental generation to offspring.

---

## [Decision Letter]

**Decision letter after peer review:**

Thank you for submitting your article "Paternal obesity alters the sperm epigenome and is associated with changes in the placental transcriptome and cellular composition" for consideration by *eLife*. Your article has been reviewed by 4 peer reviewers, one of whom is a member of our Board of Reviewing Editors, and the evaluation has been overseen by Marianne Bronner as the Senior Editor. The reviewers have opted to remain anonymous.

This important study presents data suggesting that HFD-induced epimutations in sperm may impact the transcriptome of the placenta, thereby contributing to the paternal transmission of metabolic disorders to offspring. The strength of this work includes the interesting idea and the initial data generated. However, the entire study remains purely correlative without any validation experiment to support the correlation. The conclusion needs to be further supported by a bigger sample size and more functional analyses demonstrating the causal relationship among histone epimutations detected, dysregulated mRNA expression in the placenta, and phenotypes in offspring.

Essential revisions (for the authors):

1) Proper statistical analyses for correlating the H3K4me3 changes with DEGs of RNA-seq data, for evaluating effect size vs. intra-sample variations, and for analyzing other sequencing data.

2) More functional validation to support that the altered placental gene expression causes phenotypes in both placental functions and offspring development and health.

*Reviewer #1 (Recommendations for the authors):*

This study reports H3K4me3 ChIP-seq data on sperm from HDF male mice and the placental tissue of their offspring. By comparing the altered H3K4me3 marks in sperm with the same mark and gene expression levels in the placenta, the authors identified numerous histone epimutation-dysregulated mRNA pairs of a potential causal relationship. The finding is interesting, but the conclusion is premature due to a lack of functionally supportive data. Therefore, the authors need to either tone down their claim or provide more functional data to support their notion. In addition, this reviewer has the following specific comments:

1) The ideal control for HFD is the ingredient control diet manufactured to specifically match the composition of HFD except for in fat and sugar contents. The so-called control chow may contain many components that do not match the HFD, thus contributing to the epimutations detected.

2) The sample size appears to be very small, and intra-samples/mice variations in the histone marks might be greater than the effect size. Unless proper statistical analyses are employed to prove otherwise, the differences detected may not represent true signals.

3) The notion that sperm tend to regulate placental gene expression needs to be supported experimentally.

4) It is essential to phenotype the offspring to determine whether the changes in placental transcriptome indeed have an impact on offspring metabolic states.

*Reviewer #2 (Recommendations for the authors):*

General comments:

The text is well written and the figures are generally well presented and clear. The general flow of the manuscript could be improved, as it is not always entirely clear to the reader what the authors were thinking when carrying out a set of experiments. More specific suggestions and comments are provided below.

1. Specifically, regarding the general flow, the part about the transcription factor binding motifs is interesting but the authors should specify what they hypothesize may be happening: Would transcription in the sperm be altered prior to conception? Do they think this is carried over into the zygote? Where would the transcription factors be coming from?

2. In the part where sequencing data from Chu et al. (2019) is used, more information is needed about this study to put the presented findings into context for the reader: What kind of model for hypoxia is used in this publication? Why was this chosen and not any other paternal intervention – the justification l. 304 "Placentas derived from obese sires, like hypoxic placentas, exhibit changes in gene expression and altered angiogenesis, vasculature, and development" is very unspecific and likely applies to a number of different interventions therefore the choice of this dataset and model over others should be justified well.

3. The manuscript would greatly benefit from an additional experiment probing a causal relationship between the examined mechanisms. At the moment it is full of correlations, some of which are overstated, and the limitations of the methods are not discussed enough.

4. Regarding the Methods, the control diet is not the best control for the HFD, because it is from a different supplier. The fact that it is from a different manufacturer means that the content, not just fat levels, is likely completely different. This should at least be mentioned by the authors, within the limitations of this methodological issues addressed in the text. However, ideally the key experiments should ideally be performed again, using controlled diets (high fat and lower/control fat levels) from the same manufacturer, so that any differences can be related to fat (rather than potentially other molecular constituents that differ between manufacturers).

5. Were the males group or single housed during the diet intervention? How much weight did the HFD males put on compared to controls i.e. how severe was the obesity? How much food did they consume? Was their reproductive function impacted? And is this why n value for HFD was double that of controls? Some information about this aspect (e.g. in further supplementary figures) would be informative.

6. Several descriptions of procedures appear vague and need to be clarified. Why were the procedures not always the same? In which cases did they vary? The authors should be clear why there were various different protocols in place.

– E.g.: high-fat diet was given to males for "8-10 weeks"

– E.g.: Males were mated with "1-2 females"

– E.g.: Pools of sperm for ChIP-seq consisted of "2-7 males"

This detail specifically, in combination with the knowledge that F0 groups were n=8 for the controls and n=16 for the HFD, gives the impression that the samples were likely not equal for both experimental groups in that the pools for HFD would have been systematically larger, leading to deeper sequencing and other differences? This may constitute a methodological flaw as it may introduce a bias in the sequencing data.

7. With n=3 in ChIP-seq and n=4 per group in RNA Seq, the sample sizes are on the lower end (bare minimum) of what is considered valid. These analyses are likely underpowered and at least require independent validation, especially if the role of specific genes is discussed as mechanistically relevant.

8. In the same vein, in the GO analysis in Figure 2E(i), many of the significantly enriched pathways are only represented with a single gene, including the placenta morphogenesis pathway. The authors seem to overstate this result on l. 234 by using the plural of genes: "As deH3K4me3 in sperm was located at genes involved in placental formation (Figure 2 E and Pepin et al., 2022), we assessed whether paternal obesity was associated with changes in gene expression of the placenta." If this result is assigned such importance to, it would be important to validate the result.

9. In the Results, the formulation of some results gives the wrong impression/ overstates the actual result. The authors should be generally more careful in their statements and qualify them.

10. For example, on l.166: "Despite differences in experimental design and animal models, we found a significant overlap in regions showing differential (H3K4me3 deH3K4me3) from both studies (128 overlapping regions, Fisher's exact test P=2.2e-16, Figure S1 D)". These are 128 regions out of 1632 and 1410 regions, so less than 10% overlap. Therefore, in fact this is a surprisingly low overlap despite methodological similarities, and should be discussed as such, not the opposite.

11. Another example is l. 322: "Overall, the trends for directionality of changes in specific cellular abundances were consistent across the two mouse models (Figure 6B)." While the effects on cell type abundance cannot be said to substantially diverge, saying they are consistent may be overstating things. From the data, it appears that hypoxia has a much stronger effect. This should be discussed in terms of differences between the models etc, not glossed over as "consistent".

12. In Figure S3D (ii)/(iii): Do those genes which are differentially expressed in males and females from HFD fathers and also differentially methylated in sperm (45 in F, 48 in M) overlap at all?

13. The authors state they perform the cell-type based deconvolution of sequencing data because transcriptional differences between cell types can be obscured by averaging of expression data over many cell types in Bulk tissue RNA-Seq. This is true, however the re-analysis actually leads to a much lower number of DEGs detected (as shown in FiguresS6G,H). This should be noted and discussed by the authors.

14. In the same deconvolution analysis, the authors state they detect 15 cell types, out of 28 detected by the original study from which the dataset was taken. Why do the authors think this is the case? This means almost half the cell types are not included in the tissue sample? Or are they too low in abundance to detect? This should be mentioned.

15. The Discussion seems a little superficial. It should delve more into the implications and limitations of the study.

16. The discussion of specific genes is inappropriate in the absence of validation experiments.

17. In Figure S1 C: This PCA analysis looks strange: the points for both groups are perfectly symmetrical. Is this an artefact? I cannot make sense of a scenario where this would be the result.

18. Figure S1E: It is unclear from the figure and figure legend what the numbers 1-15 indicate.

19. The representations of cell type abundance in the placenta are too small, in combination with very thick lines and bold fonts, they are difficult to read. (e.g. Figure 5A, 6A, S5, S7D).

*Reviewer #3 (Recommendations for the authors):*

1) The study would be significantly strengthened by a more nuanced analysis of the data, as well as more specific follow up and functional validation of individual genes that may be responsible for placental phenotypes in offspring. For example, in situ hybridization or Western blots in placenta to confirm the expression differences detected by RNA-seq would better support the claims.

2) Validation of deconvolution analysis, for example by sorting or immunofluorescence to experimentally validate the cell type composition of the placentas, would increase confidence in the deconvolution results.

3) Statistical tests (e.g. Fisher's Exact test) should be done to show the significance of overlaps when comparing gene sets.

4) Because the differences in H3K4me3 signal between conditions appear to be very modest (Figure 2B, 2F), it is not clear whether these differences are real or just due to variations between libraries. Spike-ins in the ChIP libraries would drastically improve confidence in these differences.

5) In Figure S3C (PCA for placenta RNA-seq data), there does not seem to be good separation between placental gene expression in offspring of HFD compared to control sires. This should be explained.

6) In Figure 2D: regions losing H3K4me3 are predominantly >5kb from the TSS. Can an explanation of what these sites are likely to correspond to be provided? Most sites of strong H3K4me3 are at a TSS.

*Reviewer #4 (Recommendations for the authors):*

The authors build upon their previous work on mouse models testing the contribution of parental diet to the health and development of the offspring. Previously, the Sloboda lab had reported that HFD induced placental hypoxia while the Kimmins lab had identified genomic regions that show differentially enrichment of H3K4me3 upon HFD. With this work the authors aim to explore "the role of paternally driven gene expression in placenta" as well as the potential role of H3K4me3 in transmitting information from sperm to embryo.

Firstly, the authors assessed H3K4me3 changes in sperm samples of the HFD model from the Kimmins lab. They identified DE regions of H3K4me3 between HFD and control sperm and annotated these regions based on their genomic location (promoter/enhancer). Next, the authors associated these genomic locations to genes for which they performed ontology analysis. In addition, they performed Motif enrichment analysis

Secondly, the authors performed bulk RNAseq in placentas of embryos sired either from HFD or control males mated with C57BL6/J females. They performed differential gene expression analysis and tried to associate the placenta DE genes to sperm DE H3K4me3 regions. They further performed deconvolution of the bulk RNA seq data to identify cellularity biases of the placentae and finally they compared their data to external RNA seq data from hypoxic placentae.

The bioinformatic analysis is rigorous, however the design of the experiments cannot support the initial aims of the authors. For example, to identify paternally driven gene expression the authors should have used female animals on a different genetic background (other than C57BL6/J) to be able to differentiate between the parental origins of transcripts based on SNPs. The current experiment shows transcriptional differences that cannot be attributed to the parental origin. We cannot exclude the possibility of paternal expression affecting maternal expression in trans. In addition, as the authors acknowledge in the manuscript that the association of DEGs in placenta to sperm differentially H3K4 methylated regions is difficult given the high cellular heterogeneity within the tissue. Hence, the authors are recommended to isolate cells that they identified to be more affected. In addition, or alternatively, the authors could investigate trophoblast cells from blastocyst embryos that give rise to placenta. In summary, the overall design of the study is not optimal, which together with rather confusing writing does not help the field to understand the impact of paternal HFD diet on chromatin landscape in sperm and physiology in offspring.

1. One major objective of the study is to relate changes in H3K4me3 occupancy levels in sperm to transcriptional changes in placenta. From the current data analyses, it is not clear at which genomic regions H3K4me3 levels have changed in a statistically significant manner between dietary conditions using the data of the replicates. When describing changes, please include fold change and adjusted p-values. Given that the authors used the csaw package to determine and normalize the H3K4me3 regions why they didn't perform the statistical significance analysis of differential enrichment using the csaw embedded functions?

2. Related to the first comment, this reviewer does not observe any difference in H3K4me3 levels for Cbx7 nor for Igf2, while Prdx6 and Slc19a1 seems to have increased H3K4me3 levels (Figure 2F). What is the significance value for changes for these 4 genomic windows? It should be indicated in the figure. For Igf2, it is not clear whether the peaks localize at its promoter or somewhere in the gene. It appears more likely to be the latter. If so, what is the significance for such occupancy for gene regulation potential? Moreover, given the prominent role that Igf2 plays within the manuscript, have the authors observed any change in H3K4me3 at the germ line ICR for this locus, which has been shown to be critical for paternal transmission of the paternal epiallele and imprinted expression in offspring? Any change in DNA methylation? Likewise, for Kcnq1 which in placental tissues is normally maternally expressed and paternally repressed by Polycomb repression system. What about H3K4me3 levels at the Kcnq1ot1 ICR element?

3. For a histone modification to fulfill a possible role in paternal inheritance, what are the overall nucleosomal occupancy levels at the deH3K4me3 regions in sperm, compared to genome wide average levels in control and HFD samples?

4. Figure S1D/E: What is the directionality of H3K4me3 changes (increased, decreased) between the two studies?

5. The authors report that "Regions losing H3K4me3 showed moderate H3K4me3-enrichment in 173 CON sperm, with predominantly low CpG density, whereas regions gaining H3K4me3 showed low-to moderate enrichment with mainly high CpG density (Figure 2 C)". What could the reason be for the sequence specificity of such changes? Are there any changes in H3K27me3 levels at deH3K4me3? It is quite noticeable that none of the sites gaining H3K4me3 in HFD samples reach the level of H3K4me3 observed at CpG-rich regions in sperm of controls (grey dots in figure 2C). Why? Would it have any impact on possible transmission rates?

6. To be able to interpret / "normalize" the data presented in panel 2D, please provide the distribution of windows relative to TSS according to their presence in genome.

7. Figure 2E: It seems as if the GO-term for regions with decreased H3K4me3 levels serve more likely functions in placenta than those with increased levels.

8. Regarding enhancers, the current data presentation is rather superficial. To substantiate some of these findings, to what extent do the changes in H3K4me3 as monitored in sperm reflect changes in expression in testicular cells of HFD exposed males? Are any of the three genes highlighted (Tmem174, Plag1, Pdgfb) differentially expressed in placentas of male and female embryos?

9. Discovery of TF motif: when searching for TF motif enrichments in genomic windows, have the authors taken into account that the GC percentage underlying these windows is very different between regions with decreased and increased H3K4me3 levels? This has a major impact on the obtained results.

10. The authors state that previous reports indicated changes in DNA methylation after HFD exposure (line 217). Is there any link between DNA methylation and H3K4me3 alterations?

11. "In response to paternal obesity, we detected 2,035 and 2,365 differentially expressed genes (DEGs) in female and male placentas, respectively (Figure 4 A-B)." How many DEGs are detected if authors compare the male control to female control placentas. This will help to understand which is ground truth of sex specific differences of placentas. How many are UP versus DOWN regulated in the respective sexes? How have the authors harvested the placentas? Which parts of placenta and decidua was included in the RNA-sample prep.?

12. When reviewing the heatmaps of imprinted genes in females, 1 out of 4 HFD placenta clusters with controls; for males, only 2 HFD placentas show altered gene expression. How do these changes relate to the histological changes? Why is there variation at all? How relevant would the DEGs be for phenotypic changes? Could imprinted expression serve as a means to classify the severity of the perturbation in embryos/placenta? Histological data would be needed for benchmarking.

13. Regarding imprinted genes, which ones are commonly mis-expressed in male and female placenta (e.g. Runx1, Ano1) and which are sex specific? Why? How have the imprinted genes been chosen? Please indicate and discuss which are controlled by germline derived DNA methylation and which by maternal H3K27me3.

14. "To assess the link between sperm H3K4me3 and the placental transcriptome, we overlapped deH3K4me3 at promoters (n=508) with DEGs in the placenta, and identified 45 and 48 DEGs in female and male placentas, respectively (Figure S3 D ii-iii)." Please specify how many UP-regulated and DOWN-regulated. The authors do not dwell further on this finding. What was the fold change in expression?

15. "Of note, although a significant number of DEGs overlapped between female and male placentas (n=359, Fisher's exact test P=1.5e-19; Figure S3 D i), 82% of female DEGs and 85% of male DEGs were uniquely de-regulated, indicating sex-specific placental responses to paternal obesity."

Similarly, the overlap between the H3K4me3 peaks when comparing the 2 HFD models is less than 10%. The authors present this overlap as great similarity without acknowledging protocol specific responses. "Despite differences in experimental design and animal models, we found a significant overlap in regions showing differential H3K4me3 (deH3K4me3) from both studies (128 overlapping regions, Fisher's exact test P=2.2e-16, Figure S1 D)." Please discuss in a more balanced manner these results.

16. The authors should perform a histological analysis of the placenta or use the previously published histological analysis to support/or reject the deconvolution method of the bulk RNAseq data.

17. Next, it is not clear to this reviewer how the authors can assign nor normalize expression of genes that do not belong to specific cell types to be differentially expressed between experimental conditions. The meaning of any of these DEGs results can not be evaluated without any detailed follow-up analysis.

[Editors' note: further revisions were suggested prior to acceptance, as described below.]

Thank you for resubmitting your work entitled "Determining the effects of paternal obesity on sperm chromatin at histone H3 lysine 4 tri-methylation in relation to the placental transcriptome and cellular composition" for further consideration by *eLife*. Your revised article has been evaluated by Yamini Dalal (Senior Editor) and a Reviewing Editor.

The manuscript has been improved, for which we thank you and acknowledge your efforts.

However, there are some remaining issues that need to be addressed: the first reviewer suggests pointing out limitations of the study given the difference in the control food vs. the HFD food. The second reviewer strongly advocates for additional controls that would support the central claims. If you can provide these, we feel it would significantly strengthen the final paper.

*Reviewer #2 (Recommendations for the authors):*

The manuscript has been extensively revised to address the comments of myself and the other reviewers, and is substantially improved. The remaining major weakness, commented upon by myself and at least on other reviewer, was the fact that the control chow was not matched to the HFD chow, which impacts on interpretation of the study. This major limitation of the study could be discussed in more detail, as it is possible that the results could be at least partly due to non-fat differences between the control and HFD chow, and not entirely due to increased fat consumption (although the authors state that food intake was not measured, which is another weakness in the study that should be discussed) and/or the consequent state of obesity.

*Reviewer #3 (Recommendations for the authors):*

The authors have updated and improved this manuscript and have provided important clarifications about some aspects. Most importantly, statistical analysis is now provided for overlaps between ChIP-seq and transcriptomic datasets. Additionally, a better discussion of the biological relevance of differentially enriched and differentially expressed genes is provided, as well as a better explanation of the deconvolution analysis.

A concern that has not been addressed is the lack of validation studies. The authors argue that validation is not required because the datasets are robust and comparisons are statistically well supported. However, these arguments do not address the main reason for validation, which is to test the conclusion using a different assay in a different sample. Validation experiments would be especially valuable since in many cases the effect sizes are small, albeit statistically significant. To this end, RT-qPCR or Western blotting could be used to validate differential expression in placentas, and ChIP-qPCR could be used to validate differential H3K4me3 enrichment in sperm.

A new concern is that after significance testing, there is no significant overlap between genes with differential H3K4me3 enrichment in sperm and differential expression in the placenta. However, because of the variable nature and small effect size of paternal epigenetic effects, this finding is not unexpected and the datasets and analysis are still interesting in demonstrating intergenerational regulatory effects of paternal diet.

---

## [Author Response]

Essential revisions (for the authors):1) Proper statistical analyses for correlating the H3K4me3 changes with DEGs of RNA-seq data, for evaluating effect size vs. intra-sample variations, and for analyzing other sequencing data.

Author response 1. The following statistical additions have been made:

Fisher’s exact test to overlap paternal obesity-associated differentially enriched region of H3K4me3 (deH3K4me3) compared to paternal obesity-associated deH3K4me3 identified in our other obesity model (Pepin, Lafleur, Lambrot, Dumeaux, and Kimmins, 2022) (PMID: 35183795)(Figure 2 —figure supplement 1 Di and ii).Fisher’s exact test to overlap paternal obesity-associated differentially enriched regions of H3K4me3 (deH3K4me3) with female and male placenta differentially enriched genes (Figure 4 —figure supplement 1 D i and ii).Fisher’s exact test to overlap paternal obesity-associated differentially expressed genes in male placentas with hypoxia-related differentially expressed genes in placentas (Chu et al., 2019) (PMID: 30718791) (Figure 6 D; P=5.1e-16)Fisher’s exact test to overlap the top 5% or top 10% regions contributing to PC1 and associated with sample separation according to dietary group, from this study and our previous study (Pepin et al., 2022) (PMID: 35183795) (Figure 2 —figure supplement 1 D i and ii, P=2.2e-16 and P=2.2e-16, respectively).Fisher’s exact test to overlap paternal obesity-induced de-regulated genes between female and male placentas with the top 5% and the top 10% promoter regions most sensitive to obesity in sperm (Figure 4 —figure supplement 1 D i and ii, see figure for P values)Fisher’s exact test to overlap paternal obesity-induced de-regulated genes in female and male placentas, with the nearest gene to placental-specific enhancer overlapping sperm deH3K4me3, using the top 5% or top 10% sperm regions associated with dietary group (Figure 4 —figure supplement 1 E i-iv; see figure for P values)Fisher’s exact test to overlap the differentially expressed genes in female and male placentas, before and after adjusting for cell-type proportions (Figure 5 —figure supplement 3 G and H, P=1.8e-105 and P=0e+00, respectively)Fisher’s exact test to overlap the differentially expressed genes detected in hypoxic placentas, before and after adjusting for cell-type proportions (Figure 6 —figure supplement 1 H, P=8.4e-51).Regarding sample number and effect size: It appears that the animal numbers used for the ChIP-seq were confused with the number of replicates by the reviewers. These details were in Supplementary file 1a. There were 3 replicates per experimental group and each replicate contained sperm from pooled samples that was equalized in cell number and comprised of sperm from n=7 control males, or n=16 HFD males. For the RNA-seq n=4 placentas were used from each experimental group from both males and females for a total N of 16. Although the sample size is moderate, we followed the Canadian Council of Animal Care guideline which calls for the use of the lowest animal number that elicits significant effects (CCAC guidelines p6 “Consideration must also be given to reduction, to determine the fewest number of animals appropriate to provide valid information and statistical power, while still minimizing the welfare impact for each animal”).Regarding intrasample variation: The PCA plot (Figure 2 —figure supplement 1 C) and heatmap (Figure 2 A) show little intra-group variability, and the heatmap (Figure 2 A) shows consistency of the H3K4me3 signal within groups. The PCA plot, which is a statistical analysis, shows that the variance explained by the dietary treatment is greater than the inter-sample variance (Figure 2 —figure supplement 1 C, PC1=51.9% versus PC2=28.9% variance, respectively). Intra-variation is unlikely to driving the significant differences identified between groups for H3K4me3 enrichment differences for the following reasons: we compared H3K4me3 ChIP-seq data sets from this study to our prior obesity model (Pepin et al., 2022) (PMID: 35183795), where individual mice rather than pooled samples were used and determined there was significant overlap in the differentially enriched regions detected in these models. This consistency occurred despite the models coming from different animal facilities, differences in HFD and controls diets and mouse strain.

2) More functional validation to support that the altered placental gene expression causes phenotypes in both placental functions and offspring development and health.

Author response 2. We appreciate that we should have emphasized and written more clearly that we had indeed phenotyped the placentas and offspring metabolic health from the same model we derived the placenta tissue from as we reported in (Jazwiec et al., 2022)(PMID: 35377412). This was referenced in our submitted manuscript (Lines 105-107; 131-133; 135-139; 147-150; 232-235; 270-273; 297-300; 384-386; 433-435; 441-448; 507-514). We have made this more apparent in the manuscript by expanding our description of the offspring phenotypes in the introduction and clarified that it was from this model that the placenta’s used in this study were derived from (Jazwiec et al., 2022) (PMID: 35377412).

Lines 127-139: “In two research settings using different obesity mouse models, we studied male mice exposed to either a preconception control (CON) or a high-fat diet (HFD), and the paternal effects on offspring. As we described in our paternal obesity model (Pepin et al., 2022), sperm from obese males had aberrant H3K4me3 enrichment at genes implicated in metabolism and placenta development, and sired offspring with metabolic disturbances (Pepin et al., 2022). Here we used sperm and placenta from a similar paternal obesity model described in (Jazwiec et al., 2022), where we demonstrated paternal offspring born to obese fathers have impaired whole body energetics and are glucose intolerant. In addition, we showed that pregnancies sired by obese males were associated with placenta showing characteristics of hypoxia, accompanied by histo-morphological changes in blood vessel integrity. Specifically, histological and immunohistology analyses of placentas collected at embryonic day 14.5 and 18.5 in (Jazwiec et al., 2022) revealed that paternal obesity resulted in significant changes in transcription factors that regulate blood vessel development, blood vessel integrity and signalling pathways governed by hypoxia (HIF1A, VEGF and VEGR).”

Reviewer #1 (Recommendations for the authors):1) This study reports H3K4me3 ChIP-seq data on sperm from HFD male mice and the placental tissue of their offspring. By comparing the altered H3K4me3 marks in sperm with the same mark and gene expression levels in the placenta, the authors identified numerous histone epimutation-dysregulated mRNA pairs of a potential causal relationship. The finding is interesting, but the conclusion is premature due to a lack of functionally supportive data. Therefore, the authors need to either tone down their claim or provide more functional data to support their notion. In addition, this reviewer has the following specific comments:

Author response 4: Request for further functional data to support placenta gene expression changes are linked with altered placenta function (also please see author response 2): We appreciate that we should have emphasized and written more clearly that we had indeed characterized placenta function and offspring metabolic health from the same model we derived the placenta tissue from as we reported in (Jazwiec et al., 2022); (PMID: 35377412). This was referenced in our submitted manuscript (Lines 105-107; 131-133; 135-139; 147-150; 232-235; 270-273; 297-300; 384-386; 433-435; 441-448; 507-514). We have made this more apparent in the manuscript by expanding our description of the offspring phenotypes in the introduction and clarified that it was from this model that the placenta’s used in this study were derived from (Jazwiec et al., 2022); (PMID: 35377412). In this study we show that the immunopathological analysis of the placenta is directly in line with our placenta transcriptomic data.

Lines 127-139: “In two research settings using different obesity mouse models, we studied male mice exposed to either a preconception control (CON) or a high-fat diet (HFD), and the paternal effects on offspring. As we described in our paternal obesity model (Pepin et al., 2022), sperm from obese males had aberrant H3K4me3 enrichment at genes implicated in metabolism and placenta development, and sired offspring with metabolic disturbances (Pepin et al., 2022). Here we used sperm and placenta from a similar paternal obesity model described in (Jazwiec et al., 2022), where we demonstrated paternal offspring born to obese fathers have impaired whole body energetics and are glucose intolerant. In addition, we showed that pregnancies sired by obese males were associated with placenta showing characteristics of hypoxia, accompanied by histo-morphological changes in blood vessel integrity. Specifically, histological and immunohistology analyses of placentas collected at embryonic day 14.5 and 18.5 in (Jazwiec et al., 2022) revealed that paternal obesity resulted in significant changes in transcription factors that regulate blood vessel development, blood vessel integrity and signalling pathways governed by hypoxia (HIF1A, VEGF and VEGR).”

The ideal control for HFD is the ingredient control diet manufactured to specifically match the composition of HFD except for in fat and sugar contents. The so-called control chow may contain many components that do not match the HFD, thus contributing to the epimutations detected.

Author response 5. It is worth reminding that we are studying the effects of obesity and not diet. To that effect the HFD induces obesity and metabolic dysfunction while the control diet does not. Although it is fair to point out that the composition of the control diet should be kept in mind, considering the desired outcomes within the scope of the study, the diets elicited the desired phenotypic effects serving as a model for obesity. We see this experimental design as a strength as in the study we compared this model to our previous published obesity model (Pepin et al., 2022) (PMID: 35183795) and there was significant overlap in the regions of differential enrichment detected between both models even though they were conducted in different research settings, with different mouse substrain and different diet combinations (Figure 1 and Figure 2 —figure supplement 1). In our opinion this demonstrates that we are measuring robust effects of paternal obesity that can be replicated under different conditions. This comparative study design has been lacking in the field of epigenetic inheritance.

This is discussed in lines 157-173: “Of the top 5% and top 10% differentially enriched H3K4me3 (deH3K4me3) regions, some were common to those previously identified in Pepin et al., (2022) (128 and 423 overlapping deH3K4me3 regions, Fisher’s exact test P=2.2e-16 and P=2.2e-16, (Figure 2 —figure supplement 1 D i and ii, respectively). This was despite substantial differences in animal models (timing of diet exposure [3 vs 6 weeks of age], control diet [chow vs low-fat diet], and mouse substrain [C57BL/6J vs C57BL/6NCrl]. Focussing on the top 5% regions most impacted by obesity, the majority of regions showed an increase in enrichment for H3K4me3, consistent with our previous study (71.4%, n=1,257 versus n=503, Figure 2 A-B)). Regions losing H3K4me3 showed moderate H3K4me3-enrichment in CON sperm, with predominantly low CpG density, whereas regions gaining H3K4me3 showed low-to-moderate enrichment, with mainly high CpG density (Figure 2 C). Regions not impacted by diet showed high H3K4me3 enrichment in CON sperm, with low and high CpG density (Figure 2 C). Similarly to our previous findings, regions losing H3K4me3 were predominantly located >5 kilobase (kb) from the transcription start site (TSS), likely at transposable elements, putative enhancer regions, and intergenic regions (Figure 2 D i) (Pepin et al., 2022). Regions gaining H3K4me3 in HFD sperm were located near the TSS (within 1 kb), likely at promoter regions (Figure 2 D ii). Furthermore, obesity-associated deH3K4me3 at promoter regions detected in each study showed enrichment at similar gene ontology (GO) processes, such as metabolic processes, cellular stress responses, transcription, and development (Figure 2 —figure supplement 1 E, Supplementary files 1b-d, Supplementary file 2).”

2) The sample size appears to be very small, and intra-samples/mice variations in the histone marks might be greater than the effect size. Unless proper statistical analyses are employed to prove otherwise, the differences detected may not represent true signals.

Author response 6.

Regarding effect and sample size: It appears that on review the animal numbers used for the ChIP-seq were confused with the number of replicates by the reviewers. These details were in Supplementary file 1a. There were 3 replicates per experimental group and each replicate contained sperm from pooled samples that was equalized in cell number and comprised of sperm from n=7 control males, or n=16 HFD males. For the RNA-seq n=4 placentas were used from each experimental group from both males and females for a total N of 16. Although the sample size is moderate, we followed the Canadian Council of Animal Care guideline which calls for the use of the lowest animal number that elicits significant effects (CCAC guidelines p6 “Consideration must also be given to reduction, to determine the fewest number of animals appropriate to provide valid information and statistical power, while still minimizing the welfare impact for each animal”).Regarding intrasample variation: The PCA plot (Figure 2 —figure supplement 1 C) and heatmap (Figure 2A) show little intra-group variability, and the heatmap (Figure 2A) shows consistency of the H3K4me3 signal within groups. The PCA plot shows that the variance explained by the dietary treatment is greater than the inter-sample variance (Figure 2 —figure supplement 1 C, PC1=51.9% versus PC2=28.9% variance, respectively). Intra-variation is unlikely to be driving the differences identified between groups for H3K4me3 enrichment differences for the following reasons: we compared H3K4me3 ChIP-seq datasets from this study to our prior obesity model (Pepin et al., 2022) (PMID: 35183795), where individual mice rather than pooled samples were used and determined there was significant overlap in the differentially enriched regions detected in these models. This consistency occurred despite the models coming from different animal facilities, differences in HFD and controls diets and mouse substrain.

We added the following to the text for clarification:

Lines 539-541: “Pools of sperm from 2-7 male mice per sample were used for ChIP-seq (Supplementary file 1a). Each pool was equalized in cell number and comprised of sperm from n=8 control males, or n=16 HFD males. This pooling was required to create replicates with an equal amount of total sperm. “

The following statistical additions have been made:

Fisher’s exact test to overlap paternal obesity-associated differentially enriched region of H3K4me3 (deH3K4me3) compared to paternal obesity-associated deH3K4me3 identified in our other obesity model (Pepin et al., 2022) (PMID: 35183795) (Figure 2 —figure supplement 1 Di and ii).Fisher’s exact test to overlap paternal obesity-associated differentially enriched regions of H3K4me3 (deH3K4me3) with female and male placenta differentially enriched genes (Figure 4 —figure supplement 1 Di and ii).Fisher’s exact test to overlap paternal obesity-associated differentially expressed genes in male placentas with hypoxia-related differentially expressed genes in placentas (Chu et al., 2019) (Figure 6 D; P=5.1e-16)Fisher’s exact test to overlap the top 5% or top 10% regions contributing to PC1 and associated with sample separation according to dietary group, from this study and our previous study (Pepin et al., 2022) (Figure 2 —figure supplement 1 D i and ii, P=2.2e-16 and P=2.2e-16, respectively).Fisher’s exact test to overlap paternal obesity-induced de-regulated genes between female and male placentas with the top 5% and the top 10% promoter regions most sensitive to obesity in sperm (Figure 4 —figure supplement 1 D i and ii, see figure for P values)Fisher’s exact test to overlap paternal obesity-induced de-regulated genes in female and male placentas, with the nearest gene to placental-specific enhancer overlapping sperm deH3K4me3, using the top 5% or top 10% sperm regions associated with dietary group (Figure 4 —figure supplement 1 E i-iv; see figure for P values)Fisher’s exact test to overlap the differentially expressed genes in female and male placentas, before and after adjusting for cell-type proportions (Figure 5 —figure supplement 3 G and H, P=1.8e-105 and P=0e+00, respectively)Fisher’s exact test to overlap the differentially expressed genes detected in hypoxic placentas, before and after adjusting for cell-type proportions (Figure 6 —figure supplement 1 H, P=8.4e-51).

3) The notion that sperm tend to regulate placental gene expression needs to be supported experimentally.

Author response 7: Our experimental data showed paternal obesity associated with deH3K4me3 in sperm at genes implicated in placenta formation and metabolism (this study and Pepin, Lafleur, Lambrot, Dumeaux, and Kimmins, 2022) (PMID: 35183795). This aligns with the metabolic dysfunction in offspring and abnormal placenta development we reported in this model (Jazwiec et al., 2022) (PMID: 35377412). To address this comment, in the revised version we provide more details on the phenotypic and immuno-histopathological changes in placenta sired by obese males collected from our same model as reported in Jazwiec et al., 2022 (PMID: 35377412). These functional and phenotypic changes are linked to our experimental data identifying epigenomic alterations in sperm (Figure 2 and Figure 4 —figure supplement 1) and transcriptomic alterations in placenta and to developmental changes in cellular distribution as described in Figures 4-6. The sperm epigenomic link between placenta function and transcriptome is shown in Figure 4 —figure supplement 1. Briefly, 98 and 110 deH3K4me3 regions overlap DEGs in female and male placentas respectively (Figure 4 —figure supplement 1 Dii). While not statistically significant we would argue that this does not rule out biological significance. For example, in a recent epigenome editing paper from (Takahashi et al., 2023) (PMID: 36754048), they targeted CpG islands in promoters for either the *Ankrd26* or *Ldlr* genes in embryonic stem cells, then injected the edited cells into 8-cell embryos, to give rise to mice with metabolic phenotypes. This is a key example showing that a single epigenetic change at one gene can give rise to a phenotype. Furthermore, an accumulation of evidence shows that the paternal genome and epigenetic imprinting is critical for placental development. This was first proposed ~40 years ago with studies showing that the paternal and maternal genomes display divergent potential for the development of the inner cell mass versus the trophectoderm (PMIDs: 6482961, 6722870, 6709062, 3834032, 3625116). More specifically, through genetic manipulation studies, it was demonstrated that androgenetic embryos have a greater developmental potential for the trophoblast lineage compared to parthenogenetic embryos. Further evidence supporting this notion is the finding that paternally expressed genes are enriched in the placenta (PMID: 23754418). All things considered, we therefore interpret our experimental data as: obesity altered sperm H3K4me3 enrichment levels may lead to altered placenta gene expression and development. With the cumulative effects being abnormal placenta function and metabolic phenotypes as we demonstrated our same model (Jazwiec et al., 2022); (PMID: 35377412) from which the placenta was derived for the genomic analysis.

We made the following revisions to the text:

Lines 127-139: “In two research settings using different obesity mouse models, we studied male mice exposed to either a preconception control (CON) or a high-fat diet (HFD), and the paternal effects on offspring. As we described in our paternal obesity model (Pepin et al., 2022), sperm from obese males had aberrant H3K4me3 enrichment at genes implicated in metabolism and placenta development, and sired offspring with metabolic disturbances (Pepin et al., 2022). Here we used sperm and placenta from a similar paternal obesity model described in (Jazwiec et al., 2022), where we demonstrated paternal offspring born to obese fathers have impaired whole body energetics and are glucose intolerant. In addition, we showed that pregnancies sired by obese males were associated with placenta showing characteristics of hypoxia, accompanied by histo-morphological changes in blood vessel integrity. Specifically, histological and immunohistology analyses of placentas collected at embryonic day 14.5 and 18.5 in (Jazwiec et al., 2022) revealed that paternal obesity resulted in significant changes in transcription factors that regulate blood vessel development, blood vessel integrity and signalling pathways governed by hypoxia (HIF1A, VEGF and VEGR).”

The following text has been added or edited to better emphasize the sperm placenta connection:

Lines# 107-126: “Notably, genetic manipulation studies have determined that the paternal genome has greater potential for extraembryonic and trophoblast development compared to the maternal genome, and paternally expressed genes dominate placenta gene expression (S. C. Barton, Adams, Norris, and Surani, 1985; Sheila C. Barton, Surani, and Norris, 1984; J. McGrath and Solter, 1986; James McGrath and Solter, 1984; Surani, Barton, and Norris, 1984; Wang, Miller, Harman, Antczak, and Clark, 2013). Additionally, we recently showed that sperm chromatin profiles highly resemble that of trophectoderm and placenta tissue, and H3K4me3-enriched regions in sperm are expressed in these tissues, highlighting a molecular connection from the male germline to the developing placenta (Pepin et al., 2022).

The connection between paternal gene expression and placenta development has led to a growing interest in the role of paternal factors in placental development and function and offspring health (Wang et al., 2013). In mice, we demonstrated that paternal folate deficiency was associated with an altered sperm epigenome, differential gene expression in the placenta, and abnormal fetal development (Lambrot et al., 2013). In other mouse models, advanced paternal age and toxicant exposure have been linked to altered placental imprinting and reduced placental weight, and paternal obesity was linked to alterations in placental DNA methylation, aberrant allocation of cell lineage to trophectoderm (TE) (Binder, Hannan, and Gardner, 2012; Binder, Sheedy, Hannan, and Gardner, 2015; Denomme et al., 2020; Ding, Mokshagundam, Rinaudo, Osteen, and Bruner-Tran, 2018). Male partner metabolic syndrome and being overweight have been associated with an increased risk for pre-eclampsia and negative pregnancy outcomes (Lin, Gu, and Huang, 2022; Murugappan et al., 2021).”

4) It is essential to phenotype the offspring to determine whether the changes in placental transcriptome indeed have an impact on offspring metabolic states.

Author response 8: We appreciate that we should have better emphasized that we had indeed phenotyped the offspring metabolic states from this same obese mouse model, from which the placenta’s used here were derived from. We referenced these results in our manuscript (Lines 105-107; 131-133; 135-139; 147-150; 232-235; 270-273; 297-300; 384-386; 433-435; 441-448; 507-514). To summarize, as we reported in (Jazwiec et al., 2022) (PMID: 35377412), male and female young adult offspring sired by either control or obese male mice underwent metabolic profiling using a Comprehensive Lab Animal Monitoring System and were subjected to a glucose tolerance test to assess changes in glucose tolerance. In Jazwiec et al., we report that offspring born to obese fathers have impaired whole body energetics and are glucose intolerant as young adults (Figure 7 in the published paper). We also showed that obese sires produce hypoxic placentae – accompanied by changes to placental vessel integrity. In the current study, we set out to investigate whether obesity induced changes in sperm histones which are potentially upstream of placental signalling pathways associated with vessel development, and/or hypoxia signalling pathways – we show this to be the case. To further substantiate this finding, we used an available data set based on the placenta transcriptome from a hypoxic model (Chu et al., 2019) (PMID: 30718791). We compared how paternal preconception obesity and in utero late-gestational exposure to hypoxia similarly impact the placental transcriptome and cell-type proportions. Here we show that placenta transcriptomes as well as tissue cellular compositions altered by hypoxia were similarly altered in placenta sired by obese males (Figures 4-6 and Figure 5 —figure supplement 2-3 and Figure 6 —figure supplement 1).

Reviewer #2 (Recommendations for the authors):General comments:The text is well written, and the figures are generally well presented and clear. The general flow of the manuscript could be improved, as it is not always entirely clear to the reader what the authors were thinking when carrying out a set of experiments. More specific suggestions and comments are provided below.1. Specifically, regarding the general flow, the part about the transcription factor binding motifs is interesting but the authors should specify what they hypothesize may be happening: Would transcription in the sperm be altered prior to conception? Do they think this is carried over into the zygote? Where would the transcription factors be coming from?

Author response 11: We altered the text in the Results section to clarify these points:

Lines 197-200: “Many TSS in sperm marked by H3K4me3 are enriched for transcription factor (TF) binding sites and bound by components of the transcriptional machinery complex (phosphorylated RNA pol II and Med12). In sperm, such TF interactions have been reported to occur at open chromatin regions and suggested to potentially confer gene expression in the embryo (Jung et al., 2017).”

2. In the part where sequencing data from Chu et al. (2019) is used, more information is needed about this study to put the presented findings into context for the reader: What kind of model for hypoxia is used in this publication? Why was this chosen and not any other paternal intervention – the justification l. 304 "Placentas derived from obese sires, like hypoxic placentas, exhibit changes in gene expression and altered angiogenesis, vasculature, and development" is very unspecific and likely applies to a number of different interventions therefore the choice of this dataset and model over others should be justified well.

Author response 12: We address this point with the following revisions:

Lines 294-305: “During placental development, hypoxia is a tightly regulated process that is essential for proper vascular formation supporting fetal growth. Hypoxia is also a hallmark of placental insufficiency, reflecting poor oxygen and nutrient supply to the fetus, resulting in fetal growth restriction, low birth weight, and consequently heightened risk for cardiometabolic disease. Placentas derived from obese sires, like hypoxic placentas, exhibit changes in gene expression and altered angiogenesis, vasculature, and development (Binder et al., 2015, 2012; Jazwiec et al., 2022; Lin et al., 2022; McPherson et al., 2015; Mitchell et al., 2017, and this study). To determine whether transcriptomic and pathological phenotypes in paternal obese-sired placentas relate to that of hypoxic placentas, we compared our HFD placenta RNA-seq data to a hypoxia-induced IUGR mouse model RNA-seq dataset (Chu et al., 2019). In this model, hypoxia was induced during late gestation, which resulted in pregnancies with aberrant placental transcriptome, IUGR, decreased birth weights, and offspring exhibiting adult-onset cardiometabolic disturbances (Chu et al., 2019).”

3. The manuscript would greatly benefit from an additional experiment probing a causal relationship between the examined mechanisms. At the moment it is full of correlations, some of which are overstated, and the limitations of the methods are not discussed enough.

Author response 13 (also please see response 3 and 10): We agree with this reviewer that our study is correlative, and that causal relationships between obesity-induced sperm epimutations and the placental transcriptome and functions should be investigated in the future using epigenome edited approaches. We also agree that we should delineate the limitations of our approach and analysis. We have modified the text in response to these concerns that some of our remarks come across as overstating our findings including modifying the text and elaborating on the limitations of the study in the discussion. However before progressing to epigenome editing in sperm these experimental studies were a necessary proof of principle and make a major advance for the field. The novelty in our study is that we have identified a paternal factor (obesity) that effect the sperm epigenome and showed that it is a non-genetic route influencing placenta development and offspring health. Also, the paternal contribution to placental insufficiency is understudied, and how paternal preconception exposures including obesity can impact placental development is unknown.

Lines 1-2: Revised title to: Determining the effects of paternal obesity on sperm chromatin at histone H3 lysine 4 tri-methylation in relation to the placental transcriptome and cellular composition

In the discussion, we expand on the limitations of the study and softened the conclusions that can be drawn from our work:

Lines 402-421: “We previously reported in the same model from which our placenta samples are derived that paternal obesity leads to functional and histopathologic abnormalities in placenta and dysfunctional offspring metabolism. The epigenomic link between sperm enrichment differences in H3K4me3 and DEGs in placenta are minor. While not statistically significant we would argue that this does not rule out the likely biological significance of deH3K4me3 in sperm. For example, in a recent epigenome editing model from (Takahashi et al., 2023) using embryonic stem cells, CpG islands in promoters for either the *Ankrd26* or *Ldlr* genes were targeted to alter DNA methylation then injected into 8-cell embryos. These embryos gave rise to mice with metabolic phenotypes. This is a key example showing that a single epigenetic change at one gene can give rise to a phenotype (McNamara et al., 2018; Nativio et al., 2011; Takahashi et al., 2023). In addition, the limited overlap between sperm deH3K4me3 and placenta DEGs may also reflect the terminally differentiated state and heterogenous nature of the placenta at E14.5. Perhaps a greater correspondence between sperm deH3K4me3 may have been observed at earlier developmental timepoints such as at first lineage segregation corresponding to trophoblast formation. In future studies, it will be worthwhile to examine the trophoblast gene expression in comparison to deH3K4me3 in sperm. Indeed, we previously showed by *in silico* analysis that most regions bearing H3K4me3 in sperm are enriched for this mark in trophectoderm (TE), correlate with TE H3K4me3 signal, and correspond to genes expressed in TE (Pepin et al., 2022). Another limitation of this study is that placenta profiles are from bulk RNA-seq and measures average gene expression across a heterogenous cell population and identification of DEGs can therefore be confounded by cell composition. To address this, we used a deconvolution approach in our analysis using a scRNA-seq dataset.”

Lines 423-438: “The identification of alterations of cell type proportion must be considered within the limitations of a deconvolution analysis. This analysis only provides estimates of cell-type relative within a heterogeneous tissue. While this allowed us to adjust for the effect of differences in cell-type composition, the exact cell-type composition and their specific gene expression changes need to be validated by single-cell approaches such as single-cell RNA-seq or spatial transcriptomics. Furthermore, even though we used a reference dataset which included cells representative of placental tissues, the detection capacity of this approach is limited for low-abundant cell types, such as blood cells, immune cells, and inflammatory cells, which would be informative of placental pathological states. For example, aberrant abundance of decidual inflammatory cells, such as natural killer (NK) cells, has been linked to the pathogenesis of preeclampsia (Aneman et al., 2020; Bachmayer, Rafik Hamad, Liszka, Bremme, and Sverremark-Ekström, 2006; Du et al., 2022; Milosevic-Stevanovic et al., 2016; Williams, Bulmer, Searle, Innes, and Robson, 2009). Incidentally, it was previously shown that paternal diet-induced obesity is associated with placental inflammation (Claycombe-Larson, Bundy, and Roemmich, 2020; Jazwiec et al., 2022). Interestingly, many GO terms related to inflammatory processes were enriched in the obesity-induced deH3K4me3 in sperm (Figure 2 E, Supplementary file 1c, and Pepin et al., 2022), suggesting sperm deH3K4me3 might be partly influencing placental inflammation. However due to the low representation of immune cells in the data set this could not be assessed.”

Lines 478-484: “Next, it will be important to assess earlier developmental time points to determine when and how these paternally-induced effects originate. Indeed, studies have shown that paternal obesity alters preimplantation development, such as cellular allocation to TE versus ICM lineages (Binder et al., 2012). Investigating multiple and earlier time points would help reveal the dynamic trajectory of paternally-induced deregulated transcriptomic and epigenetic signatures which might be at the origin of adult-onset disease. Translating these findings to humans will be beneficial to further understand and emphasize the paternal preconception contribution to pregnancy outcomes, placental integrity and offspring health.”

4. Regarding the Methods, the control diet is not the best control for the HFD, because it is from a different supplier. The fact that it is from a different manufacturer means that the content, not just fat levels, is likely completely different. This should at least be mentioned by the authors, within the limitations of this methodological issues addressed in the text. However, ideally the key experiments should ideally be performed again, using controlled diets (high fat and lower/control fat levels) from the same manufacturer, so that any differences can be related to fat (rather than potentially other molecular constituents that differ between manufacturers).

Author response 14: This concern has been addressed in author response 5.

5. Were the males group or single housed during the diet intervention? How much weight did the HFD males put on compared to controls i.e. how severe was the obesity? How much food did they consume? Was their reproductive function impacted? And is this why n value for HFD was double that of controls? Some information about this aspect (e.g. in further supplementary figures) would be informative.

Author response 15: We added to the following information for clarification:

Lines 502-516: “Six-week-old C57BL/6J male mice were co-housed (2 per cage) and randomly allocated to either the control (n=8; CON; standard chow diet, Harlan 8640, Teklad 22/5 Rodent Diet; 17% kcal fat, 54% kcal carbohydrates, 29% kcal protein, 3 kcal/g) or high-fat diet (n=16; HFD; Research Diets Inc, D12492; 20% kcal protein, 20% kcal carbohydrates, 60% kcal fat, 5.21 kcal/g) group, for 10-12 weeks. All animals had free access to water and food ad libitum, housed in the same room which was maintained at 25°C on a controlled 12-hour/12-hour light/dark cycle. Two weeks prior mating and throughout the mating period, male mice were housed individually to prevent fighting and aggression, and to facilitate timed-mating. We have previously reported and characterized this model of high-fat diet-induced obesity (Jazwiec et al., 2022). As reported in our previous work, male mice fed a high-fat diet became significantly heavier weighing approximately 40 g (20 g in weight gain); had elevated body adiposity; elevated fasting blood glucose; were glucose intolerant; had significantly increased serum insulin concentrations, and were insulin resistant as assessed by HOMA-IR (Jazwiec et al., 2022). Food consumption data was not collected. High fat male mice had impaired mating efficiency (Jazwiec et al., 2022), and therefore more males were needed in the high fat fed group compared to controls to generate a sufficient number of pregnancies and placental tissue for RNA-sequencing experiments.”

Given that we have thoroughly documented the male phenotype, and we reference it in our manuscript (Lines 105-107; 131-133; 135-139; 147-150; 232-235; 270-273; 297-300; 384-386; 433-435; 441-448; 507-514), and the aim of our current study was to investigate the relationship between obesity-induced sperm histone changes and placental signalling pathways, we do not believe it necessary to include this information in the current manuscript. The impact of a high fat diet on male metabolism has been thoroughly, and repeatedly investigated previously (Binder, Beard, et al., 2015; Binder et al., 2012; Binder, Sheedy, et al., 2015; Chambers, Morgan, Heger, Sharpe, and Drake, 2016; de Castro Barbosa et al., 2016a; Ghanayem, Bai, Kissling, Travlos, and Hoffler, 2010; Jazwiec et al., 2022; Ng et al., 2010; Pepin et al., 2022; Schjenken et al., 2021).

6. Several descriptions of procedures appear vague and need to be clarified. Why were the procedures not always the same? In which cases did they vary? The authors should be clear why there were various different protocols in place.– E.g.: high-fat diet was given to males for "8-10 weeks"– E.g.: Males were mated with "1-2 females"– E.g.: Pools of sperm for ChIP-seq consisted of "2-7 males"This detail specifically, in combination with the knowledge that F0 groups were n=8 for the controls and n=16 for the HFD, gives the impression that the samples were likely not equal for both experimental groups in that the pools for HFD would have been systematically larger, leading to deeper sequencing and other differences? This may constitute a methodological flaw as it may introduce a bias in the sequencing data.

Author response 16: We added details to the methods for clarification. Sperm was collected from (for chromatin immunoprecipitation sequencing experiments) male mice fed either a standard chow diet or a high fat diet for a minimum of 8 weeks before timed-mating – it often takes a few weeks to confirm pregnancies – thus this timing varies (see (Jazwiec et al., 2022); (PMID: 35377412)). This is normal for mouse breeding studies to have a small range and several weeks that are unlikely to change the results as the mice are still being bred with sperm from the same spermatogenic cycle. Importantly all mice were similarly altered in metabolic profiles where at 8 or 10 weeks as detailed in our paper (Jazwiec et al., 2022); (PMID: 35377412).

To generate pregnancies, a single male mouse was paired with either one or two females overnight. When a copulation plug was identified in a female mouse, this individual mouse is housed individually, and body weight was monitored for 10 days to determine whether she was pregnant. Which then leaves one individual mouse, left with a male overnight to mate once again. Therefore, it is correct to state that males are housed with 1-2 females overnight.

We added the following text:

Lines 502-516: “Six-week-old C57BL/6J male mice were co-housed (2 per cage) and randomly allocated to either the control (n=8; CON; standard chow diet, Harlan 8640, Teklad 22/5 Rodent Diet; 17% kcal fat, 54% kcal carbohydrates, 29% kcal protein, 3 kcal/g) or high-fat diet (n=16; HFD; Research Diets Inc, D12492; 20% kcal protein, 20% kcal carbohydrates, 60% kcal fat, 5.21 kcal/g) group, for 10-12 weeks. All animals had free access to water and food *ad libitum*, housed in the same room which was maintained at 25°C on a controlled 12-hour/12-hour light/dark cycle. Two weeks prior mating and throughout the mating period, male mice were housed individually to prevent fighting and aggression, and to facilitate timed-mating. We have previously reported and characterized this model of high-fat diet-induced obesity (Jazwiec et al., 2022). As reported in our previous work, male mice fed a high-fat diet became significantly heavier weighing approximately 40 g (20 g in weight gain); had elevated body adiposity; elevated fasting blood glucose; were glucose intolerant; had significantly increased serum insulin concentrations, and were insulin resistant as assessed by HOMA-IR (Jazwiec et al., 2022). Food consumption data was not collected. High fat male mice had impaired mating efficiency (Jazwiec et al., 2022), and therefore more males were needed in the high fat fed group compared to controls to generate a sufficient number of pregnancies and placental tissue for RNA-sequencing experiments.”

Regarding the number of males used for ChIP-seq. The animal N used was 7 (control) and 16 (HFD) mice. The pooling was required to create replicates with an equal amount of total sperm to perform the ChIP-seq. The strength in pooling samples is that you are assessing the most robust changes within the experimental group. The details on samples and pooling can be found in Supplementary file 1a. With regards to sequencing bias that is highly unlikely. The replicates were based on the same cell number per sample, and the sequencing depth was excellent for each replicate ranging from 37-45M reads (Supplementary file 1a). Differences in library size are accounted for in the normalization.

We have added the following to the text for clarification:

Lines 541-543: “Pools of sperm from 2-7 male mice per sample were used for ChIP-seq (Supplementary file 1a). Each pool was equalized in cell number and comprised of sperm from n=8 control males, or n=16 HFD males. This pooling was required to create replicates with an equal amount of total sperm.”

7. With n=3 in ChIP-seq and n=4 per group in RNA Seq, the sample sizes are on the lower end (bare minimum) of what is considered valid. These analyses are likely underpowered and at least require independent validation, especially if the role of specific genes is discussed as mechanistically relevant.

Author response 17: Regarding effect and sample size: It appears that on review the animal numbers used for the ChIP-seq were confused with the number of replicates by the reviewers. These details were in Supplementary file 1a. There were 3 replicates per experimental group and each replicate contained sperm from pooled samples that was equalized in cell number and comprised of sperm from n=7 control males, or n=16 HFD males. For the RNA-seq n=4 placentas were used from each experimental group from both males and females for a total n of 16. Although the sample size is moderate, we followed the Canadian Council of Animal Care guideline which calls for the use of the lowest animal number that elicits significant effects (CCAC guidelines p6 “Consideration must also be given to reduction, to determine the fewest number of animals appropriate to provide valid information and statistical power, while still minimizing the welfare impact for each animal”).

Regarding intrasample variation: The PCA plot (Figure 2 —figure supplement 1 C) and heatmap (Figure 2 A) show little intra-group variability, and the heatmap (Figure 2 A) shows consistency of the H3K4me3 signal within groups. The PCA plot shows that the variance explained by the dietary treatment is greater than the inter-sample variance (Figure 2 —figure supplement 1 C, PC1=51.9% versus PC2=28.9% variance, respectively). Intra-variation is unlikely to driving the significant differences identified between groups for H3K4me3 enrichment differences for the following reasons: we compared H3K4me3 ChIP-seq data sets from this study to our prior obesity model (Pepin et al., 2022) (PMID: 35183795), where individual mice rather than pooled samples were used and determined there was significant overlap in the differentially enriched regions detected in these models. This consistency occurred despite the models coming from different animal facilities, differences in HFD and controls diets and mouse substrain.

Regarding validation: We used a high standard of computational validation and visualization strategies, to ensure confidence in genomic data. This also allowed for a comprehensive understanding of the biological and physiological impacts of paternal obesity on the sperm epigenome and placenta transcriptome. In our experimental design we also included biological and technical replicates. Together these methods provide robustness checks of the experimental data and support our conclusions. These are the validation strategies we used:

Technical and experimental validation

We evaluated the quality of sequencing data using metrics of read quality, alignment and coverage. These are summarized in Supplementary file 1a.Visualized and performed statistical analysis of data to check for anomalies and discrepancies, Pearson correlation analysis shown on heatmap to look for variance and patterns in samples- all here highly correlated (Figure 2 —figure supplement 1 B and Figure 4 —figure supplement 1 A). We checked for batch effects and normalized the data (Figure 4 —figure supplement 1 B) we used PCA plot analysis as a second check for sample behaving oddly (Figure 2 —figure supplement 1 C and Figure 4 —figure supplement 1 C).We used a deconvolution approach to improve the biological meaning of our bulk RNA-seq data (Figure 6, Figure 5 —figure supplement 1 and 2).Performed functional enrichment analysis to gain insight into biological functions, pathways, and genome ontology and visualized individual regions identified to be altered as a confirmation (Figure 2 D and 2 E; Figure 4 E and F; Figure 6, Figure 2 —figure supplement 1 E; Figure 3 —figure supplement 1).

Comparison to external data sets**:**

We compared our data with external data sets using the same tissues and cell and to our prior studies: (a) We compared ChIP-seq data from this obesity model with our former obesity ChIP-seq data (Figure 2 —figure supplement 1); (b) re-analyzed and compared placenta RNA-seq data from an *in utero* exposure hypoxia model that shared similar offspring and placenta phenotypes as we observed in the obesity model (Figure 6 and Figure 6 —figure supplement 1).We used a deconvolution approach to improve the biological meaning of our bulk RNA-seq data (Figure 6, Figure 5 —figure supplement 1 and 2).

Statistical Significance and False Discovery Rate (FDR):

We applied statistical tests and multiple testing corrections to reduce the likelihood of false positives (See also response 1 for additional testing added to the revised manuscript)

8. In the same vein, in the GO analysis in Figure 2E(i), many of the significantly enriched pathways are only represented with a single gene, including the placenta morphogenesis pathway. The authors seem to overstate this result on l. 234 by using the plural of genes: "As deH3K4me3 in sperm was located at genes involved in placental formation (Figure 2 E and Pepin et al., 2022), we assessed whether paternal obesity was associated with changes in gene expression of the placenta." If this result is assigned such importance to, it would be important to validate the result.

Author response 18: Good points. We made the following revisions: Figure 2E(i and ii) have been removed from the manuscript. The text has been updated accordingly. This figure reflected a GO analysis focused on deH3K4me3 regions showing a decrease in enrichment, which represented n=503 regions, of which most were located at intergenic regions. Because our GO analyses are focused at regions located at promoters, this analysis was therefore restricted to only n=36 promoters. This explains why some of the significantly enriched GO terms contained only 1 gene. It is also worth noting that there is a lack of research on the placenta relative to research on most (if not all) other mammalian organs, and consequently there are currently only 18 gene ontology terms annotated for the placenta, hindering the effectiveness of gene ontology analyses to study effects on placental functions and limiting the discovery capacity of various conditions on placental effects (Naismith and Cox, 2021)(PMID: 34237528).

9. In the Results, the formulation of some results gives the wrong impression/overstates the actual result. The authors should be generally more careful in their statements and qualify them.10. For example, on l.166: "Despite differences in experimental design and animal models, we found a significant overlap in regions showing differential (H3K4me3 deH3K4me3) from both studies (128 overlapping regions, Fisher's exact test P=2.2e-16, Figure S1 D)". These are 128 regions out of 1632 and 1410 regions, so less than 10% overlap. Therefore, in fact this is a surprisingly low overlap despite methodological similarities, and should be discussed as such, not the opposite.

Author response 18 (and also response 7): We respectfully disagree that we have overinterpreted the findings regarding overlap between the two studies in the context of the differences in experimental, methodological, exposure window, control diets and mouse substrains. These differences are highlighted in lines 159-161. Across the two studies, the animals were generated independently in two different animal facilities (McMaster University and McGill University), using two different mouse substrains (C57BL6J versus C57BL6NCrl), which have been shown to show varying degrees of severity of phenotypes induced by high-fat diet feeding (Hull et al., 2017; Mekada et al., 2009; Mekada and Yoshiki, 2021; Siersbæk et al., 2020) (PMID: 28138002; 19448337; 33441510; 32820201). The control diets differed across studies (chow diet versus low-fat diet) which could differentially impact metabolism (Hu et al., 2018)(PMID: 30017356) and consequently the epigenome, therefore the epigenetic differences when comparing chow- or low-fat diet feeding to high-fat diet could vary. Furthermore, the dietary interventions were performed at different time points; starting at 3 weeks during the first wave of spermatogenesis, versus at 6 weeks entering the second wave of spermatogenesis. These waves are known to be transcriptionally unique (Hermann et al., 2018). In terms of the ChIP-seq experiment, in this study the sperm from multiple mice was pooled in each sample, versus in our previous study sperm from individual mice were used per sample.

Therefore, our interpretation of the data is that, remarkably, despite substantial differences in terms of animal models, experimental design, and methodological approaches, we obtain a statistically significant overlap of regions detected across there two studies, as tested by a Fisher’s exact test. Further supporting similarities across the two datasets generated, the diet-sensitive genomic regions detected show common feature characteristics. These observations support the idea that diet-induced obesity associated regions are not random and therefore not as a result of chance.

In light of your comment, we have altered the text and analysis as follows: We extended the analysis to include overlapping both region sets with the top 10% regions impacted by diet.

Lines 157-159: “Of the top 5% and top 10% differentially enriched H3K4me3 (deH3K4me3) regions, some were common to those previously identified in Pepin et al., (2022) (128 and 423 overlapping deH3K4me3 regions, Fisher’s exact test P=2.2e-16 and P=2.2e-16, Figure 2 —figure supplement 1 D i and ii, respectively).”

11. Another example is l. 322: "Overall, the trends for directionality of changes in specific cellular abundances were consistent across the two mouse models (Figure 6B)." While the effects on cell type abundance cannot be said to substantially diverge, saying they are consistent may be overstating things. From the data, it appears that hypoxia has a much stronger effect. This should be discussed in terms of differences between the models etc, not glossed over as "consistent".

Author response 19: It is expected that the effects observed in the hypoxic placenta model are greater than that of our paternal obesity model. Indeed, the former is a direct exposure of hypoxia inducing placental insufficiency during gestation, while in contrast, our model represents a paternal preconception exposure model, whereby the developing placenta is not directly exposed. Consequently, our interpretation of the data is that it is remarkable that a direct (gestational) exposure model can show similar effects as that of a paternal preconception exposure model, with less pronounced effects, as expected.

To address your concern, we revised the text:

Lines 317-319: “It is intriguing that despite the hypoxia model being a direct exposure to the developing placenta and the obesity model being a paternal pre-conception exposure, there were similar trends for directionality of changes in cell-type abundances (Figure 6B).”

12. In Figure S3D (ii)/(iii): Do those genes which are differentially expressed in males and females from HFD fathers and also differentially methylated in sperm (45 in F, 48 in M) overlap at all?

Author response 19: We updated Figure 4 —figure supplement 1 D to show the overlap of genes that are commonly de-regulated in male and female placentas sperm in relation to sperm deH3K4me3 and performed the appropriate statistical analysis.

13. The authors state they perform the cell-type based deconvolution of sequencing data because transcriptional differences between cell types can be obscured by averaging of expression data over many cell types in Bulk tissue RNA-Seq. This is true, however the re-analysis actually leads to a much lower number of DEGs detected (as shown in FiguresS6G,H). This should be noted and discussed by the authors.

Author response 19: The list of detected DEGs is indeed smaller as this analysis takes into account estimated cell type proportions and therefore in theory removes the initially detected DEGs associated with changes in cell-type proportion. A similar analysis using human term placenta (normal versus preeclamptic) showed that differential expression analysis adjusted for estimated cell-type proportions via data deconvolution resulted in the detection of zero DEGs (Campbell et al., 2023)(PMID: 36914823).

We made the following revisions to the text:

Lines 282-290: “This reduction in the number of detected DEGs before versus after accounting for cellular composition suggests that changes in cell-type proportions at least partly drive tissue-level differential expression. This is consistent with the recent finding that preeclampsia-associated cellular heterogeneity in human placentas mediates previously detected bulk gene expression differences (Campbell et al., 2023). There were similarities between the bulk RNA-seq and deconvoluted analysis in that there was overlap of DEGs detected before and after adjusting for cell-type proportions (Figure 5 —figure supplement 3 G and H, Fisher’s exact test P=1.8e-105 and P=0e+00, respectively). This differential gene expression analysis accounting for cellular composition provides insight into how paternal obesity may impact placental development and function and underscores the contribution of cellular heterogeneity in this process.”

14. In the same deconvolution analysis, the authors state they detect 15 cell types, out of 28 detected by the original study from which the dataset was taken. Why do the authors think this is the case? This means almost half the cell types are not included in the tissue sample? Or are they too low in abundance to detect? This should be mentioned.

Author response 20: The cell types that were not detected in our dataset are mainly immune or blood cells (11 cell types), and two types of trophoblasts cells (trophoblast progenitor cells and spongiotrophoblasts) that are characterized by high expression of a specific gene. These trophoblast cells were found in relatively lower abundances (68/4346 and 180/4346) compared to other cell types that may show similar transcriptomic profiles and therefore may be more difficult to differentiate from one another in this analysis. Overall, the undetected cell types are those that are likely to be found in lower abundance in the tissue. The number of cells profiled per cell type in the reference dataset can be found in Supplementary file 1g.

We have edited the text to mention this information:

Lines 260-264: “Of the 28 different cell types identified (Han et al., 2018) (Figure 5 —figure supplement 1 A, Supplementary file 1g), we detected 15 cell types in our deconvolved placenta bulk RNA-seq data (Figure 5 A and Figure 5 —figure supplement 2 A). The remaining undetected cell types include blood/inflammatory cells and rare or poorly characterized trophoblast subtypes that are found in relatively lower abundances in the reference dataset (Supplementary file 1g).”

15. The Discussion seems a little superficial. It should delve more into the implications and limitations of the study.

Author response 21 (see also response 13): We made significant changes in the revised version of the discussion including the implications and limitations of the study and its findings. These are detailed in response 13.

16. The discussion of specific genes is inappropriate in the absence of validation experiments.

Author response 22 (see also response 17 regarding computational and statistical validation):

In our opinion it is appropriate to highlight genes that are differentially expressed or enriched in the analyses we have performed, as is the case in most genomic based studies. It is acceptable not to experimentally validate RNA-sequencing experiments that include at least 3 replicates where no sample is behaving as an outlier and data is confirmed by high correlation and consistency between replicates. Such was the case here in terms of replicates there were no outliers and we had n=3 replicates for ChIP-seq and n=4 replicates for RNA-seq. Moreover, we used robust computational validation and statistical analysis (response 17).

17. In Figure S1 C: This PCA analysis looks strange: the points for both groups are perfectly symmetrical. Is this an artefact? I cannot make sense of a scenario where this would be the result.

Author response 23: The data points do look symmetrical but not perfectly symmetrical. This may be an artefact of the normalization and batch correction performed using surrogate variable analysis (*sva*) (Leek, Johnson, Parker, Jaffe, and Storey, 2012) (PMID: 22257669). The methods section describes the approach used for the normalization of the read counts.

18. Figure S1E: It is unclear from the figure and figure legend what the numbers 1-15 indicate.

Author response 24: The numbers referred to the cluster number for the gene ontology terms which is generated by the ViSEAGO package used to produce this figure (Brionne, Juanchich, and Hennequet-Antier, 2019) (PMID: 31406507). This comment has been addressed and a label has been added to show what these numbers refer to.

19. The representations of cell type abundance in the placenta are too small, in combination with very thick lines and bold fonts, they are difficult to read. (e.g. Figure 5A, 6A, S5, S7D).

Author response 25: The visualization is now improved for these figures. Fisher’s exact test to overlap paternal obesity-associated differentially enriched regions of (H3K4me3 deH3K4me3) with female and male placenta differentially enriched genes (Figure 4 —figure supplement 1 Di and ii).

Reviewer #3 (Recommendations for the authors):1) The study would be significantly strengthened by a more nuanced analysis of the data, as well as more specific follow up and functional validation of individual genes that may be responsible for placental phenotypes in offspring. For example, in situ hybridization or Western blots in placenta to confirm the expression differences detected by RNA-seq would better support the claims.

Author response 29 (see also response 2, 17 and 26): Regarding validation: We used a high standard of computational validation and visualization strategies, to ensure confidence in our genomic data analysis. This also allowed for a comprehensive understanding of the biological and physiological impacts of paternal obesity on the sperm epigenome and placenta transcriptome. In our experimental design we also included biological and technical replicates. Together these methods provide robustness checks of the experimental data and support our conclusions. These validation strategies are detailed in Author responses 2, 17 and 25.

2) Validation of deconvolution analysis, for example by sorting or immunofluorescence to experimentally validate the cell type composition of the placentas, would increase confidence in the deconvolution results.

Author response 30: The consistency of results from the deconvolution analyses across the two models (paternal obesity model versus hypoxic placenta model, i.e. preconception exposure versus direct in utero exposure), which were independently generated, substantially increase confidence in our findings. The experiments and analysis as presented provide new information on routes of paternal disease transmission intergenerationally that makes a solid advance for the field and can be built upon in future studies. There are always more experiments to do and we have planned future studies using single-cell RNA-sequencing.

3) Statistical tests (e.g. Fisher's Exact test) should be done to show the significance of overlaps when comparing gene sets.

Author response 31: This analysis has been performed and the p-values have been added on the respective figures (Figure 2 —figure supplement 1 D, Figure 4 —figure supplement 1 D, Figure 5 —figure supplement 3 C-G, Figure 6 D, Figure 6 —figure supplement 1 H).

4) Because the differences in H3K4me3 signal between conditions appear to be very modest (Figure 2B, 2F), it is not clear whether these differences are real or just due to variations between libraries. Spike-ins in the ChIP libraries would drastically improve confidence in these differences.

Author response 31. The ChIP-sequencing data has been normalized to account for library size and batch effects, allowing to detect differences in H3K4me3 signal across conditions rather than resulting from confounding differences in libraries or experiment. The Principal Component Analysis plot (Figure 2 —figure supplement 1 C) as well as the heatmap (Figure 2 A) both show separation of samples according to dietary treatment, with consistent changes across samples within a given experimental group and clustering of samples by group with hierarchical clustering (Figure 2 A). Furthermore, we find similarities in the feature characteristics of the diet-sensitive regions detected across our two studies (Pepin et al., 2022 and this study) with some overlap across datasets, suggesting the effects detected are not random.

5) In Figure S3C (PCA for placenta RNA-seq data), there does not seem to be good separation between placental gene expression in offspring of HFD compared to control sires. This should be explained.

Author response 31: As shown on the principal component analysis plot (Figure 4 —figure supplement 1 C), samples separate according to paternal dietary treatment along the X axis (PC1) which explains 11% of the variance, whereas the samples separate according to placenta sex along the y axis (PC2) which explains 9% of the variance. The differential expression analysis revealed significant changes in expression associated with paternal diet, suggesting that at least for the detected genes the variation associated with diet is greater than inter-individual variation. We do not expect extreme effects on the placenta given that we are working with a paternal preconception exposure model, as opposed to the effects that would be observed from a direct exposition such as the hypoxic placenta model dataset used in this study (section 2.5, Figure 6).

6) In Figure 2D: regions losing H3K4me3 are predominantly >5kb from the TSS. Can an explanation of what these sites are likely to correspond to be provided? Most sites of strong H3K4me3 are at a TSS.

Author response 32. Regions showing a decrease in H3K4me3 in sperm from obese mice include intergenic regions whereas those showing an increased for H3K4me3 were predominantly at promoter regions (as stated in Lines 159-162). We include analysis on functional regions with deH3K4me3 in relation to enhancers, transcription factor binding motifs and promoters (Sections 2.1 and 2.2; Figures 2-3; Figure 4 —figure supplement 1).

In addition, we previously did a full genomic analysis characterizing the sperm H3K4me3 profiles in our similar model of paternal obesity (Pepin et al., 2022) (PMID: 35183795), and in a model of folate deficiency (Ariane Lismer et al., 2021)(PMID: 33596408) and in a transgenic epigenome edited mouse model (Ariane Lismer, Siklenka, Lafleur, Dumeaux, and Kimmins, 2020) (PMID: 33068438). In these studies, we showed that H3K4me3 is present at functional genomic regions that include developmental enhancers, transposable elements and more (see our review (Ariane Lismer and Kimmins, 2023); PMID: 37059740). The data generated in this study showed a genomic distribution consistent with these previous studies and therefore we do not include repetitive findings in the current study.

Reviewer #4 (Recommendations for the authors):The authors build upon their previous work on mouse models testing the contribution of parental diet to the health and development of the offspring. Previously, the Sloboda lab had reported that HFD induced placental hypoxia while the Kimmins lab had identified genomic regions that show differentially enrichment of H3K4me3 upon HFD. With this work the authors aim to explore "the role of paternally driven gene expression in placenta" as well as the potential role of H3K4me3 in transmitting information from sperm to embryo.Firstly, the authors assessed H3K4me3 changes in sperm samples of the HFD model from the Kimmins lab. They identified DE regions of H3K4me3 between HFD and control sperm and annotated these regions based on their genomic location (promoter/enhancer). Next, the authors associated these genomic locations to genes for which they performed ontology analysis. In addition, they performed Motif enrichment analysis.Secondly, the authors performed bulk RNAseq in placentas of embryos sired either from HFD or control males mated with C57BL6/J females. They performed differential gene expression analysis and tried to associate the placenta DE genes to sperm DE H3K4me3 regions. They further performed deconvolution of the bulk RNA seq data to identify cellularity biases of the placentae and finally they compared their data to external RNA seq data from hypoxic placentae.The bioinformatic analysis is rigorous, however the design of the experiments cannot support the initial aims of the authors. For example, to identify paternally driven gene expression the authors should have used female animals on a different genetic background (other than C57BL6/J) to be able to differentiate between the parental origins of transcripts based on SNPs.The current experiment shows transcriptional differences that cannot be attributed to the parental origin. We cannot exclude the possibility of paternal expression affecting maternal expression in trans. In addition, as the authors acknowledge in the manuscript that the association of DEGs in placenta to sperm differentially H3K4 methylated regions is difficult given the high cellular heterogeneity within the tissue. Hence, the authors are recommended to isolate cells that they identified to be more affected. In addition, or alternatively, the authors could investigate trophoblast cells from blastocyst embryos that give rise to placenta. In summary, the overall design of the study is not optimal, which together with rather confusing writing does not help the field to understand the impact of paternal HFD diet on chromatin landscape in sperm and physiology in offspring.1. One major objective of the study is to relate changes in H3K4me3 occupancy levels in sperm to transcriptional changes in placenta. From the current data analyses, it is not clear at which genomic regions H3K4me3 levels have changed in a statistically significant manner between dietary conditions using the data of the replicates. When describing changes, please include fold change and adjusted p-values. Given that the authors used the csaw package to determine and normalize the H3K4me3 regions why they didn't perform the statistical significance analysis of differential enrichment using the csaw embedded functions?

Author response 34. As described in the Methods section, we used csaw (Lun and Smyth, 2016) (PMID: 26578583) to select the genomic regions with read counts of a fold change enrichment of more than 4. Using those called regions we chose the top 5% enriched regions that contribute the most to the Principal component associated with diet as shown in Figure 2 —figure supplement 1 C. This same approach was used in our folate deficient diet study (Ariane Lismer et al., 2021)(PMID: 33596408) and we consider it highly stringent as we are only selecting the top 5% of regions associated with diet by PCA analysis.

2. Related to the first comment, this reviewer does not observe any difference in H3K4me3 levels for Cbx7 nor for Igf2, while Prdx6 and Slc19a1 seems to have increased H3K4me3 levels (Figure 2F). What is the significance value for changes for these 4 genomic windows? It should be indicated in the figure. For Igf2, it is not clear whether the peaks localize at its promoter or somewhere in the gene. It appears more likely to be the latter. If so, what is the significance for such occupancy for gene regulation potential? Moreover, given the prominent role that Igf2 plays within the manuscript, have the authors observed any change in H3K4me3 at the germ line ICR for this locus, which has been shown to be critical for paternal transmission of the paternal epiallele and imprinted expression in offspring? Any change in DNA methylation? Likewise, for Kcnq1 which in placental tissues is normally maternally expressed and paternally repressed by Polycomb repression system. What about H3K4me3 levels at the Kcnq1ot1 ICR element?

Author response 35 (also response 34): We revised Figure 2 F (now Figure 2 E) to improve the visualization of the enrichment differences at the selected genes and the genomic region. We detect deH3K4me3 at Igf2 (lines 178-180) is in the list of regions showing a decrease in H3K4me3 in sperm, Kcnq1 (lines 182-184) is in the list of regions showing an increase in H3K4me3. In Figure 4 E and F the heatmaps show only deregulation of imprinted genes by sex and experimental group. There are significant differences in expression that are sex-specific in maternally and paternally imprinted genes in response to paternal obesity. We did not include DNAme analysis in this study.

3. For a histone modification to fulfill a possible role in paternal inheritance, what are the overall nucleosomal occupancy levels at the deH3K4me3 regions in sperm, compared to genome wide average levels in control and HFD samples?

Author response 36**:** The nucleosome occupancy is not expected to change – see correlation heat map Figure 2 —figure supplement 1 B that shows peaks for H3K4me3 are highly homogenous between experimental and control groups (Pearson correlation values 0.99). In our KDM1A transgenic model where there were severe epigenetic changes at over 5000 promoters in H3K4me2 and me3, we detected no differences in nucleosome occupancy (Ariane Lismer et al., 2020; Siklenka et al., 2015) (PMID: 33068438; 26449473). In the context of obesity, changes in the epigenome have been observed at various marks (here histone methylation, others found changes at DNAme and ncRNA abundance), but not nucleosome occupancy (Donkin et al., 2016) (PMID: 26669700).

4. Figure S1D/E: What is the directionality of H3K4me3 changes (increased, decreased) between the two studies?

Author response 37**:** There was similarity in directionality of deH3K4me3 between this obesity model and Pepin et al., 2022**.** Both studies revealed that the majority of obesity-associated regions show an increase in H3K4me3 enrichment (71% in this study and 86% in Pepin et al., 2022), whereas a smaller proportion of the detected regions show a decrease in enrichment (29% in this study and 14% Pepin et al., 2022).

5. The authors report that "Regions losing H3K4me3 showed moderate H3K4me3-enrichment in 173 CON sperm, with predominantly low CpG density, whereas regions gaining H3K4me3 showed low-to moderate enrichment with mainly high CpG density (Figure 2 C)". What could the reason be for the sequence specificity of such changes? Are there any changes in H3K27me3 levels at deH3K4me3? It is quite noticeable that none of the sites gaining H3K4me3 in HFD samples reach the level of H3K4me3 observed at CpG-rich regions in sperm of controls (grey dots in figure 2C). Why? Would it have any impact on possible transmission rates?

Author response 38: The DNA sequence specificity of low CpG regions that lost deH3K4me relates to the genomic functional region implicated that include those in intergenic spaces that include placenta, testes and ESC enhancers (Lines 167-169) for example. In contrast, the high CpG regions that gained H3K4me3 are mostly at promoters and contain binding sites for TF (section 2.2 and Figure 3). Figure 2 C shows the H3K4me3 signal level versus CpG density, including all regions in sperm bearing H3K4me3, and the deH3K4me3 regions labeled based on directionally change upon HFD (increased versus decreased enrichment). It appears that the regions least sensitive to paternal obesity are those with high H3K4me3 levels, potentially regions that are less dynamic and therefore less likely to be subjected to epigenetic changes upon environmental exposures. This is certainly the case for DNAme in sperm where we report in several studies the regions bearing intermediate levels of methylation are the most responsive to environmental exposures (Chan et al., 2019; A Lismer et al., 2022) (PMID: 31393794). We speculate that these lower methylated regions are similar for H3K4me3 because they may be subject to more transcriptional turnover for example which is linked to re-establishment of H3K4me. We did not perform ChIP-seq targeting H3K27me3 in this study.

6. To be able to interpret / "normalize" the data presented in panel 2D, please provide the distribution of windows relative to TSS according to their presence in genome.

Author response 39: We performed this analysis in our previous article (Pepin et al., 2022) (PMID: 35183795). Because the results were consistent and similar with the datasets generated from the current study, we therefore decided not to include this information in this article to avoid showing repetitive findings in both studies. See Pepin et al., 2022 (Figure S5) (Pepin et al., 2022) (PMID: 35183795).

7. Figure 2E: It seems as if the GO-term for regions with decreased H3K4me3 levels serve more likely functions in placenta than those with increased levels.

Author response 40. This is true – see lines 170-184. We removed Figure 2 E as it showed only 6 enriched pathways which are detailed in the manuscript.

8. Regarding enhancers, the current data presentation is rather superficial. To substantiate some of these findings, to what extent do the changes in H3K4me3 as monitored in sperm reflect changes in expression in testicular cells of HFD exposed males? Are any of the three genes highlighted (Tmem174, Plag1, Pdgfb) differentially expressed in placentas of male and female embryos?

Author response 41: These genes did not appear in our DER list. However, it is important to keep in mind that we are looking at one developmental time point and the effect of sperm deH3K4me3 at enhancers for these genes could be very early in placenta development, or not at all. It is not reasonable to expect that every deH3K4me3 will impact gene expression in the placenta. For example, many of the deH3K4me3 could impact the embryo and not the placenta. Some deH3K4me in sperm may not affect the expression of the corresponding gene at all. We did not examine testis germ cells in this study.

9. Discovery of TF motif: when searching for TF motif enrichments in genomic windows, have the authors taken into account that the GC percentage underlying these windows is very different between regions with decreased and increased H3K4me3 levels? This has a major impact on the obtained results.

Author response 42**:** The HOMER software takes into account GC-content distribution (Heinz et al., 2010) (PMID: 20513432). Indeed, regions showing increased versus decreased H3K4me3 levels are likely to be enriched for different TF motifs, for that reason we have performed the TF motif enrichment analysis separately for these two sets of regions (Figure 3).

10. The authors state that previous reports indicated changes in DNA methylation after HFD exposure (line 217). Is there any link between DNA methylation and H3K4me3 alterations?

Author response 43: We have updated the text in order to answer this question and include this information in the discussion.

Lines 388-401: “Interestingly, changes in sperm DNA methylation upon HFD feeding has been previously reported, and ETS motifs (enriched in obesity-associated deH3K4me3 in sperm) have been found to be DNA-methylation sensitive, including in spermatogonial stem cells (Domcke et al., 2015; Dura et al., 2022; Lea et al., 2018; Yin et al., 2017). Given the interplay between H3K4 and DNA methylation, it is conceivable that HFD-induced epimutations at either of these marks could influence one another (reviewed in Janssen and Lorincz, 2022), and these could in turn alter TF functions. This phenomenon has been described in a mouse model of paternal low-protein diet, where oxidative stress-induced phosphorylation of the *Atf7* TF was suggested to impede its DNA-binding affinity in germ cells, leading to a decrease in H3K9me2 at target regions (Yoshida et al., 2020). As in the low-protein diet model, oxidative stress is a hallmark of obesity and increased levels of reactive oxygen species have been observed in testes of diet-induced obesity mouse models and linked to impaired embryonic development (T. Fullston et al., 2012; Lane et al., 2014; Mitchell et al., 2011). These findings provide avenues for further investigation such as whether epigenetic changes on paternal alleles may impact TF binding during early embryogenesis.”

11. "In response to paternal obesity, we detected 2,035 and 2,365 differentially expressed genes (DEGs) in female and male placentas, respectively (Figure 4 A-B)." How many DEGs are detected if authors compare the male control to female control placentas. This will help to understand which is ground truth of sex specific differences of placentas. How many are UP versus DOWN regulated in the respective sexes? How have the authors harvested the placentas? Which parts of placenta and decidua was included in the RNA-sample prep.?

Author response 44: The uterus was dissected from the peritoneal cavity; vertical cuts were made across the short axis of the uterine horn to isolate individual implantation sites. In each individual implantation site (containing placenta and fetus), a cut was made longitudinally along the antimesometrial side of the uterus to expose the amniotic sac containing the fetus. The chorion, amnion, umbilical cords, and fetus were removed from the placenta. The uterus was peeled away from the placenta, separating the decidua from the placenta. Whole placental homogenates containing both placental zones were used in the RNA sequencing.

We did a gene-level analysis at single-transcript resolution using the Lancaster method (Yi, Pimentel, Bray, and Pachter, 2018) (PMID: 29650040), where differential expression is performed on transcripts, then transcript p values are aggregated generating gene-level p values. By using only the p-values from the differential analysis, the aggregation prevents issues that can arise from variance in directionality. Therefore, we cannot state which genes are up-regulated versus which genes are down-regulated since we select the genes based on their transcriptional changes.

12. When reviewing the heatmaps of imprinted genes in females, 1 out of 4 HFD placenta clusters with controls; for males, only 2 HFD placentas show altered gene expression. How do these changes relate to the histological changes? Why is there variation at all? How relevant would the DEGs be for phenotypic changes? Could imprinted expression serve to classify the severity of the perturbation in embryos/placenta? Histological data would be needed for benchmarking.

Author response 45. We did a gene-level analysis at transcript resolution (Lancaster method, See author response 44), and the heatmaps in Figure 4 A and B show the *transcripts* that code for the detected differentially expressed genes – that is the strength of the analysis because it is sensitive to these transcript-level changes which clusters perfectly here (Yi, Pimentel, Bray, and Pachter, 2018). In Figure 4 E and F we show the expression levels of the *genes* and with this visualization we list these imprinted genes. These heatmaps show the *gene* signal rather than the transcript signal so that is why the clustering differs, but they are still being detected as differentially expressed in the transcript-level analysis. When one shows the gene signal rather than the transcript signal, the individual transcriptional changes can look like they are "cancelled out" as they are aggregated into a gene-level signal. For these reasons, using the Lancaster method avoids masking transcript-level dynamics (Yi, Pimentel, Bray, and Pachter, 2018).

Although imprinted genes have been implicated in placental defects and imprinted disorders, we believe it is too early at this point to conclude that these targets could be used to classify placentas and embryos phenotypic severity. More research is required to establish reliable markers that can predict placental integrity and potentially long-term metabolic health.

*13. Regarding imprinted genes, which ones are commonly mis-expressed in male and female placenta (e.g. Runx1, Ano1) and which are sex specific? Why? How have the imprinted genes been chosen? Please indicate and discuss which are controlled by germline derived DNA methylation and which by maternal H3K27me3.*

Author response 46. To assess whether there are differentially expressed imprinted genes in placentas, we overlapped the lists of DEGs from males and females with a list of previously identified imprinted genes (Tucci et al., 2019) (PMID: 30794780). Therefore, the differentially expressed imprinted genes shown on the heatmaps in Figure 4 E-F are those overlapping between these two lists. There are 7 imprinted genes differentially expressed in both male and female placentas (Tnfrsf23, Ano1, Htra3, Runx1, Bag3, Bbx, Cdkn1c), while the remaining genes are sex-specific. We have updated Figure 4 E-F to highlight the imprinted genes that are deregulated in both male and female placentas, by denoting these genes with asterisks. As for the genes that are controlled by germline-derived DNAme versus maternal H3K27me3, we did not assess this and we do not see the relevance of discussing this point for the purpose of the study.

14. "To assess the link between sperm H3K4me3 and the placental transcriptome, we overlapped deH3K4me3 at promoters (n=508) with DEGs in the placenta, and identified 45 and 48 DEGs in female and male placentas, respectively (Figure S3 D ii-iii)." Please specify how many UP-regulated and DOWN-regulated. The authors do not dwell further on this finding. What was the fold change in expression?

Author response 46 (also response 44)**:** To detect DEGs, we used a gene-level analysis approach at single transcript resolution (the Lancaster method (Yi et al., 2018) (PMID: 29650040)). This approach involves a differential analysis on transcript counts, followed by aggregation of p-values from individual transcript to obtain a list of differentially expressed genes based on changes at the transcript level. This approach allows to consider transcript-level dynamics and therefore increases sensitivity and accuracy. Because multiple transcripts that code for the same gene can show varying directionality changes (i.e. some transcripts show increased expression whereas other transcripts that code for the same gene show decreased expression), and because the detected DEGs are based on changes in transcript expression, it is not possible to show which genes are *up-regulated* or *down-regulated*, and therefore can only describe *de-regulated* genes.

15. "Of note, although a significant number of DEGs overlapped between female and male placentas (n=359, Fisher's exact test P=1.5e-19; Figure S3 D i), 82% of female DEGs and 85% of male DEGs were uniquely de-regulated, indicating sex-specific placental responses to paternal obesity."Similarly, the overlap between the H3K4me3 peaks when comparing the 2 HFD models is less than 10%. The authors present this overlap as great similarity without acknowledging protocol specific responses. "Despite differences in experimental design and animal models, we found a significant overlap in regions showing differential H3K4me3 (deH3K4me3) from both studies (128 overlapping regions, Fisher's exact test P=2.2e-16, Figure S1 D)." Please discuss in a more balanced manner these results.

Author response 46. We have adapted the text to better balance the discussion of these findings.

Lines 230-236: “Although a significant number of DEGs overlapped between female and male placentas (n=359, Fisher’s exact test P=1.5e-19; Figure 4 —figure supplement 1 D i), 82% of female DEGs and 85% of male DEGs were uniquely de-regulated in response to paternal obesity. These findings may reflect the previously observed sex-specific effects of paternal factors on offspring metabolism (Binder, Beard, et al., 2015; Claycombe-Larson et al., 2020; Glavas et al., 2021; Jazwiec et al., 2022; Pepin et al., 2022). This suggests some of the sexually dimorphic responses may originate *in utero* due to differences in placental development and function.”

Lines 157-161: “Of the top 5% and top 10% differentially enriched H3K4me3 (deH3K4me3) regions, some were common to those previously identified in Pepin et al., (2022) (128 and 423 overlapping deH3K4me3 regions, Fisher’s exact test P=2.2e-16 and P=2.2e-16, Figure 2 —figure supplement 1 D i and ii, respectively). This was despite substantial differences in animal models (timing of diet exposure [3 vs 6 weeks of age], control diet [chow vs low-fat diet], and mouse substrain [C57BL/6J vs C57BL/6NCrl]).”

16. The authors should perform a histological analysis of the placenta or use the previously published histological analysis to support/or reject the deconvolution method of the bulk RNAseq data.

Author response 47. Thank you for this comment and we apologize for not being clearer regarding our previously published observations. We have previously performed extensive histological analyses at embryonic day 14.5 and 18.5 in Jazwiec et al. 2022 and found that male diet-induced obesity results in significant changes in transcription factor that regulate blood vessel development, blood vessel integrity and signalling pathways governed by hypoxia (HIF1A, VEGF and VEGR). We found that paternal obesity did not appreciably impact fetal or placental growth, placental nutrient transporter transcripts, placental endocrine-related cell number (giant cells for placental lactogen), or cellular stress related signalling transcripts and proteins (ER stress pathways) (Jazwiec et al., 2022); (PMID: 35377412). Future studies should perform single-cell RNA sequencing on placentas derived from control- versus obese-sires in order to confirm changes in cell-type proportions within these placentas, as well as to identify cell-type-specific changes in gene expression. Furthermore, these methods should be applied to earlier time points in development such as the trophectoderm (blastocyst stage) in order to assess when these transcriptional defects originate during placental development.

17. Next, it is not clear to this reviewer how the authors can assign nor normalize expression of genes that do not belong to specific cell types to be differentially expressed between experimental conditions. The meaning of any of these DEGs results can not be evaluated without any detailed follow-up analysis.

Author response 48: A deconvolution analysis allows to infer cell-type proportions from a bulk RNA-seq data set based on a reference single-cell RNA-seq dataset previously annotated (Aliee and Theis, 2021; Kuhn et al., 2011) (PMID: 34293324; 21983921). To account for cellular composition within a tissue for bulk RNA-sequencing differential gene expression analysis, the inferred cell type proportions can be accounted for in the differential analysis, as described in (Campbell et al., 2023) (PMID: 36914823). To do so, a principal component analysis is first applied to the deconvoluted proportion values, and the principal components are added as covariates into the model (see Methods). This approach allows to limit the number of covariates in the model and consider the dependence between the individual cell type proportions. We carefully chose available single-cell RNA-sequencing data from mouse placenta that matched for the same mouse strain as well as developmental time point as that of our samples to ensure the data could be comparable. We would also like to remind this reviewer of the current state of the field: this is the first placenta RNA-seq dataset generated from a paternal obesity model. Therefore, prior to the current study, there was no rationale to perform placenta single-cell RNA-sequencing in such model. While generating single-cell RNA-sequencing data from samples from our HFD model was out of the scope of this study, future work should focus on performing such experiments to be able to directly assess changes in cell-type proportions and delineate the cell-type-specific changes in gene expression. The datasets generated in the current study are now publicly available and can be used to make novel findings and move the field forwar

References

Aliee, H., and Theis, F. J. (2021). AutoGeneS: Automatic gene selection using multi-objective optimization for RNA-seq deconvolution. *Cell Systems*, *12*(7), 706-715.e4. https://doi.org/https://doi.org/10.1016/j.cels.2021.05.006

Andescavage, N. N., and Limperopoulos, C. (2021). Placental abnormalities in congenital heart disease. *Translational Pediatrics*, *10*(8), 2148–2156. https://doi.org/10.21037/tp-20-347

Aneman, I., Pienaar, D., Suvakov, S., Simic, T. P., Garovic, V. D., and McClements, L. (2020). Mechanisms of Key Innate Immune Cells in Early- and Late-Onset Preeclampsia. *Frontiers in Immunology*, *11*, 1864. https://doi.org/10.3389/fimmu.2020.01864

Bachmayer, N., Rafik Hamad, R., Liszka, L., Bremme, K., and Sverremark-Ekström, E. (2006). Aberrant uterine natural killer (NK)-cell expression and altered placental and serum levels of the NK-cell promoting cytokine interleukin-12 in pre-eclampsia. *American Journal of Reproductive Immunology (New York, N.Y. : 1989)*, *56*(5–6), 292–301. https://doi.org/10.1111/j.1600-0897.2006.00429.x

Barton, S. C., Adams, C. A., Norris, M. L., and Surani, M. A. H. (1985). Development of gynogenetic and parthenogenetic inner cell mass and trophectoderm tissues in reconstituted blastocysts in the mouse. *Journal of Embryology and Experimental Morphology*, *VOL. 90*, 267–285.

Barton, Sheila C., Surani, M. A. H., and Norris, M. L. (1984). Role of paternal and maternal genomes in mouse development. *Nature*, *311*(5984), 374–376. https://doi.org/10.1038/311374a0

Binder, N. K., Beard, S. A., Kaitu’U-Lino, T. J., Tong, S., Hannan, N. J., and Gardner, D. K. (2015). Paternal obesity in a rodent model affects placental gene expression in a sex-specific manner. *Reproduction*, *149*(5), 435–444. https://doi.org/10.1530/REP-14-0676

Binder, N. K., Hannan, N. J., and Gardner, D. K. (2012). Paternal Diet-Induced Obesity Retards Early Mouse Embryo Development, Mitochondrial Activity and Pregnancy Health. *PLoS ONE*, *7*(12). https://doi.org/10.1371/journal.pone.0052304

Binder, N. K., Sheedy, J. R., Hannan, N. J., and Gardner, D. K. (2015). Male obesity is associated with changed spermatozoa Cox4i1 mRNA level and altered seminal vesicle fluid composition in a mouse model. *Molecular Human Reproduction*, *21*(5), 424–434. https://doi.org/10.1093/molehr/gav010

Brionne, A., Juanchich, A., and Hennequet-Antier, C. (2019). ViSEAGO: A Bioconductor package for clustering biological functions using Gene Ontology and semantic similarity. *BioData Mining*, *12*(1), 1–13. https://doi.org/10.1186/s13040-019-0204-1

Campbell, K. A., Colacino, J. A., Puttabyatappa, M., Dou, J. F., Elkin, E. R., Hammoud, S. S., … Bakulski, K. M. (2023). Placental cell type deconvolution reveals that cell proportions drive preeclampsia gene expression differences. *Communications Biology*, *6*(1), 264. https://doi.org/10.1038/s42003-023-04623-6

Chambers, T. J. G., Morgan, M. D., Heger, A. H., Sharpe, R. M., and Drake, A. J. (2016). High-fat diet disrupts metabolism in two generations of rats in a parent-of-origin specific manner. *Scientific Reports*, *6*, 1–11. https://doi.org/10.1038/srep31857

Chan, D., Shao, X., Dumargne, M. C., Aarabi, M., Simon, M. M., Kwan, T., … Trasler, J. M. (2019). Customized methylc-capture sequencing to evaluate variation in the human sperm DNA methylome representative of altered folate metabolism. *Environmental Health Perspectives*, *127*(8), 87002. https://doi.org/10.1289/EHP4812

Chu, A., Casero, D., Thamotharan, S., Wadehra, M., Cosi, A., and Devaskar, S. U. (2019). The Placental Transcriptome in Late Gestational Hypoxia Resulting in Murine Intrauterine Growth Restriction Parallels Increased Risk of Adult Cardiometabolic Disease. *Scientific Reports*, *9*(1), 1243. https://doi.org/10.1038/s41598-018-37627-y

Claycombe-Larson, K. G., Bundy, A. N., and Roemmich, J. N. (2020). Paternal high-fat diet and exercise regulate sperm miRNA and histone methylation to modify placental inflammation, nutrient transporter mRNA expression and fetal weight in a sex-dependent manner. *Journal of Nutritional Biochemistry*, *81*, 108373. https://doi.org/10.1016/j.jnutbio.2020.108373

Cropley, J. E., Eaton, S. A., Aiken, A., Young, P. E., Giannoulatou, E., Ho, J. W. K., … Suter, C. M. (2016). Male-lineage transmission of an acquired metabolic phenotype induced by grand-paternal obesity. *Molecular Metabolism*, *5*(8), 699–708. https://doi.org/10.1016/j.molmet.2016.06.008

de Castro Barbosa, T., Ingerslev, L. R., Alm, P. S., Versteyhe, S., Massart, J., Rasmussen, M., … Barrès, R. (2016a). High-fat diet reprograms the epigenome of rat spermatozoa and transgenerationally affects metabolism of the offspring. *Molecular Metabolism*, *5*(3), 184–197. https://doi.org/10.1016/j.molmet.2015.12.002

de Castro Barbosa, T., Ingerslev, L. R., Alm, P. S., Versteyhe, S., Massart, J., Rasmussen, M., … Barrès, R. (2016b). High-fat diet reprograms the epigenome of rat spermatozoa and transgenerationally affects metabolism of the offspring. *Molecular Metabolism*, *5*(3), 184–197. https://doi.org/10.1016/j.molmet.2015.12.002

Denomme, M. M., Parks, J. C., McCallie, B. R., McCubbin, N. I., Schoolcraft, W. B., and Katz-Jaffe, M. G. (2020). Advanced paternal age directly impacts mouse embryonic placental imprinting. *PloS One*, *15*(3), e0229904. https://doi.org/10.1371/journal.pone.0229904

Ding, T., Mokshagundam, S., Rinaudo, P. F., Osteen, K. G., and Bruner-Tran, K. L. (2018). Paternal developmental toxicant exposure is associated with epigenetic modulation of sperm and placental Pgr and Igf2 in a mouse model. *Biology of Reproduction*, *99*(4), 864–876. https://doi.org/10.1093/biolre/ioy111

Domcke, S., Bardet, A. F., Adrian Ginno, P., Hartl, D., Burger, L., and Schübeler, D. (2015). Competition between DNA methylation and transcription factors determines binding of NRF1. *Nature*, *528*(7583), 575–579. https://doi.org/10.1038/nature16462

Donkin, I., Versteyhe, S., Ingerslev, L. R., Qian, K., Mechta, M., Nordkap, L., … Barrès, R. (2016). Obesity and bariatric surgery drive epigenetic variation of spermatozoa in humans. *Cell Metabolism*, *23*(2), 369–378. https://doi.org/10.1016/j.cmet.2015.11.004

Du, M., Wang, W., Huang, L., Guan, X., Lin, W., Yao, J., and Li, L. (2022). Natural killer cells in the pathogenesis of preeclampsia: a double-edged sword. *The Journal of Maternal-Fetal and Neonatal Medicine : The Official Journal of the European Association of Perinatal Medicine, the Federation of Asia and Oceania Perinatal Societies, the International Society of Perinatal Obstetricians*, *35*(6), 1028–1035. https://doi.org/10.1080/14767058.2020.1740675

Dura, M., Teissandier, A., Armand, M., Barau, J., Lapoujade, C., Fouchet, P., … Bourc’his, D. (2022). DNMT3A-dependent DNA methylation is required for spermatogonial stem cells to commit to spermatogenesis. *Nature Genetics*, *54*(4), 469–480. https://doi.org/10.1038/s41588-022-01040-z

Fullston, T., Palmer, N. O., Owens, J. A., Mitchell, M., Bakos, H. W., and Lane, M. (2012). Diet-induced paternal obesity in the absence of diabetes diminishes the reproductive health of two subsequent generations of mice. *Human Reproduction*, *27*(5), 1391–1400. https://doi.org/10.1093/humrep/des030

Fullston, Tod, Teague, E. M. C. O., Palmer, N. O., Deblasio, M. J., Mitchell, M., Corbett, M., … Lane, M. (2013). Paternal obesity initiates metabolic disturbances in two generations of mice with incomplete penetrance to the F2 generation and alters the transcriptional profile of testis and sperm microRNA content. *FASEB Journal*, *27*(10), 4226–4243. https://doi.org/10.1096/fj.12-224048

Ghanayem, B. I., Bai, R., Kissling, G. E., Travlos, G., and Hoffler, U. (2010). Diet-induced obesity in male mice is associated with reduced fertility and potentiation of acrylamide-induced reproductive toxicity. *Biology of Reproduction*, *82*(1), 96–104. https://doi.org/10.1095/biolreprod.109.078915

Glavas, M. M., Lee, A. Y., Miao, I., Yang, F., Mojibian, M., O’Dwyer, S. M., and Kieffer, T. J. (2021). Developmental Timing of High-Fat Diet Exposure Impacts Glucose Homeostasis in Mice in a Sex-Specific Manner. *Diabetes*, *70*(12), 2771–2784. https://doi.org/10.2337/db21-0310

Grandjean, V., Fourré, S., De Abreu, D. A. F., Derieppe, M. A., Remy, J. J., and Rassoulzadegan, M. (2015). RNA-mediated paternal heredity of diet-induced obesity and metabolic disorders. *Scientific Reports*, *5*(June), 1–9. https://doi.org/10.1038/srep18193

Han, X., Wang, R., Zhou, Y., Fei, L., Sun, H., Lai, S., … Guo, G. (2018). Mapping the Mouse Cell Atlas by Microwell-Seq. *Cell*, *172*(5), 1091-1107.e17. https://doi.org/10.1016/j.cell.2018.02.001

Heinz, S., Benner, C., Spann, N., Bertolino, E., Lin, Y. C., Laslo, P., … Glass, C. K. (2010). Simple combinations of lineage-determining transcription factors prime cis-regulatory elements required for macrophage and B cell identities. *Molecular Cell*, *38*(4), 576–589. https://doi.org/10.1016/j.molcel.2010.05.004

Hemberger, M., Hanna, C. W., and Dean, W. (2020). Mechanisms of early placental development in mouse and humans. *Nature Reviews Genetics*, *21*(1), 27–43. https://doi.org/10.1038/s41576-019-0169-4

Hermann, B. P., Cheng, K., Singh, A., Roa-De La Cruz, L., Mutoji, K. N., Chen, I.-C., … McCarrey, J. R. (2018). The Mammalian Spermatogenesis Single-Cell Transcriptome, from Spermatogonial Stem Cells to Spermatids. *Cell Reports*, *25*(6), 1650-1667.e8. https://doi.org/10.1016/j.celrep.2018.10.026

Hu, S., Wang, L., Yang, D., Li, L., Togo, J., Wu, Y., … Speakman, J. R. (2018). Dietary Fat, but Not Protein or Carbohydrate, Regulates Energy Intake and Causes Adiposity in Mice. *Cell Metabolism*, 415–431. https://doi.org/10.1016/j.cmet.2018.06.010

Hull, R. L., Willard, J. R., Struck, M. D., Barrow, B. M., Brar, G. S., Andrikopoulos, S., and Zraika, S. (2017). High fat feeding unmasks variable insulin responses in male C57BL/6 mouse substrains. *Journal of Endocrinology*, *233*(1), 53–64. https://doi.org/10.1530/JOE-16-0377

Huypens, P., Sass, S., Wu, M., Dyckhoff, D., Tschöp, M., Theis, F., … Beckers, J. (2016). Epigenetic germline inheritance of diet-induced obesity and insulin resistance. *Nature Genetics*, *48*(5), 497–499. https://doi.org/10.1038/ng.3527

Janssen, S. M., and Lorincz, M. C. (2022). Interplay between chromatin marks in development and disease. *Nature Reviews Genetics*, *23*(3), 137–153. https://doi.org/10.1038/s41576-021-00416-x

Jazwiec, P. A., Patterson, V. S., Ribeiro, T. A., Yeo, E., Kennedy, K. M., Mathias, P. C. F., … Sloboda, D. M. (2022). Paternal obesity induces placental hypoxia and sex-specific impairments in placental vascularization and offspring metabolism†. *Biology of Reproduction*, ioac066. https://doi.org/10.1093/biolre/ioac066

Kuhn, A., Thu, D., Waldvogel, H. J., Faull, R. L. M., and Luthi-Carter, R. (2011). Population-specific expression analysis (PSEA) reveals molecular changes in diseased brain. *Nature Methods*, *8*(11), 945–947. https://doi.org/10.1038/nmeth.1710

Lambrot, R., Xu, C., Saint-Phar, S., Chountalos, G., Cohen, T., Paquet, M., … Kimmins, S. (2013). Low paternal dietary folate alters the mouse sperm epigenome and is associated with negative pregnancy outcomes. *Nature Communications*, *4*. https://doi.org/10.1038/ncomms3889

Lane, M., Mcpherson, N. O., Fullston, T., Spillane, M., Sandeman, L., Kang, X., and Zander-fox, D. L. (2014). Oxidative Stress in Mouse Sperm Impairs Embryo Development , Fetal Growth and Alters Adiposity and Glucose Regulation in Female Offspring. *PLoS ONE*, *9*(7), 1–9. https://doi.org/10.1371/journal.pone.0100832

Lea, A. J., Vockley, C. M., Johnston, R. A., Del Carpio, C. A., Barreiro, L. B., Reddy, T. E., and Tung, J. (2018). Genome-wide quantification of the effects of DNA methylation on human gene regulation. *ELife*, *7*, e37513. https://doi.org/10.7554/*eLife*.37513

Leek, J. T., Johnson, W. E., Parker, H. S., Jaffe, A. E., and Storey, J. D. (2012). The sva package for removing batch effects and other unwanted variation in high-throughput experiments. *Bioinformatics (Oxford, England)*, *28*(6), 882–883. https://doi.org/10.1093/bioinformatics/bts034

Lin, J., Gu, W., and Huang, H. (2022). Effects of Paternal Obesity on Fetal Development and Pregnancy Complications: A Prospective Clinical Cohort Study. *Frontiers in Endocrinology*, *13*. https://doi.org/10.3389/fendo.2022.826665

Lismer, A, Shao, X., Dumargne, M. C., Lafleur, C., Lambrot, R., Chan, D., … Kimmins, S. (2022). Exposure of Greenlandic Inuit and South African VhaVenda men to the persistent DDT metabolite is associated with an altered sperm epigenome at regions implicated in paternal epigenetic transmission and developmental disease – a cross-sectional study. *BioRxiv*, 2022.08.15.504029. https://doi.org/10.1101/2022.08.15.504029

Lismer, Ariane, Dumeaux, V., Lafleur, C., Lambrot, R., Brind’Amour, J., Lorincz, M. C., and Kimmins, S. (2021). Histone H3 lysine 4 trimethylation in sperm is transmitted to the embryo and associated with diet-induced phenotypes in the offspring. *Developmental Cell*, *56*(5), 671-686.e6. https://doi.org/10.1016/j.devcel.2021.01.014

Lismer, Ariane, and Kimmins, S. (2023). Emerging evidence that the mammalian sperm epigenome serves as a template for embryo development. *Nature Communications*, *14*(1), 2142. https://doi.org/10.1038/s41467-023-37820-2

Lismer, Ariane, Siklenka, K., Lafleur, C., Dumeaux, V., and Kimmins, S. (2020). Sperm histone H3 lysine 4 trimethylation is altered in a genetic mouse model of transgenerational epigenetic inheritance. *Nucleic Acids Research*, *48*(20), 11380–11393. https://doi.org/10.1093/nar/gkaa712

Lun, A. T. L., and Smyth, G. K. (2016). csaw: a Bioconductor package for differential binding analysis of ChIP-seq data using sliding windows. *Nucleic Acids Research*, *44*(5), e45. https://doi.org/10.1093/nar/gkv1191

McGrath, J., and Solter, D. (1986). Nucleocytoplasmic interactions in the mouse embryo. *Journal of Embryology and Experimental Morphology*, *97*(SUPPL.), 277–290.

McGrath, James, and Solter, D. (1984). Completion of mouse embryogenesis requires both the maternal and paternal genomes. *Cell*, *37*(1), 179–183. https://doi.org/10.1016/0092-8674(84)90313-1

McNamara, G. I., Creeth, H. D. J., Harrison, D. J., Tansey, K. E., Andrews, R. M., Isles, A. R., and John, R. M. (2018). Loss of offspring Peg3 reduces neonatal ultrasonic vocalizations and increases maternal anxiety in wild-type mothers. *Human Molecular Genetics*, *27*(3), 440–450. https://doi.org/10.1093/hmg/ddx412

McPherson, N. O., Bell, V. G., Zander-Fox, D. L., Fullston, T., Wu, L. L., Robker, R. L., and Lane, M. (2015). When two obese parents are worse than one! Impacts on embryo and fetal development. *American Journal of Physiology-Endocrinology and Metabolism*, *309*(6), E568–E581. https://doi.org/10.1152/ajpendo.00230.2015

Mekada, K., Abe, K., Murakami, A., Nakamura, S., Nakata, H., Moriwaki, K., … Yoshiki, A. (2009). Genetic Differences among C57BL/6 Substrains. *Exp. Anim*, *58*(2), 141–149. https://doi.org/10.1538/expanim.58.141

Mekada, K., and Yoshiki, A. (2021). Substrains matter in phenotyping of C57BL/6 mice. *Experimental Animals*, *70*(2), 145–160. https://doi.org/10.1538/expanim.20-0158

Milosevic-Stevanovic, J., Krstic, M., Radovic-Janosevic, D., Popovic, J., Tasic, M., and Stojnev, S. (2016). Number of decidual natural killer cells and macrophages in pre-eclampsia. *The Indian Journal of Medical Research*, *144*(6), 823–830. https://doi.org/10.4103/ijmr.IJMR_776_15

Mitchell, M., Bakos, H. W., and Lane, M. (2011). Paternal diet-induced obesity impairs embryo development and implantation in the mouse. *Fertility and Sterility*, *95*(4), 1349–1353. https://doi.org/10.1016/j.fertnstert.2010.09.038

Mitchell, M., Strick, R., Strissel, P. L., Dittrich, R., McPherson, N. O., Lane, M., … El Hajj, N. (2017). Gene expression and epigenetic aberrations in F1-placentas fathered by obese males. *Molecular Reproduction and Development*, *84*(4), 316–328. https://doi.org/10.1002/mrd.22784

Murugappan, G., Li, S., Leonard, S. A., Winn, V. D., Druzin, M. L., and Eisenberg, M. L. (2021). Association of preconception paternal health and adverse maternal outcomes among healthy mothers. *American Journal of Obstetrics and Gynecology MFM*, *3*(5). https://doi.org/10.1016/j.ajogmf.2021.100384

Naismith, K., and Cox, B. (2021). Human placental gene sets improve analysis of placental pathologies and link trophoblast and cancer invasion genes. *Placenta*, *112*, 9–15. https://doi.org/https://doi.org/10.1016/j.placenta.2021.06.011

Nativio, R., Sparago, A., Ito, Y., Weksberg, R., Riccio, A., and Murrell, A. (2011). Disruption of genomic neighbourhood at the imprinted IGF2-H19 locus in Beckwith-Wiedemann syndrome and Silver-Russell syndrome. *Human Molecular Genetics*, *20*(7), 1363–1374. https://doi.org/10.1093/hmg/ddr018

Ng, S.-F., Lin, R. C. Y., Laybutt, D. R., Barres, R., Owens, J. A., and Morris, M. J. (2010). Chronic high-fat diet in fathers programs β-cell dysfunction in female rat offspring. *Nature*, *467*(7318), 963–966. https://doi.org/10.1038/nature09491

Parrettini, S., Caroli, A., and Torlone, E. (2020). Nutrition and Metabolic Adaptations in Physiological and Complicated Pregnancy: Focus on Obesity and Gestational Diabetes. *Frontiers in Endocrinology*, *11*. https://doi.org/10.3389/fendo.2020.611929

Pepin, A.-S., Lafleur, C., Lambrot, R., Dumeaux, V., and Kimmins, S. (2022). Sperm histone H3 lysine 4 tri-methylation serves as a metabolic sensor of paternal obesity and is associated with the inheritance of metabolic dysfunction. *Molecular Metabolism*, *59*, 101463. https://doi.org/https://doi.org/10.1016/j.molmet.2022.101463

Perez-Garcia, V., Fineberg, E., Wilson, R., Murray, A., Mazzeo, C. I., Tudor, C., … Hemberger, M. (2018). Placentation defects are highly prevalent in embryonic lethal mouse mutants. *Nature*, *555*(7697), 463–468. https://doi.org/10.1038/nature26002

Prasad, M. S., Charney, R. M., and García-Castro, M. I. (2019). Specification and formation of the neural crest: Perspectives on lineage segregation. *Genesis (New York, N.Y. : 2000)*, *57*(1), e23276. https://doi.org/10.1002/dvg.23276

Rosenfeld, C. S. (2021). The placenta-brain-axis. *Journal of Neuroscience Research*, *99*(1), 271–283. https://doi.org/10.1002/jnr.24603

Schjenken, J. E., Moldenhauer, L. M., Sharkey, D. J., Chan, H. Y., Chin, P. Y., Fullston, T., … Robertson, S. A. (2021). High-fat Diet Alters Male Seminal Plasma Composition to Impair Female Immune Adaptation for Pregnancy in Mice. *Endocrinology*, *162*(10). https://doi.org/10.1210/endocr/bqab123

Siersbæk, M. S., Ditzel, N., Hejbøl, E. K., Præstholm, S. M., Markussen, L. K., Avolio, F., … Grøntved, L. (2020). C57BL/6J substrain differences in response to high-fat diet intervention. *Scientific Reports*, *10*(1), 14052. https://doi.org/10.1038/s41598-020-70765-w

Siklenka, K., Erkek, S., Godmann, M., Lambrot, R., McGraw, S., Lafleur, C., … Kimmins, S. (2015). Disruption of histone methylation in developing sperm impairs offspring health transgenerationally. *Science*, *350*(6261). https://doi.org/10.1126/science.aab2006

Surani, M. A. H., Barton, S. C., and Norris, M. L. (1984). Development of reconstituted mouse eggs suggests imprinting of the genome during gametogenesis. *Nature*, *308*(5959), 548–550. https://doi.org/10.1038/308548a0

Takahashi, Y., Morales Valencia, M., Yu, Y., Ouchi, Y., Takahashi, K., Shokhirev, M. N., … Izpisua Belmonte, J. C. (2023). Transgenerational inheritance of acquired epigenetic signatures at CpG islands in mice. *Cell*. https://doi.org/10.1016/j.cell.2022.12.047

Tang, H., Pan, L., Xiong, Y., Wang, L., Cui, Y., Liu, J., and Tang, L. (2021). Down‐regulation of the Sp1 transcription factor by an increase of microRNA-4497 in human placenta is associated with early recurrent miscarriage. *Reproductive Biology and Endocrinology*, *19*(1), 21. https://doi.org/10.1186/s12958-021-00701-8

Terashima, M., Barbour, S., Ren, J., Yu, W., Han, Y., and Muegge, K. (2015). Effect of high fat diet on paternal sperm histone distribution and male offspring liver gene expression. *Epigenetics*, *10*(9), 861–871. https://doi.org/10.1080/15592294.2015.1075691

Thornburg, K. L., Kolahi, K., Pierce, M., Valent, A., Drake, R., and Louey, S. (2016). Biological features of placental programming. *Placenta*, *48 Suppl 1*(Suppl 1), S47–S53. https://doi.org/10.1016/j.placenta.2016.10.012

Thornburg, K. L., and Marshall, N. (2015). The placenta is the center of the chronic disease universe. *American Journal of Obstetrics and Gynecology*, *213*(4 Suppl), S14–S20. https://doi.org/10.1016/j.ajog.2015.08.030

Tucci, V., Isles, A. R., Kelsey, G., Ferguson-Smith, A. C., Tucci, V., Bartolomei, M. S., … Ferguson-Smith, A. C. (2019). Genomic Imprinting and Physiological Processes in Mammals. *Cell*, *176*(5), 952–965. https://doi.org/https://doi.org/10.1016/j.cell.2019.01.043

Ueda, M., Tsuchiya, K. J., Yaguchi, C., Furuta-Isomura, N., Horikoshi, Y., Matsumoto, M., … Itoh, H. (2022). Placental pathology predicts infantile neurodevelopment. *Scientific Reports*, *12*(1), 2578. https://doi.org/10.1038/s41598-022-06300-w

Wang, X., Miller, D. C., Harman, R., Antczak, D. F., and Clark, A. G. (2013). Paternally expressed genes predominate in the placenta. *Proceedings of the National Academy of Sciences*, *110*(26), 10705–10710. https://doi.org/10.1073/pnas.1308998110

Wei, Y., Yang, C.-R., Wei, Y.-P., Zhao, Z.-A., Hou, Y., Schatten, H., and Sun, Q.-Y. (2014). Paternally induced transgenerational inheritance of susceptibility to diabetes in mammals. *Proceedings of the National Academy of Sciences*, *111*(5), 1873–1878. https://doi.org/10.1073/pnas.1321195111

Williams, P. J., Bulmer, J. N., Searle, R. F., Innes, B. A., and Robson, S. C. (2009). Altered decidual leucocyte populations in the placental bed in pre-eclampsia and foetal growth restriction: a comparison with late normal pregnancy. *Reproduction (Cambridge, England)*, *138*(1), 177–184. https://doi.org/10.1530/REP-09-0007

Yi, L., Pimentel, H., Bray, N. L., and Pachter, L. (2018). Gene-level differential analysis at transcript-level resolution. *Genome Biology*, *19*(1), 1–11. https://doi.org/10.1186/s13059-018-1419-z

Yin, Y., Morgunova, E., Jolma, A., Kaasinen, E., Sahu, B., Khund-Sayeed, S., … Taipale, J. (2017). Impact of cytosine methylation on DNA binding specificities of human transcription factors. *Science (New York, N.Y.)*, *356*(6337). https://doi.org/10.1126/science.aaj2239

Yoshida, K., Maekawa, T., Ly, N. H., Fujita, S. ichiro, Muratani, M., Ando, M., … Ishii, S. (2020). ATF7-Dependent Epigenetic Changes Are Required for the Intergenerational Effect of a Paternal Low-Protein Diet. *Molecular Cell*, *78*(3), 445-458.e6. https://doi.org/10.1016/j.molcel.2020.02.028

[Editors’ note: what follows is the authors’ response to the second round of review.]

The manuscript has been improved, for which we thank you and acknowledge your efforts.However, there are some remaining issues that need to be addressed: the first reviewer suggests pointing out limitations of the study given the difference in the control food vs. the HFD food. The second reviewer strongly advocates for additional controls that would support the central claims. If you can provide these, we feel it would significantly strengthen the final paper.Reviewer #2 (Recommendations for the authors):The manuscript has been extensively revised to address the comments of myself and the other reviewers, and is substantially improved. The remaining major weakness, commented upon by myself and at least on other reviewer, was the fact that the control chow was not matched to the HFD chow, which impacts on interpretation of the study. This major limitation of the study could be discussed in more detail, as it is possible that the results could be at least partly due to non-fat differences between the control and HFD chow, and not entirely due to increased fat consumption (although the authors state that food intake was not measured, which is another weakness in the study that should be discussed) and/or the consequent state of obesity.

This reviewer states as a limitation of our study that the results we are showing may be due to non-fat differences between the standard chow and the high-fat diet (i.e. macro or micronutrient composition), rather than increased fat consumption. It is important to note that the goal of the present study was not to assess the effects of high-fat feeding, nor of varying macronutrient or micronutrient intake, but instead to study the effects of paternal *obesity* on the sperm epigenome, its links to placental development, gene expression, and cellular composition. The diets used elicited the desired effects on weight gain, excess adiposity, and metabolic outcomes, as described in our previously characterized model (https://pubmed.ncbi.nlm.nih.gov/35377412/). The standard chow diet has been used as a control in many previous studies that described diet-induced obesity models (for example: https://www.ncbi.nlm.nih.gov/pmc/articles/PMC4452998/#R24 https://pubmed.ncbi.nlm.nih.gov/23924601/ https://pubmed.ncbi.nlm.nih.gov/35377412/ https://pubmed.ncbi.nlm.nih.gov/28434881/).

Additionally, it was shown that a standard chow diet versus a purified low-fat diet show similar phenotypic, metabolic, and behavioral outcomes (https://pubmed.ncbi.nlm.nih.gov/28721750/). Furthermore, it was previously shown that dietary fat, and no other macronutrients, regulates increased adiposity in diet-induced obesity mouse models (https://pubmed.ncbi.nlm.nih.gov/30017356/). Importantly, given the current study was a follow-up on our previously published work (https://pubmed.ncbi.nlm.nih.gov/35377412/), it was necessary to be consistent with our characterized preconception paternal obesity mouse model, and therefore use the same experimental design such as the dietary treatments. Using the same experimental model allowed us to further characterize the model, demonstrate consistencies in obesity-associated changes in the sperm epigenome despite differences in experimental models, and newly demonstrate links between paternal preconception obesity, placenta development, and cellular composition.

We have added a sentence to highlight that we are not comparing two diets that are experimentally controlled for macro and micronutrients, but instead to induce paternal obesity and assess the effects on the sperm epigenome and placental development.

Lines 149-151: “Here we used the same pre-conception paternal obesity model as described in Jazwiec et al., (2022), where offspring were previously phenotyped for metabolic function and placenta was histopathologically characterized (Figure 1). Of note, this model does not compare diets that are experimentally controlled for macro or micronutrients, but instead is used to assess the effects of obesity on the sperm epigenome and its impacts on placental development.”

We had also highlighted that despite differences in paternal obesity models, including the use of the standard chow diet as a control (versus a low-fat diet matching for macro and micronutrients), we are able to replicate a significant number of obesity-associated changes in the sperm epigenome.

Lines 160-165: “Of the top 5% and top 10% differentially enriched H3K4me3 (deH3K4me3) regions, some were common to those previously identified in Pepin et al., (2022) (128 and 423 overlapping deH3K4me3 regions, Fisher’s exact test P=2.2e-16 and P=2.2e-16, Figure 2 —figure supplement 1 D i and ii, respectively). This was despite substantial differences in animal models (timing of diet exposure [3 vs 6 weeks of age], control diet [chow vs low-fat diet], and mouse substrain [C57BL/6J vs C57BL/6NCrl]).”

Reviewer #3 (Recommendations for the authors):The authors have updated and improved this manuscript and have provided important clarifications about some aspects. Most importantly, statistical analysis is now provided for overlaps between ChIP-seq and transcriptomic datasets. Additionally, a better discussion of the biological relevance of differentially enriched and differentially expressed genes is provided, as well as a better explanation of the deconvolution analysis.A concern that has not been addressed is the lack of validation studies. The authors argue that validation is not required because the datasets are robust and comparisons are statistically well supported. However, these arguments do not address the main reason for validation, which is to test the conclusion using a different assay in a different sample. Validation experiments would be especially valuable since in many cases the effect sizes are small, albeit statistically significant. To this end, RT-qPCR or Western blotting could be used to validate differential expression in placentas, and ChIP-qPCR could be used to validate differential H3K4me3 enrichment in sperm.A new concern is that after significance testing, there is no significant overlap between genes with differential H3K4me3 enrichment in sperm and differential expression in the placenta. However, because of the variable nature and small effect size of paternal epigenetic effects, this finding is not unexpected and the datasets and analysis are still interesting in demonstrating intergenerational regulatory effects of paternal diet.

Our previous study (https://pubmed.ncbi.nlm.nih.gov/35377412/) involved the characterization of a paternal preconception obesity model that revealed impacts on placental vascularization, hypoxia, and offspring metabolic functions. In the current study, we take a step further with genome-wide approaches to study the effects of paternal obesity on placental gene expression, we compare our findings with an intra-uterine growth restriction model, infer placental cell-type proportion changes using publicly available single-cell datasets that match our samples’ developmental time point and mouse strain, connect our findings with paternal obesity-associated changes on the sperm epigenome, and demonstrate consistent changes across different studies.

As a validation for the differential expression analysis in placentas, we show that some of the detected differentially expressed genes by RNA-seq in this study were also found to show significant changes associated by paternal obesity, as detected by RT-qPCR in the initial characterization of the model (*Igf1* and *Irs1*; https://pubmed.ncbi.nlm.nih.gov/35377412/).

We have added these findings in the text:

Lines 234-236: “Of note, the gene *Irs1* was found to be differentially expressed in placentas in association with paternal obesity, validating our previous findings in the initial characterization of the model (Jazwiec et al., 2022).”

Lines 295-296: “Similarly to our previous characterization of the model, here we detected differential expression of *Igf2* in placentas sired by obese sires (Jazwiec et al., 2022).

Lines 185-186: “Interestingly, *Igf2* was also previously shown to be differentially expressed in placentas derived from obese males (Jazwiec et al., 2022).”

Additionally, throughout the text we demonstrate findings that are consistent with our previous characterization of our model showing that paternal obesity is associated with a hypoxic placenta (https://pubmed.ncbi.nlm.nih.gov/35377412/). We show here that inferred changes in placental cell-type proportions associated with paternal obesity are indicative of a hypoxic placenta (Figure 5). Furthermore, we show significant overlap of DEGs in our placentas derived from obese sires, with DEGs in a model of hypoxia-induced IUGR, as well as similar associated changes in cell-type proportion (Figure 6).

As a validation for the H3K4me3 enrichment analysis in sperm, we show consistent changes on the sperm epigenome across our two different paternal obesity models. Specifically, we show a significant overlap of regions showing obesity-associated changes in H3K4me3 enrichment in sperm as detected in the current study and our other previous study (Figure 2 —figure supplement 1; https://pubmed.ncbi.nlm.nih.gov/35183795/). Furthermore, we show that the characteristic features of those impacted genomic regions are consistent across both datasets (Figure 2 and Figure 2 —figure supplement 1).